# TOWARDS FEW-SHOT ADAPTATION OF FOUNDATION MODELS VIA MULTITASK FINETUNING

**Zhuoyan Xu, Zhenmei Shi, Junyi Wei, Fangzhou Mu, Yin Li, Yingyu Liang**
University of Wisconsin-Madison
`{zhuoyan.xu,jwei53,fmu2,yin.li}@wisc.edu, {zhmeishi,yliang}@cs.wisc.edu`

## ABSTRACT

Foundation models have emerged as a powerful tool for many AI problems. Despite the tremendous success of foundation models, effective adaptation to new tasks, particularly those with limited labels, remains an open question and lacks theoretical understanding. An emerging solution with recent success in vision and NLP involves finetuning a foundation model on a selection of relevant tasks, before its adaptation to a target task with limited labeled samples. In this paper, we study the theoretical justification of this multitask finetuning approach. Our theoretical analysis reveals that with a diverse set of related tasks, this multitask finetuning leads to reduced error in the target task, in comparison to directly adapting the same pretrained model. We quantify the relationship between finetuning tasks and target tasks by diversity and consistency metrics, and further propose a practical task selection algorithm. We substantiate our theoretical claims with extensive empirical evidence. Further, we present results affirming our task selection algorithm adeptly chooses related finetuning tasks, providing advantages to the model performance on target tasks. We believe our study shed new light on the effective adaptation of foundation models to new tasks that lack abundant labels. Our code is available at https://github.com/OliverXUZY/Foudation-Model_Multitask.

## 1 INTRODUCTION

The advent of large-scale deep models trained on massive amounts of data has ushered in a new era of *foundation models* (Bommasani et al., 2021). These models, exemplified by large language models (e.g., BERT (Devlin et al., 2019) and GPT-3 (Brown et al., 2020)) and vision models (e.g., CLIP (Radford et al., 2021) and DINOv2 (Oquab et al., 2023)), offer the promise for adapting to a wide range of downstream tasks, and have led to some of the most exciting developments in AI to date, including the latest conversational AI — ChatGPT (OpenAI, 2022) and GPT4 (OpenAI, 2023). Despite encouraging empirical results (Zhang et al., 2020; Brown et al., 2020; Gao et al., 2021a), the effective adaptation of foundation models, especially to new tasks with limited labels, remains a practical challenge and lacks theoretical understanding.

In this paper, we focus on the problem of adapting a pretrained foundation model to a new task with a few labeled samples, where the target task can differ significantly from pretraining and the limited labeled data are insufficient for finetuning. This few-shot learning problem has been a long-standing challenge in machine learning (Wang et al., 2020). Prior approaches include learning from examples in the context prompt (in-context learning) (Brown et al., 2020), constructing simple classifiers based on the pretrained representation (Zhang et al., 2020), or finetuning the model using text prompts converted from labeled data (Gao et al., 2021a). An emerging solution involves finetuning a pretrained model on multiple auxiliary tasks pertaining to the target task. This multitask finetuning approach, related to meta learning (Hospedales et al., 2021), has been recently explored in NLP and vision (Murty et al., 2021; Vu et al., 2021; Zhong et al., 2021; Hu et al., 2022b; Chen et al., 2022; Min et al., 2022a). For example, latest studies (Sanh et al., 2022; Muennighoff et al., 2023) show that finetuning language models on a large set of tasks enables strong zero-shot generalization on unseen tasks. Nonetheless, the lack of sound theoretical explanations behind these previous approaches raises doubts about their ability to generalize on real-world tasks (Perez et al., 2021).

To bridge the gap, we study the theoretical justification of multitask finetuning. We consider an intermediate step that finetunes a pretrained model with a set of relevant tasks before adapting to a target task. Each of these auxiliary tasks might have a small number of labeled samples, and categories of these samples might not overlap with those on the target task. Our key intuition is that a sufficiently diverse set of relevant tasks can capture similar latent characteristics as the target task, thereby producing meaningful representation and reducing errors in the target task. To this end, we present rigorous theoretical analyses, provide key insight into conditions necessary for successful multitask finetuning, and introduce a novel algorithm for selecting tasks suitable for finetuning.

Our key contributions are three folds. *Theoretically*, we present a framework for analyzing pretraining followed by multitask finetuning. Our analysis (Section 3) reveals that with limited labeled data from diverse tasks, finetuning can improve the prediction performance on a downstream task. *Empirically*, we perform extensive experiments on both vision and language tasks (Section 4) to verify our theorem. Our results suggest that our theorem successfully predicts the behavior of multitask finetuning across datasets and models. *Practically*, inspired by our theorem, we design *a task selection algorithm* for multitask finetuning. On the Meta-Dataset (Triantafillou et al., 2020), our algorithm shows significantly improved results in comparison to finetuning using all possible tasks.

## 1.1 RELATED WORK

We briefly summarize related work and refer our readers to Appendix B for a detailed discussion.

Foundation models (Bommasani et al., 2021) are typically pretrained over broad data using approaches include *contrastive learning* (Oord et al., 2018; Chen et al., 2020; He et al., 2020; Tian et al., 2020a; Grill et al., 2020; Radford et al., 2021) in vision and *masked modeling* in NLP (Devlin et al., 2019; Liu et al., 2019). Adapting foundation models to downstream target tasks has received significant attention, e.g., minorly finetuning (Vinyals et al., 2016; Chen et al., 2020; He et al., 2020; 2022), prompt-based finetuning (Gao et al., 2021a; Hu et al., 2022a), prompt tuning (Lester et al., 2021; Li & Liang, 2021), and in-context learning (Min et al., 2022b; Wei et al., 2022a). Our work studies an emerging solution of multitask finetuning (Zhong et al., 2021; Sanh et al., 2022; Min et al., 2022a; Chen et al., 2022; Wang et al., 2023), which finetunes the pretrained foundation model using multiple relevant tasks before adapting to the target task. Multitask finetuning has been shown to induce zero-shot generalization in large language models (Sanh et al., 2022; Muennighoff et al., 2023), and enable parameter efficient tuning by prompt tuning (Wang et al., 2023). Our work seeks to provide theoretical justification to prior approaches. Part of our results also confirm previous findings.

A line of theoretical work provides the error bound of the target task in terms of sample complexity (Du et al., 2021; Tripuraneni et al., 2021; Shi et al., 2023a). Their work mainly analyzed representations from supervised pretraining using multitasks. In contrast, our work considers representations from self-supervised pretraining, and focuses on multitask finetuning. Our approach and analysis guarantee that limited but diverse finetuning data can improve the prediction performance on a target task with novel classes. On the other hand, with only limited target labels, few-shot learning necessitates the generalization to new tasks (Wang et al., 2020; Yang et al., 2022). Direct training with limited data is prone to overfitting. Meta learning offers a promising solution that allows the model to adapt to the few-shot setting (Snell et al., 2017; Finn et al., 2017; Raghu et al., 2020; Chen et al., 2021b; Hu et al., 2022b). Inspired by meta learning, our analysis extends the idea of multitask finetuning by providing sound theoretic justifications and demonstrating strong empirical results. We further introduce a task selection algorithm that bridges our theoretical findings with practical multitask finetuning.

## 2 BACKGROUND: MULTITASK FINETUNING FOR FEW-SHOT LEARNING

This section reviews the pretraining of foundation models and adaptation for few-shot learning, and then formalizes the multitask finetuning approach.

**Pretraining Foundation Models.** We consider three common pretraining methods: contrastive learning, masked language modeling, and supervised pretraining. *Contrastive learning* is widely considered in vision and multi-modal tasks. This approach pretrains a model $\phi$ from a hypothesis class $\Phi$ of foundation models via loss on contrastive pairs generated from data points $x$. First sample a point $x$ and then apply some transformation to obtain $x^+$; independently sample another point $x^-$. The population contrastive loss is then $\mathcal{L}_{con-pre}(\phi) := \mathbb{E}\left[\ell_u\left(\phi(x)^\top\left(\phi(x^+) - \phi(x^-)\right)\right)\right]$,

where the loss function $\ell_u$ is a non-negative decreasing function. In particular, logistic loss $\ell_u(v) = \log(1 + \exp(-v))$ recovers the typical contrastive loss in most empirical work (Logeswaran & Lee, 2018; Oord et al., 2018; Chen et al., 2020). *Masked language modeling* is a popular self-supervised learning approach in NLP. It can be regarded as a kind of *supervised pretraining*: the masked word is viewed as the class (see Appendix C for more details). In what follows we provide a unified formulation. On top of the representation function $\phi$, there is a linear function $f \in \mathcal{F} \subset \{\mathbb{R}^d \to \mathbb{R}^K\}$ predicting the labels where $K$ is the number of classes. The supervised loss is: $\mathcal{L}_{sup-pre}(\phi) := \min_{f \in \mathcal{F}} \mathbb{E}[\ell(f \circ \phi(x), y)]$, where $\ell(\cdot, y)$ is the cross-entropy loss. To simplify the notation, we unify $\mathcal{L}_{pre}(\phi)$ as the pretraining loss.

**Adapting Models for Few-shot Learning.** A pretrained foundation model $\phi$ can be used for downstream target tasks $\mathcal{T}$ by learning linear classifiers on $\phi$. We focus on binary classification (the general multiclass setting is in Appendix D). A linear classifier on $\phi$ is given by $\boldsymbol{w}^\top \phi(x)$ where $\boldsymbol{w} \in \mathbb{R}^d$. The supervised loss of $\phi$ w.r.t the task $\mathcal{T}$ is then:

$$\mathcal{L}_{sup}(\mathcal{T}, \phi) := \min_{\boldsymbol{w}} \mathbb{E}_{(x,y) \sim \mathcal{D}_{\mathcal{T}}}\left[\ell\left(\boldsymbol{w}^\top \phi(x), y\right)\right], \tag{1}$$

where $\mathcal{D}_{\mathcal{T}}(x, y)$ is the distribution of data $(x, y)$ in task $\mathcal{T}$. In few-shot learning with novel classes, there are *limited labeled data points* for learning the linear classifier. Further, the target task $\mathcal{T}_0$ may contain *classes different from those in pretraining*. We are interested in obtaining a model $\phi$ such that $\mathcal{L}_{sup}(\mathcal{T}_0, \phi)$ is small.

**Multitask Finetuning.** In the challenging setting of few-shot learning, the data in the target task is limited. On the other hand, we can have prior knowledge of the target task characteristics and its associated data patterns, and thus can collect additional data from relevant and accessible sources when available. Such data may cover the patterns in target task and thus can be used as auxiliary tasks to finetune the pretrained model before adaptation to the target task. Here we formalize this idea in a general form and provide analysis in later sections. Formally, suppose we have $M$ auxiliary tasks $\{\mathcal{T}_1, \mathcal{T}_2, \ldots, \mathcal{T}_M\}$, each with $m$ labeled samples $\mathcal{S}_i := \{(x_j^i, y_j^i) : j \in [m]\}$. The finetuning data are $\mathcal{S} := \cup_{i \in [M]} \mathcal{S}_i$. Given a pretrained model $\hat{\phi}$, we further finetune it using the objective:

$$\min_{\phi \in \Phi} \frac{1}{M} \sum_{i=1}^{M} \widehat{\mathcal{L}}_{sup}(\mathcal{T}_i, \phi), \quad \text{where} \quad \widehat{\mathcal{L}}_{sup}(\mathcal{T}_i, \phi) := \min_{\boldsymbol{w}_i \in \mathbb{R}^d} \frac{1}{m} \sum_{j=1}^{m} \ell(\boldsymbol{w}_i^\top \phi(x_j^i), y_j^i). \tag{2}$$

This can be done via gradient descent from the initialization $\hat{\phi}$ (see Algorithm 2 in the Appendix). Multitask finetuning is conceptually simple, and broadly applicable to different models and datasets. While its effectiveness has been previously demonstrated (Murty et al., 2021; Vu et al., 2021; Zhong et al., 2021; Hu et al., 2022b; Chen et al., 2022; Min et al., 2022a; Sanh et al., 2022; Muennighoff et al., 2023), the theoretical justification remains to be fully investigated and understood.

## 3    THEORETICAL ANALYSIS: BENEFIT OF MULTITASK FINETUNING

To understand the potential benefit of multitask finetuning, we will compare the performance of $\hat{\phi}$ (from pretraining) and $\phi'$ (from pretraining and multitask finetuning) on a target task $\mathcal{T}_0$. That is, we will compare $\mathcal{L}_{sup}(\mathcal{T}_0, \hat{\phi})$ and $\mathcal{L}_{sup}(\mathcal{T}_0, \phi')$, where $\mathcal{L}_{sup}(\mathcal{T}, \phi)$ is the population supervised loss of $\phi$ on the task $\mathcal{T}$ defined in Equation (1). For the analysis, we first formalize the data distributions and learning models, then introduce the key notions, and finally present the key theorems.

**Data Distributions.** Let $\mathcal{X}$ be the input space and $\overline{\mathcal{Z}} \subseteq \mathbb{R}^d$ be the output space of the foundation model. Following Arora et al. (2019), suppose there is a set of latent classes $\mathcal{C}$ with $|\mathcal{C}| = K$, and a distribution $\eta$ over the classes; each class $y \in \mathcal{C}$ has a distribution $\mathcal{D}(y)$ over inputs $x$. In pretraining using contrastive learning, the distribution $\mathcal{D}_{con}(\eta)$ of the contrastive data $(x, x^+, x^-)$ is given by: $(y, y^-) \sim \eta^2$ and $x, x^+ \sim \mathcal{D}(y)$, $x^- \sim \mathcal{D}(y^-)$. In masked self-supervised or fully supervised pretraining, $(x, y)$ is generated by $y \sim \eta, x \sim \mathcal{D}(y)$. In a task $\mathcal{T}$ with binary classes $\{y_1, y_2\}$, the data distribution $\mathcal{D}_{\mathcal{T}}(x, y)$ is by first uniformly drawing $y \in \{y_1, y_2\}$ and then drawing $x \sim \mathcal{D}(y)$. Finally, let $\zeta$ denote the conditional distribution of $(y_1, y_2) \sim \eta^2$ conditioned on $y_1 \neq y_2$, and suppose the tasks in finetuning are from $\zeta$. Note that in few-shot learning with novel classes, the target task's classes may not be the same as those in the pretraining. Let $\mathcal{C}_0$ be the set of possible classes in the target task, which may or may not overlap with $\mathcal{C}$.

**Learning Models.** Recall that $\Phi$ is the hypothesis class of foundation models $\phi : \mathcal{X} \to \overline{\mathcal{Z}}$. To gauge the generalization performance, let $\phi^* \in \Phi$ denote the model with the lowest target task loss $\mathcal{L}_{sup}(\mathcal{T}_0, \phi^*)$ and $\phi_\zeta^* \in \Phi$ denote the model with the lowest average supervised loss over the set of auxiliary tasks $\mathcal{L}_{sup}(\phi_\zeta^*) := \mathbb{E}_{\mathcal{T} \sim \zeta}[\mathcal{L}_{sup}(\mathcal{T}, \phi_\zeta^*)]$. Note that if all $\phi \in \Phi$ have high supervised losses, we cannot expect the method to lead to a good generalization performance, and thus we need to calibrate w.r.t. $\phi^*$ and $\phi_\zeta^*$. We also need some typical regularity assumptions.

**Assumption 1** (Regularity Assumptions). $\|\phi\|_2 \leq R$ and linear operator $\|\boldsymbol{w}\|_2 \leq B$. The loss $\ell_u$ is bounded in $[0, C]$ and $L$-Lipschitz. The supervised loss $\mathcal{L}_{sup}(\mathcal{T}, \phi)$ is $\tilde{L}$-Lipschitz with respect to $\phi$.

**Diversity and Consistency.** Central to our theoretical analysis lies in the definitions of *diversity* in auxiliary tasks used for finetuning and their *consistency* with the target task.

**Definition 1** (Diversity). *The averaged representation difference for two model $\phi, \tilde{\phi}$ on a distribution $\zeta$ over tasks is $\bar{d}_\zeta(\phi, \tilde{\phi}) := \mathbb{E}_{\mathcal{T} \sim \zeta} \left[ \mathcal{L}_{sup}(\mathcal{T}, \phi) - \mathcal{L}_{sup}(\mathcal{T}, \tilde{\phi}) \right] = \mathcal{L}_{sup}(\phi) - \mathcal{L}_{sup}(\tilde{\phi})$. The worst-case representation difference between representations $\phi, \tilde{\phi}$ on the family of classes $\mathcal{C}_0$ is $d_{\mathcal{C}_0}(\phi, \tilde{\phi}) := \sup_{\mathcal{T}_0 \subseteq \mathcal{C}_0} \left| \mathcal{L}_{sup}(\mathcal{T}_0, \phi) - \mathcal{L}_{sup}(\mathcal{T}_0, \tilde{\phi}) \right|$. We say the model class $\Phi$ has $\nu$-diversity (for $\zeta$ and $\mathcal{C}_0$) with respect to $\phi_\zeta^*$, if for any $\phi \in \Phi$, $d_{\mathcal{C}_0}(\phi, \phi_\zeta^*) \leq \bar{d}_\zeta(\phi, \phi_\zeta^*)/\nu$.*

Such diversity notion has been proposed and used to derive statistical guarantees (e.g., Tripuraneni et al. (2020); Zhao et al. (2023)). Intuitively, diversity measures whether the data from $\zeta$ covers the characteristics of the target data in $\mathcal{C}_0$, e.g., whether the span of the linear mapping solutions $\boldsymbol{w}$'s for tasks from $\zeta$ can properly cover the solutions for tasks from $\mathcal{C}_0$ (Zhao et al., 2023). Existing work showed that diverse pretraining data will lead to a large diversity parameter $\nu$ and can improve the generalization in the target task. Our analysis will show the diversity in finetuning tasks from $\zeta$ can benefit the performance of a target task from $\mathcal{C}_0$.

**Definition 2** (Consistency). *We say the model class $\Phi$ has $\kappa$-consistency (for $\zeta$ and $\mathcal{C}_0$) with respect to $\phi^*$ and $\phi_\zeta^*$, where $\kappa := \sup_{\mathcal{T}_0 \subseteq \mathcal{C}_0} \left[ \mathcal{L}_{sup}(\mathcal{T}_0, \phi_\zeta^*) - \mathcal{L}_{sup}(\mathcal{T}_0, \phi^*) \right]$.*

This consistency notion measures the similarity between the data in tasks from $\zeta$ and the data in the target task from $\mathcal{C}_0$. Intuitively, when the tasks from $\zeta$ are similar to the target task $\mathcal{T}_0$, their solutions $\phi_\zeta^*$ and $\phi^*$ will be similar to each other, resulting in a small $\kappa$. Below we will derive guarantees based on the diversity $\nu$ and consistency $\kappa$ to explain the gain from multitask finetuning.

**Key Results.** We now present the results for a uniform distribution $\eta$, and include the full proof and results for general distributions in Appendix C and Appendix D. Recall that we will compare the performance of $\hat{\phi}$ (the model from pretraining) and $\phi'$ (the model from pretraining followed by multitask finetuning) on a target task $\mathcal{T}_0$. For $\hat{\phi}$ without multitask finetuning, we have:

---

**Theorem 3.1.** (No Multitask Finetuning) *Assume Assumption 1 and that $\Phi$ has $\nu$-diversity and $\kappa$-consistency with respect to $\phi^*$ and $\phi_\zeta^*$. Suppose $\hat{\phi}$ satisfies $\hat{\mathcal{L}}_{pre}(\hat{\phi}) \leq \epsilon_0$. Let $\tau := \Pr_{(y_1, y_2) \sim \eta^2} \{y_1 = y_2\}$. Then for any target task $\mathcal{T}_0 \subseteq \mathcal{C}_0$,*

$$\mathcal{L}_{sup}(\mathcal{T}_0, \hat{\phi}) - \mathcal{L}_{sup}(\mathcal{T}_0, \phi^*) \leq \frac{1}{\nu} \left[ \frac{2\epsilon_0}{1 - \tau} - \mathcal{L}_{sup}(\phi_\zeta^*) \right] + \kappa. \qquad (3)$$

---

In Theorem 3.1, $\widehat{\mathcal{L}}_{pre}(\phi)$ is the empirical loss of $\mathcal{L}_{pre}(\phi)$ with pretraining sample size $N$. We now consider $\phi'$ obtained by multitask finetuning. Define the subset of models with pretraining loss smaller than $\tilde{\epsilon}$ as $\Phi(\tilde{\epsilon}) := \left\{ \phi \in \Phi : \widehat{\mathcal{L}}_{pre}(\phi) \leq \tilde{\epsilon} \right\}$. Recall the Rademacher complexity of $\Phi$ on $n$ points is $\mathcal{R}_n(\Phi) := \mathbb{E}_{\{\sigma_j\}_{j=1}^n, \{x_j\}_{j=1}^n} \left[ \sup_{\phi \in \Phi} \sum_{j=1}^n \sigma_j \phi(x_j) \right]$.

Theorem 3.2 below showing that the target prediction performance of the model $\phi'$ from multitask finetuning can be significantly better than that of $\hat{\phi}$ without multitask finetuning. In particular, achieves an error reduction $\frac{1}{\nu} \left[ (1 - \alpha) \frac{2\epsilon_0}{1 - \tau} \right]$. The reduction is achieved when multitask finetuning is solved to a small loss $\epsilon_1$ for a small $\alpha$ on sufficiently many finetuning data.

**Theorem 3.2.** (With Multitask Finetuning) *Assume Assumption 1 and that $\Phi$ has $\nu$-diversity and $\kappa$-consistency with respect to $\phi^*$ and $\phi^*_\zeta$. Suppose for some constant $\alpha \in (0,1)$, we solve Equation (2) with empirical loss lower than $\epsilon_1 = \frac{\alpha}{3}\frac{2\epsilon_0}{1-\tau}$ and obtain $\phi'$. For any $\delta > 0$, if for $\tilde{\epsilon} = \widehat{\mathcal{L}}_{pre}(\phi')$,*

$$M \geq \frac{1}{\epsilon_1}\left[4\sqrt{2}\tilde{L}\mathcal{R}_M(\Phi(\tilde{\epsilon})) + \frac{4C^2}{\epsilon_1}\log(\frac{2}{\delta})\right], Mm \geq \frac{1}{\epsilon_1}\left[16LB\mathcal{R}_{Mm}(\Phi(\tilde{\epsilon})) + \frac{4C^2}{\epsilon_1}\log(\frac{2}{\delta})\right],$$

*then with probability $1 - \delta$, for any target task $\mathcal{T}_0 \subseteq \mathcal{C}_0$,*

$$\mathcal{L}_{sup}(\mathcal{T}_0, \phi') - \mathcal{L}_{sup}(\mathcal{T}_0, \phi^*) \leq \frac{1}{\nu}\left[\alpha\frac{2\epsilon_0}{1-\tau} - \mathcal{L}_{sup}(\phi^*_\zeta)\right] + \kappa. \tag{4}$$

The requirement is that the number of tasks $M$ and the total number of labeled samples $Mm$ across tasks are sufficiently large. This implies when $M$ is above the threshold, the total size $Mm$ determines the performance, and increasing either $M$ or $m$ while freezing the other can improve the performance. We shall verify these findings in our experiments (Section 4.1).

Theorem 3.2 also shows the conditions for successful multitask finetuning, in particular, the impact of the diversity and consistency of the finetuning tasks. Besides small finetuning loss on sufficiently many data, a large diversity parameter $\nu$ and a small consistency parameter $\kappa$ will result in a small target error bound. Ideally, data from the finetuning tasks should be similar to those from the target task, but also sufficiently diverse to cover a wide range of patterns that may be encountered in the target task. This inspires us to perform finer-grained analysis of diversity and consistency using a simplified data model (Section 3.1), which sheds light on the design of an algorithm to select a subset of finetuning tasks with better performance (Section 3.2).

### 3.1 CASE STUDY OF DIVERSITY AND CONSISTENCY

Our main results, rooted in notions of diversity and consistency, state the general conclusion of multitask finetuning on downstream tasks. A key remaining question is how relevant tasks should be selected for multitask finetuning in practice. Our intuition is that this task selection should promote both diversity (encompassing the characteristics of the target task) and consistency (focusing on the relevant patterns in achieving the target task's objective). To illustrate such theoretical concepts and connect them to practical algorithms, we specialize the general conclusion to settings that allow easy interpretation of diversity and consistency. In this section, we provide a toy linear case study and we put the proof and also the analysis of a more general setting in Appendix E, e.g., more general latent class $\mathcal{C}, \mathcal{C}_0$, more general distribution $\zeta$, input data with noise.

In what follows, we specify the data distributions and function classes under consideration, and present an analysis for this case study. Our goal is to explain the intuition behind diversity and consistency notions: *diversity is about coverage, and consistency is about similarity in the latent feature space*. This can facilitate the design of task selection algorithms.

**Linear Data and Tasks.** Inspired by classic dictionary learning and recent analysis on representation learning (Wen & Li, 2021; Shi et al., 2023a), we consider the latent class/representation setting where each latent class $z \in \{0, -1, +1\}^d$ is represented as a feature vector. We focus on individual binary classification tasks, where $\mathcal{Y} = \{-1, +1\}$ is the label space. Thus, each task has two latent classes $z, z'$ (denote the task as $\mathcal{T}_{z,z'}$) and we randomly assign $-1$ and $+1$ to each latent class. Namely, $\mathcal{T}_{z,z'}$ is defined

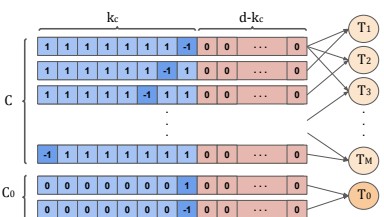

Figure 1: Illustration of features in linear data. Blue are the features encoded in $\mathcal{C}$ while red is not.

as: $x = \begin{cases} z, & \text{if } y = -1 \\ z', & \text{if } y = +1 \end{cases}$. We show a diagram in Figure 1, we denote each task containing two latent classes, namely $(z, z')$. Each task in diagram can be represented as $(T_1$ to $\mathcal{T}_{z_1, z'_1},$ $T_2$ to $\mathcal{T}_{z_2, z'_2})$. We further assume a balanced class setting in all tasks, i.e., $p(y = -1) = $

$p(y = +1) = \frac{1}{2}$. Now, we define the latent classes seen in multitask finetuning tasks: $\mathcal{C} =$

$$\left\{ (\underbrace{1, 1, \ldots, 1, 1, -1}_{k_\mathcal{C}}, \underbrace{0, \ldots, 0}_{d - k_\mathcal{C}})^\top, (\underbrace{1, 1, \ldots, 1, -1, 1}_{k_\mathcal{C}}, \underbrace{0, \ldots, 0}_{d - k_\mathcal{C}})^\top, \ldots, (\underbrace{-1, 1, \ldots, 1, 1, 1}_{k_\mathcal{C}}, \underbrace{0, \ldots, 0}_{d - k_\mathcal{C}})^\top \right\}.$$

Note that their feature vectors only encode the first $k_\mathcal{C}$ features, and $|\mathcal{C}| = k_\mathcal{C}$. We let $\mathcal{C}_0 := \{z^{(1)}, z^{(2)}\} \subseteq \{0, -1, +1\}^d$ which is used for the target task, and assume that $z^{(1)}$ and $z^{(2)}$ only differ in 1 dimension, i.e., the target task can be done using this one particular dimension. Let $\zeta$ be a distribution uniformly sampling two different latent classes from $\mathcal{C}$. Then, our data generation pipeline for getting a multitask finetuning task is (1) sample two latent classes $(z, z') \sim \zeta$; (2) assign label $-1, +1$ to two latent classes.

**Linear Model and Loss Function.** We consider a linear model class with regularity Assumption 1, i.e., $\Phi = \{\phi \in \mathbb{R}^{d \times d} : \|\phi\|_F \leq 1\}$ and linear head $w \in \mathbb{R}^d$ where $\|w\|_2 \leq 1$. Thus, the final output of the model and linear head is $w^\top \phi x$. We use the loss in Shi et al. (2023a), i.e., $\ell(w^\top \phi x, y) = -y w^\top \phi x$.

**Remark 3.1.** *Although we have linear data, linear model, and linear loss, $\mathcal{L}_{sup}(\phi)$ is a non-linear function on $\phi$ as the linear heads are different across tasks, i.e., each task has its own linear head.*

Now we can link our diversity and consistency to features encoded by training or target tasks.

---

**Theorem 3.3** (Diversity and Consistency). *If $\mathcal{C}$ encodes the feature in $\mathcal{C}_0$, i.e., the different entry dimension of $z^{(1)}$ and $z^{(2)}$ in $\mathcal{C}_0$ is in the first $k_\mathcal{C}$ dimension, then we have $\nu$ is lower bounded by constant $\tilde{c} \geq \frac{2\sqrt{2} - 2}{k_\mathcal{C} - 1}$ and $\kappa \leq 1 - \sqrt{\frac{1}{k_\mathcal{C}}}$. Otherwise, we have $\nu \to 0$ and $\kappa \geq 1$.*

---

Theorem 3.3 establishes $\tilde{c}$-diversity and $\kappa$-consistency in Definition 1 and Definition 2. The analysis shows that diversity can be intuitively understood as the coverage of the finetuning tasks on the target task in the latent feature space: If the key feature dimension of the target task is covered by the features encoded by finetuning tasks, then we have lower-bounded diversity $\nu$; if not covered, then the diversity $\nu$ tends to 0 (leading to vacuous error bound in Theorem 3.2). Also, consistency can be intuitively understood as similarity in the feature space: when $k_\mathcal{C}$ is small, a large fraction of the finetuning tasks are related to the target task, leading to a good consistency (small $\kappa$); when $k_\mathcal{C}$ is large, we have less relevant tasks, leading to a worse consistency. Such an intuitive understanding of diversity and consistency will be useful for designing practical task selection algorithms.

## 3.2 TASK SELECTION

Our analysis suggests that out of a pool of candidate tasks, a subset $S$ with good consistency (i.e., small $\kappa$) and large diversity (i.e., large $\nu$) will yield better generalization to a target task. To realize this insight, we present a greedy selection approach, which sequentially adds tasks with the best consistency, and stops when there is no significant increase in the diversity of the selected subset. In doing so, our approach avoids enumerating all possible subsets and thus is highly practical.

A key challenge is to compute the consistency and diversity of the data. While the exact computation deems infeasible, we turn to approximations that capture the key notions of consistency and diversity. We show a simplified diagram for task selection in Figure 2. Specifically, given a foundation

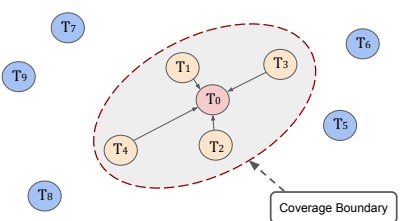

Figure 2: Illustration of the similarity and coverage. Target tasks ($\mathcal{T}_0$) with the most similar tasks in yellow and the rest in blue. The ellipsoid spanned by yellow tasks is the coverage for the target task. Adding more tasks in blue to the ellipsoid does not increase the coverage boundary.

model $\phi$, we assume any task data $\mathcal{T} = \{x_j\}$ follows a Gaussian distribution in the representation space: let $\phi(\mathcal{T}) = \{\phi(x_j)\}$ denote the representation vectors obtained by applying $\phi$ on the data points in $\mathcal{T}$; compute the sample mean $\mu_\mathcal{T}$ and covariance $C_\mathcal{T}$ for $\phi(\mathcal{T})$, and view it as the Gaussian $\mathcal{N}(\mu_\mathcal{T}, C_\mathcal{T})$. Further, following the intuition shown in the case study, we simplify consistency to similarity: for the target task $\mathcal{T}_0$ and a candidate task $\mathcal{T}_i$, if the co-

sine similarity $\text{CosSim}(\mathcal{T}_0, \mathcal{T}_i) := \mu_{\mathcal{T}_0}^\top \mu_{\mathcal{T}_i} / (\|\mu_{\mathcal{T}_0}\|_2 \|\mu_{\mathcal{T}_i}\|_2)$ is large, we view $\mathcal{T}_i$ as consistent with $\mathcal{T}_0$. Next, we simplify diversity to coverage: if a dataset $D$ (as a collection of finetuning tasks) largely "covers" the target data $\mathcal{T}_0$, we view $D$ as diverse for $\mathcal{T}_0$. Regarding the task data as Gaussians, we note that the covariance ellipsoid of $D$ covers the target data $\mu_{\mathcal{T}_0}$ iff $(\mu_D - \mu_{\mathcal{T}_0})^T C_D^{-1} (\mu_D - \mu_{\mathcal{T}_0}) \leq 1$. This inspires us to define the following coverage score as a heuristic for diversity: $\text{coverage}(D; \mathcal{T}_0) := 1/(\mu_D - \mu_{\mathcal{T}_0})^\top C_D^{-1} (\mu_D - \mu_{\mathcal{T}_0})$.

Using these heuristics, we arrive at the following selection algorithm: sort the candidate task in descending order of their cosine similarities to the target data; sequentially add tasks in the sorted order to $L$ until $\text{coverage}(L; \mathcal{T}_0)$ has no significant increase. Algorithm 1 illustrates this key idea.

---

**Algorithm 1** Consistency-Diversity Task Selection

---

**Input:** Target task $\mathcal{T}_0$, candidate finetuning tasks: $\{\mathcal{T}_1, \mathcal{T}_2, \ldots, \mathcal{T}_M\}$, model $\phi$, threshold $p$.
1: Compute $\phi(\mathcal{T}_i)$ and $\mu_{\mathcal{T}_i}$ for $i = 0, 1, \ldots, M$.
2: Sort $\mathcal{T}_i$'s in descending order of similarity $(\mathcal{T}_0, \mathcal{T}_i)$. Denote the sorted list as $\{\mathcal{T}_1', \mathcal{T}_2', \ldots, \mathcal{T}_M'\}$.
3: $L \leftarrow \{\mathcal{T}_1'\}$
4: **for** $i = 2, \ldots, M$ **do**
5:    If $\text{coverage}(L \cup \mathcal{T}_i'; \mathcal{T}_0) \geq (1 + p) \cdot \text{coverage}(L; \mathcal{T}_0)$, then $L \leftarrow L \cup \mathcal{T}_i'$; otherwise, break.
6: **end for**
**Output:** selected data $L$ for multitask finetuning.

---

## 4 EXPERIMENTS

We now present our main results, organized in three parts. Section 4.1 explores how different numbers of finetuning tasks and samples influence the model's performance, offering empirical backing to our theoretical claims. Section 4.2 investigates whether our task selection algorithm can select suitable tasks for multitask finetuning. Section 4.3 provides a more extensive exploration of the effectiveness of multitask finetuning on various datasets and pretrained models. We defer other results to the appendix. Specifically, Appendix F.4.1 shows that better diversity and consistency of finetuning tasks yield improved performance on target tasks under same sample complexity. Appendix F.4.2 shows that finetuning tasks satisfying diversity yet without consistency lead to no performance gain even with increased sample complexity. Further, Appendix G and Appendix H present additional experiments using NLP and vision-language models, respectively.

**Experimental Setup.** We use four few-shot learning benchmarks: miniImageNet (Vinyals et al., 2016), tieredImageNet (Ren et al., 2018), DomainNet (Peng et al., 2019) and Meta-dataset (Triantafillou et al., 2020). We use foundation models with different pretraining schemes (MoCo-v3 (Chen et al., 2021a), DINO-v2 (Oquab et al., 2023), and supervised learning with ImageNet (Russakovsky et al., 2015)) and architectures (ResNet (He et al., 2016) and ViT (Dosovitskiy et al., 2021)). We consider few-shot tasks consisting of $N$ classes with $K$ support samples and $Q$ query samples per class (known as $N$-way $K$-shot). The goal is to classify the query samples based on the support samples. Tasks used for finetuning are constructed by samples from the training split. Each task is formed by randomly sampling 15 classes, with every class drawing 1 or 5 support samples and 10 query samples. Target tasks are similarly constructed from the test set. We follow (Chen et al., 2021b) for multitask finetuning and target task adaptation. During multitask finetuning, we update all parameters in the model using a nearest centroid classifier, in which all samples are encoded, class centroids are computed, and cosine similarity between a query sample and those centroids are treated as the class logits For adaptation to a target task, we only retain the model encoder and consider a similar nearest centroid classifier. This multitask finetuning protocol applies to all experiments (Sections 4.1 to 4.3). We provide full experimental set up in Appendix F.

### 4.1 VERIFICATION OF THEORETICAL ANALYSIS

We conduct experiments on the tieredImageNet dataset to confirm the key insight from our theorem — the impact of the number of finetuning tasks ($M$) and the number of samples per task ($m$).

**Results.** We first investigate the influence of the number of shots. We fix the target task as a 1-shot setting but vary the number of shots from 1 to 4 in finetuning, and vary the total sample size

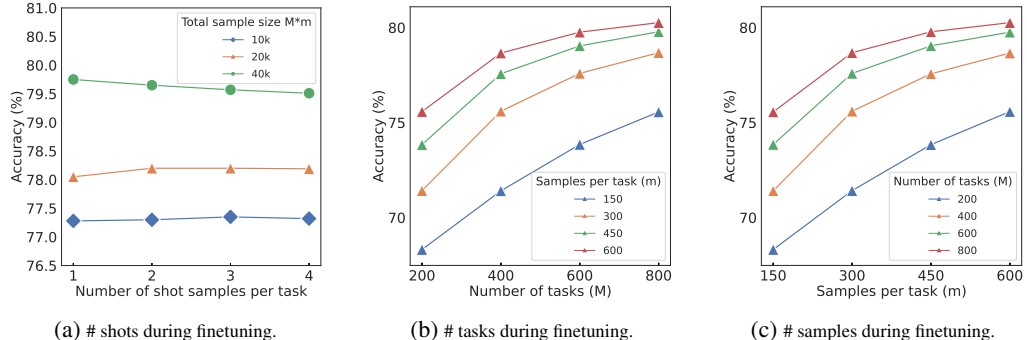

(a) # shots during finetuning.  (b) # tasks during finetuning.  (c) # samples during finetuning.

Figure 3: Results on ViT-B backbone pretrained by MoCo v3. (a) Accuracy v.s. number of shots per finetuning task. Different curves correspond to different total numbers of samples $Mm$. (b) Accuracy v.s. the number of tasks $M$. Different curves correspond to different numbers of samples per task $m$. (c) Accuracy v.s. number of samples per task $m$. Different curves correspond to different numbers of tasks $M$.

| Pretrained | Selection | INet | Omglot | Acraft | CUB | QDraw | Fungi | Flower | Sign | COCO |
|---|---|---|---|---|---|---|---|---|---|---|
| CLIP | Random | 56.29 | 65.45 | 31.31 | 59.22 | 36.74 | 31.03 | 75.17 | 33.21 | 30.16 |
| | No Con. | 60.89 | 72.18 | 31.50 | 66.73 | 40.68 | 35.17 | 81.03 | 37.67 | 34.28 |
| | No Div. | 56.85 | 73.02 | 32.53 | 65.33 | 40.99 | 33.10 | 80.54 | 34.76 | 31.24 |
| | Selected | **60.89** | **74.33** | **33.12** | **69.07** | **41.44** | **36.71** | 80.28 | **38.08** | **34.52** |
| DINOv2 | Random | 83.05 | 62.05 | 36.75 | 93.75 | 39.40 | 52.68 | 98.57 | 31.54 | 47.35 |
| | No Con. | 83.21 | 76.05 | 36.32 | 93.96 | 50.76 | 53.01 | 98.58 | 34.22 | 47.11 |
| | No Div. | 82.82 | 79.23 | 36.33 | 93.96 | 55.18 | 52.98 | 98.59 | 35.67 | 44.89 |
| | Selected | **83.21** | **81.74** | **37.01** | **94.10** | **55.39** | **53.37** | **98.65** | **36.46** | **48.08** |
| MoCo v3 | Random | 59.66 | 60.72 | 18.57 | 39.80 | 40.39 | 32.79 | 58.42 | 33.38 | 32.98 |
| | No Con. | 59.80 | 60.79 | 18.75 | 40.41 | 40.98 | 32.80 | 59.55 | 34.01 | 33.41 |
| | No Div. | 59.57 | 63.00 | 18.65 | 40.36 | 41.04 | 32.80 | 58.67 | 34.03 | 33.67 |
| | Selected | **59.80** | **63.17** | **18.80** | **40.74** | **41.49** | **33.02** | **59.64** | **34.31** | **33.86** |

Table 1: Results evaluating our task selection algorithm on Meta-dataset using ViT-B backbone. No Con.: Ignore consistency. No Div.: Ignore diversity. Random: Ignore both consistency and diversity.

$Mm = [10k, 20k, 40k]$. The results in Figure 3a show no major change in accuracy with varying the number of shots in finetuning. It is against the common belief that meta-learning like Prototypical Networks (Snell et al., 2017) has to mimic the exact few-shot setting and that a mismatch will hurt the performance. The results also show that rather than the number of shots, the total sample size $Mm$ determines the performance, which is consistent with our theorem. We next investigate the influence of $M$ and $m$. We vary the number of tasks ($M = [200, 400, 600, 800]$) and samples per task ($m = [150, 300, 450, 600]$) while keeping all tasks have one shot sample. Figure 3b shows increasing $M$ with fixed $m$ improves accuracy, and Figure 3c shows increasing $m$ with fixed $M$ has similar behavior. Furthermore, different configurations of $M$ and $m$ for the same total sample size $Mm$ have similar performance (e.g., $M = 400, m = 450$ compared to $M = 600, m = 300$ in Figure 3b). These again align with our theorem.

## 4.2 TASK SELECTION

**Setup.** To evaluate our task selection Algorithm 1, we use the Meta-Dataset (Triantafillou et al., 2020). It contains 10 extensive public image datasets from various domains, each partitioned into train/val/test splits. For each dataset except Describable Textures due to small size, we conduct an experiment, where the test-split of that dataset is used as the target task while the train-split from all the other datasets are used as candidate finetuning tasks. Each experiment follows the experiment protocol in Section 4. We performed ablation studies on the task selection algorithm, concentrating on either consistency or diversity, while violating the other. See details in Appendix F.4.

**Results.** Table 1 compares the results from finetuning with tasks selected by our algorithm to those from finetuning with tasks selected by other methods. Our algorithm consistently attains performance gains. For instance, on Omniglot, our algorithm leads to significant accuracy gains over random selection of 8.9%, 19.7%, and 2.4% with CLIP, DINO v2, and MoCo v3, respectively.

| pretrained | backbone | method | miniImageNet | | tieredImageNet | | DomainNet | |
|---|---|---|---|---|---|---|---|---|
| | | | 1-shot | 5-shot | 1-shot | 5-shot | 1-shot | 5-shot |
| MoCo v3 | ViT-B | Adaptation | 75.33 (0.30) | 92.78 (0.10) | 62.17 (0.36) | 83.42 (0.23) | 24.84 (0.25) | 44.32 (0.29) |
| | | Standard FT | 75.38 (0.30) | 92.80 (0.10) | 62.28 (0.36) | 83.49 (0.23) | 25.10 (0.25) | 44.76 (0.27) |
| | | Ours | **80.62** (0.26) | **93.89** (0.09) | **68.32** (0.35) | **85.49** (0.22) | **32.88** (0.29) | **54.17** (0.30) |
| | ResNet50 | Adaptation | 68.80 (0.30) | 88.23 (0.13) | 55.15 (0.34) | 76.00 (0.26) | 27.34 (0.27) | 47.50 (0.28) |
| | | Standard FT | 68.85 (0.30) | 88.23 (0.13) | 55.23 (0.34) | 76.07 (0.26) | 27.43 (0.27) | 47.65 (0.28) |
| | | Ours | **71.16** (0.29) | **89.31** (0.12) | **58.51** (0.35) | **78.41** (0.25) | **33.53** (0.30) | **55.82** (0.29) |
| DINO v2 | ViT-S | Adaptation | 85.90 (0.22) | 95.58 (0.08) | 74.54 (0.32) | 89.20 (0.19) | 52.28 (0.39) | 72.98 (0.28) |
| | | Standard FT | 86.75 (0.22) | 95.76 (0.08) | 74.84 (0.32) | 89.30 (0.19) | 54.48 (0.39) | 74.50 (0.28) |
| | | Ours | **88.70** (0.22) | **96.08** (0.08) | **77.78** (0.32) | **90.23** (0.18) | **61.57** (0.40) | **77.97** (0.27) |
| | ViT-B | Adaptation | 90.61 (0.19) | 97.20 (0.06) | 82.33 (0.30) | 92.90 (0.16) | 61.65 (0.41) | 79.34 (0.25) |
| | | Standard FT | 91.07 (0.19) | 97.32 (0.06) | 82.40 (0.30) | 93.07 (0.16) | 61.84 (0.39) | 79.63 (0.25) |
| | | Ours | **92.77** (0.18) | **97.68** (0.06) | **84.74** (0.30) | **93.65** (0.16) | **68.22** (0.40) | **82.62** (0.24) |
| Supervised pretraining on ImageNet | ViT-B | Adaptation | 94.06 (0.15) | 97.88 (0.05) | 83.82 (0.29) | 93.65 (0.13) | 28.70 (0.29) | 49.70 (0.28) |
| | | Standard FT | 95.28 (0.13) | 98.33 (0.04) | 86.44 (0.27) | 94.91 (0.12) | 30.93 (0.31) | 52.14 (0.29) |
| | | Ours | **96.91** (0.11) | **98.76** (0.04) | **89.97** (0.25) | **95.84** (0.11) | **48.02** (0.38) | **67.25** (0.29) |
| | ResNet50 | Adaptation | 81.74 (0.24) | 94.08 (0.09) | 65.98 (0.34) | 84.14 (0.21) | 27.32 (0.27) | 46.67 (0.28) |
| | | Standard FT | 84.10 (0.22) | 94.81 (0.09) | 74.48 (0.33) | 88.35 (0.19) | 34.10 (0.31) | 55.08 (0.29) |
| | | Ours | **87.61** (0.20) | **95.92** (0.07) | **77.74** (0.32) | **89.77** (0.17) | **39.09** (0.34) | **60.60** (0.29) |

Table 2: **Results of few-shot image classification.** We report average classification accuracy (%) with 95% confidence intervals on test splits. Adaptation: Direction adaptation without finetuning; Standard FT: Standard finetuning; Ours: Our multitask finetuning; 1-/5-shot: number of labeled images per class in the target task.

Violating consistency or diversity conditions generally result in a reduced performance compared to our approach. These results are well aligned with our expectations and affirm our diversity and consistency conclusions. We provide more ablatioin study on task selection in Table 9 in Appendix F.4. We also apply task selection algorithm on DomainNet in Appendix F.5. Furthermore, in Appendix G, we employ our algorithm for NLP models on the GLUE dataset.

## 4.3 Effectiveness of Multitask Finetuning

**Setup.** We also conduct more extensive experiments on large-scale datasets across various settings to confirm the effectiveness of multitask finetuning. We compare to two baselines: *direct adaptation* where we directly adapt pretrained model encoder on target tasks without any finetuning, and *standard finetuning* where we append encoder with a linear head to map representations to class logits and finetune the whole model. During testing, we removed the linear layer and used the same few-shot testing process with the finetuned encoders. Please refer Table 14 in Appendix F.8 for full results.

**Results.** Table 2 presents the results for various pretraining and finetuning methods, backbones, datasets, and few-shot learning settings. Multitask finetuning consistently outperforms the baselines in different settings. For example, in the most challenging setting of 1-shot on DomainNet, it attains a major gain of 7.1% and 9.3% in accuracy over standard finetuning and direct adaptation, respectively, when considering self-supervised pretraining with DINO v2 and using a Transformer model (ViT-S). Interestingly, multitask finetuning achieves more significant gains for models pretrained with supervised learning than those pretrained with contrastive learning. For example, on DomainNet, multitask finetuning on supervised pretrained ViT-B achieves a relative gain of 67% and 35% for 1- and 5-shot, respectively. In contrast, multitask finetuning on DINO v2 pretrained ViT-B only shows a relative gain of 10% and 4%. This suggests that models from supervised pretraining might face a larger domain gap than models from DINO v2, and multitask finetuning helps to bridge this gap.

## 5 Conclusions

In this work, we studied the theoretical justification of multitask finetuning for adapting pretrained foundation models to downstream tasks with limited labels. Our analysis shows that, given sufficient sample complexity, finetuning using a diverse set of pertinent tasks can improve the performance on the target task. This claim was examined in our theoretical framework and substantiated by the empirical evidence accumulated throughout our study. Built on this theoretical insight, we further proposed a practical algorithm for selecting tasks for multitask finetuning, leading to significantly improved results when compared to using all possible tasks. We anticipate that our research will shed light on the adaptation of foundation models, and stimulate further exploration in this direction.

ETHICS STATEMENT

Our work aims to study the theoretical justification of this multitask finetuning approach. Our paper is purely theoretical and empirical in nature and thus we foresee no immediate negative ethical impact. We quantify the relationship between multitasks and target task by diversity and consistency metrics and propose a practical task selection algorithm, which may have a positive impact on the machine learning community. We hope our work will inspire effective algorithm design and promote a better understanding of effective adaptation of foundation models.

REPRODUCIBILITY STATEMENT

For theoretical results in the Section 3, a complete proof is provided in the Appendix C. The theoretical results and proofs for a multiclass setting that is more general than that in the main text are provided in the Appendix D. The complete proof for linear case study on diversity and consistency is provided in the Appendix E. For experiments in the Section 4, complete details and experimental results are provided in the Appendices F to H. The source code with explanations and comments is provided in https://github.com/OliverXUZY/Foudation-Model_Multitask.

ACKNOWLEDGMENTS

The work is partially supported by Air Force Grant FA9550-18-1-0166, the National Science Foundation (NSF) Grants 2008559-IIS, CCF-2046710, and 2023239-DMS. The work also received partial support from grants by McPherson Eye Research Institute and VCGRE at UW Madison, and from the Army Research Lab under contract number W911NF-2020221.

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

CONTENTS

# Appendix

In this appendix, we first state our limitation in Appendix A. Then, we provide more related work in Appendix B. The proof of our theoretical results for the binary case is presented in Appendix C, where we formalize the theoretical settings and assumptions and elaborate on the results to contrastive pretraining in Appendix C.1 and supervised pretraining in Appendix C.2. We prove the main theory in Appendix C.4, which is a direct derivative of C.1 and C.2. We generalize the setting to multiclass and provide proof in Appendix D. We include the full proof of the general linear case study in Appendix E. We provide additional experimental results of vision tasks in Appendix F, language tasks in Appendix G, and vision-language tasks in Appendix H.

## A  LIMITATION

We recognize an interesting phenomenon within multitask finetuning and dig into deeper exploration with theoretical analysis, while our experimental results may or may not beat state-of-the-art (SOTA) performance, as our focus is not on presenting multitask finetuning as a novel approach nor on achieving SOTA performance. On the other hand, the estimation of our diversity and consistency parameters accurately on real-world datasets is valuable but time-consuming. Whether there exists an efficient algorithm to estimate these parameters is unknown. We leave this challenging problem as our future work.

## B  MORE RELATED WORK

**Training Foundation Models.**    Foundation models (Bommasani et al., 2021) are typically trained using self-supervised learning over broad data. The most commonly used training approaches include *contrastive learning* in vision and *masked modeling* in NLP. Our theoretical analysis considers both approaches under a unified framework. Here we briefly review these approaches.

*Contrastive learning*, in a self-supervised setting, aims to group randomly augmented versions of the same data point while distinguishing samples from diverse groups. The success of this approach in vision and multi-modal training tasks (Oord et al., 2018; Chen et al., 2020; He et al., 2020; Tian et al., 2020a; Grill et al., 2020; Radford et al., 2021) has spurred considerable interest. Several recent studies (Arora et al., 2019; HaoChen et al., 2021; Tosh et al., 2021; Zimmermann et al., 2021; Wei et al., 2021; Wang & Isola, 2020; Wen & Li, 2021; Wang et al., 2022; Shi et al., 2023a; Huang et al., 2023; Sun et al., 2023b;a) seek to develop its theoretical understanding. Arora et al. (2019) established theoretical guarantees on downstream classification performance. HaoChen et al. (2021) provided analysis on spectral contrastive loss. Their analysis assumes the pretraining and target tasks share the same data distribution and focus on the effect of contrastive learning on direct adaptation. Our work focuses on the novel class setting and investigates further finetuning the pretrained model with multitask to improve performance.

*Masked modeling* seeks to predict masked tokens in an input sequence. This self-supervised approach is the foundation of many large language models (Devlin et al., 2019; Liu et al., 2019; Chowdhery et al., 2022; Ni et al., 2022; Touvron et al., 2023), and has been recently explored in vision (He et al., 2022). In the theoretical frontier, Zhao et al. (2023) formulated masked language modeling as standard supervised learning with labels from the input text. They further investigated the relationship between pretrained data and testing data by diversity statement. Our work subsumes their work as a special case, and can explain a broader family of pretraining methods.

**Adapting Foundation Models.**    Adapting foundation models to downstream tasks has recently received significant attention. The conventional wisdom, mostly adopted in vision (Vinyals et al., 2016; Ge & Yu, 2017; Chen et al., 2020; He et al., 2020; 2022; Shi et al., 2023b), involves learning a simple function, such as linear probing, on the representation from a foundation model, while keeping the model frozen or minorly finetuning the whole model. In NLP, prompt-based finetuning (Gao et al., 2021a; Hu et al., 2022a; Chung et al., 2022; Song et al., 2022; Zhou et al., 2022b; Xie et al., 2023; Zhang et al., 2023) was developed and widely used, in which a prediction task is transformed into a masked language modeling problem during finetuning. With the advances in large language models, parameter-efficient tuning has emerged as an attractive solution. Prompt tuning (Lester et al.,

2021; Li & Liang, 2021; Roberts et al., 2023) learns an extra prompt token for a new task, while updating minimal or no parameters in the model backbone. Another promising approach is in-context learning (Min et al., 2022b; Wei et al., 2022a;b; Shi et al., 2023d; Xu et al., 2024), where the model is tasked to make predictions based on contexts supplemented with a few examples, with no parameter updates. In this paper, we consider adapting foundation models to new tasks with limited labels. Parameter-efficient tuning, such as in-context learning, might face major challenges (Xie et al., 2022) when the distribution of the new task deviates from those considered in pretraining. Instead, our approach finetunes the model using multiple relevant tasks. We empirically verify that doing so leads to better adaptation.

**Multitask Learning.**    Multitask supervised learning has been considered for transfer learning to a target task (Zhong et al., 2021; Sanh et al., 2022; Chen et al., 2022; Min et al., 2022a; Wang et al., 2023). Multitask has been shown to induce zero-shot generalization in large language models (Sanh et al., 2022), and also enable parameter efficient tuning by prompt tuning (Wang et al., 2023). Our work leverages multitask learning to unlock better zero-shot and few-shot performance of pretrained models. Min et al. (2022a); Chen et al. (2022) primarily focus on in-context learning, Zhong et al. (2021) focuses on the idea of task conversion where transfer classification task as question-answer format, our approach is based on utilizing original examples, in alignment with our theoretical framework. A line of theoretical work provides the error bound of the target task in terms of sample complexity (Du et al., 2021; Tripuraneni et al., 2021; Shi et al., 2023a; Xu et al., 2023). Tripuraneni et al. (2020) established a framework of multitask learning centered around the notion of task diversity for the training data. Their work mainly analyzed representations from supervised pretraining using multitasks. In contrast, our work considers representations from self-supervised pretraining, and focuses on multitask finetuning. Our approach and analysis guarantee that limited but diverse and consistent finetuning task can improve the prediction performance on a target task with novel classes.

**Few-shot Learning and Meta Learning.**    Few-shot learning necessitates the generalization to new tasks with only a few labeled samples (Wang et al., 2020; Vu et al., 2021; Murty et al., 2021; Liu et al., 2021; Yang et al., 2022; Galanti et al., 2022). Direct training with limited data is prone to overfitting. Meta learning offers a promising solution that allows the model to adapt to the few-shot setting (Finn et al., 2017; Raghu et al., 2020). This solution has been previously developed for vision tasks (Vinyals et al., 2016; Snell et al., 2017; Chen et al., 2021b; Hu et al., 2022b). Inspired by meta learning in the few-shot setting, our analysis extends the idea of multitask finetuning by providing sound theoretic justifications and demonstrating strong empirical results. We further introduce a task selection algorithm that bridges our theoretical findings with practical applications in multitask finetuning.

## C    DEFERRED PROOFS

In this section, we provide a formal setting and proof. We first formalize our setting in multiclass. Consider our task $\mathcal{T}$ contains $r$ classes where $r \geq 2$.

**Contrastive Learning.**    In contrastive learning, we sampled one example $x$ from any latent class $y$, then apply the data augmentation module that randomly transforms such sample into another view of the original example denoted $x^+$. We also sample other $r-1$ examples $\{x_k^-\}_{k=1}^r$ from other latent classes $\{y_k^-\}_{k=1}^{r-1}$. We treat $(x, x^+)$ as a positive pair and $(x, x_k^-)$ as negative pairs. We define $\mathcal{D}_{\text{con}}(\eta)$ over sample $(x, x^+, x_1^-, \ldots, x_{r-1}^-)$ by following sampling procedure

$$(y, y_1^-, \ldots, y_{r-1}^-) \sim \eta^r \tag{5}$$

$$x \sim \mathcal{D}(y), \ x^+ \sim \mathcal{D}(y), \ x_k^- \sim \mathcal{D}(y_k^-), k = 1, \ldots, r-1. \tag{6}$$

We consider general contrastive loss $\ell_u\left(\left\{\phi(x)^\top\left(\phi(x^+) - \phi(x_k^-)\right)\right\}_{k=1}^{r-1}\right)$, where loss function $\ell_u$ is non-negative decreasing function. Minimizing the loss is equivalent to maximizing the similarity between positive pairs while minimizing it between negative pairs. In particular, logistic loss $\ell_u(\boldsymbol{v}) = \log\left(1 + \sum_i \exp\left(-\boldsymbol{v}_i\right)\right)$ for $\boldsymbol{v} \in \mathbb{R}^{r-1}$ recovers the one used in most empirical works: $-\log\left(\frac{\exp\{\phi(x)^\top \phi(x^+)\}}{\exp\{\phi(x)^\top \phi(x^+)\} + \sum_{i=1}^{r-1} \exp\{\phi(x)^\top \phi(x_i^-)\}}\right)$. The popula-

tion contrastive loss is defined as $\mathcal{L}_{con-pre}(\phi) := \mathbb{E}\left[\ell_u\left(\left\{\phi(x)^\top\left(\phi(x^+) - \phi(x_k^-)\right)\right\}_{k=1}^{r-1}\right)\right]$.
Let $\mathcal{S}_{con-pre} := \left\{x_j, x_j^+, x_{j1}^-, \ldots, x_{j(r-1)}^-\right\}_{j=1}^N$ denote our contrastive training set with $N$ samples, sampled from $\mathcal{D}_{con}(\eta)$, we have empirical contrastive loss $\widehat{\mathcal{L}}_{con-pre}(\phi) :=$
$\frac{1}{N}\sum_{i=1}^N\left[\ell_u\left(\left\{\phi(x)^\top\left(\phi(x^+) - \phi(x_k^-)\right)\right\}_{k=1}^{r-1}\right)\right]$.

**Supervised Learning.** In supervised learning we have a labeled dataset denoted as $\mathcal{S}_{con-pre} :=$ $\{x_j, y_j\}_{j=1}^N$ with $N$ samples, by following sampling procedure:

$$y \sim \eta \tag{7}$$
$$x \sim \mathcal{D}(y). \tag{8}$$

There are in total $K$ classes, denote $\mathcal{C}$ as the set consists of all classes. On top of the representation function $\phi$, there is a linear function $f \in \mathcal{F} \subset \{\mathbb{R}^d \to \mathbb{R}^K\}$ predicting the labels, denoted as $g(x) = f \circ \phi(x)$. We consider general supervised loss on data point $(x, y)$ is

$$\ell(g(x), y) := \ell_u\left((g(x))_y - (g(x))_{y' \neq y, y' \in \mathcal{C}}\right). \tag{9}$$

where loss function $\ell_u$ is non-negative decreasing function. In particular, logistic loss $\ell_u(\boldsymbol{v}) = \log\left(1 + \sum_i \exp(-\boldsymbol{v}_i)\right)$ for $\boldsymbol{v} \in \mathbb{R}^{K-1}$ recovers the one used in most empirical works:

$$\ell(g(x), y) = \ell_u\left((g(x))_y - (g(x))_{y' \neq y, y' \in \mathcal{C}}\right) \tag{10}$$

$$= \log\left\{1 + \sum_{k \neq y}^K \exp\left(-\left[(g(x))_y - (g(x))_k\right]\right)\right\} \tag{11}$$

$$= -\log\left\{\frac{\exp(g(x))_y}{\sum_{k=1}^K \exp(g(x))_k}\right\}. \tag{12}$$

The population supervised loss is

$$\mathcal{L}_{sup-pre}(\phi) = \min_{f \in \mathcal{F}} \mathbb{E}_{x,y}\left[\ell\left(f \circ \phi(x), y\right)\right]. \tag{13}$$

For training set $\mathcal{S}_{sup-pre} := \{x_i, y_i\}_{i=1}^N$ with $N$ samples, the empirical supervised pretraining loss is $\widehat{\mathcal{L}}_{sup-pre}(\phi) := \min_{f \in \mathcal{F}} \frac{1}{N}\sum_{i=1}^N\left[\ell\left(f \circ \phi(x_i), y_i\right)\right]$.

**Masked Language Modeling.** Masked language modeling is a self-supervised learning method. It can be viewed as a specific form of supervised pretraining above. The pretraining data is a substantial dataset of sentences, often sourced from Wikipedia. In the pretraining phase, a random selection of words is masked within each sentence, and the training objective is to predict these masked words using the context provided by the remaining words in the sentence. This particular pretraining task can be viewed as a multi-class classification problem, where the number of classes (denoted as $K$) corresponds to the size of the vocabulary. Considering BERT and its variations, we have function $\phi$ as a text encoder. This encoder outputs a learned representation, often known as `[CLS]` token. The size of such learned representation is $d$, which is 768 for BERT$_{\text{BASE}}$ or 1024 for BERT$_{\text{LARGE}}$.

**Supervised Tasks.** Given a representation function $\phi$, we apply a task-specific linear transformation $W$ to the representation to obtain the final prediction. Consider $r$-way supervised task $\mathcal{T}$ consist a set of distinct classes $(y_1, \ldots, y_r) \subseteq \mathcal{C}$. We define $\mathcal{D}_{\mathcal{T}}(y)$ as the distribution of randomly drawing $y \in (y_1, \ldots, y_r)$, we denote this process as $y \sim \mathcal{T}$. Let $\mathcal{S}_{\mathcal{T}} := \{x_j, y_j\}_{j=1}^m$ denote our labeled training set with $m$ samples, sampled i.i.d. from $y_j \sim \mathcal{T}$ and $x_j \sim \mathcal{D}(y_j)$. Define $g(\phi(\mathbf{x})) := W\phi(x) \in \mathbb{R}^r$ as prediction logits, where $W \in \mathbb{R}^{r \times d}$. The typical supervised logistic loss is $\ell(g \circ \phi(x), y) := \ell_u\left(\{g(\phi(\mathbf{x}))_y - g(\phi(\mathbf{x}))_{y'}\}_{y' \neq y}\right)$. Similar to Arora et al. (2019), define supervised loss w.r.t the task $\mathcal{T}$

$$\mathcal{L}_{sup}(\mathcal{T}, \phi) := \min_{W \in \mathbb{R}^{r \times d}} \mathbb{E}_{y \sim \mathcal{T}} \mathbb{E}_{x \sim \mathcal{D}(y)}\left[\ell\left(W \cdot \phi(x), y\right)\right]. \tag{14}$$

---

**Algorithm 2** Multitask Finetuning

---

**Input:** multitasks $\mathcal{T}_1, \ldots, \mathcal{T}_M$, pretrained model $\hat{\phi}$ with parameter $\theta$, step size $\gamma$
 1: Initialize $\phi$ with $\hat{\phi}$
 2: **repeat**
 3:     **for all** $\mathcal{T}_i$ **do**
 4:         $\theta \leftarrow \theta - \gamma \nabla_\theta \widehat{\mathcal{L}}_{sup}(\mathcal{T}_i, \phi)$                 $\{\ \widehat{\mathcal{L}}_{sup}(\mathcal{T}_i, \phi)$ is defined in (2)$\}$
 5:     **end for**
 6: **until** converge
**Output:** The final model, denoted as $\phi'$

---

Define supervised loss with mean classifier as $\mathcal{L}^\mu_{sup}(\mathcal{T}, \phi) := \underset{y \sim \mathcal{T}}{\mathbb{E}} \underset{x \sim \mathcal{D}(y)}{\mathbb{E}} [\ell(W^\mu \cdot \phi(x), y)]$ where each row of $W^\mu$ is the mean of each class in $\mathcal{T}$, $W^\mu_{y_k} := \mu_{y_k} = \underset{x \sim y_k}{\mathbb{E}} (\phi(x)), k = 1, \ldots, r$. In the target task, suppose we have $r$ distinct classes from $\mathcal{C}$ with equal weights. Consider $\mathcal{T}$ follows a general distribution $\zeta$. Define expected supervised loss as $\mathcal{L}_{sup}(\phi) := \underset{\mathcal{T} \sim \zeta}{\mathbb{E}} [\mathcal{L}_{sup}(\mathcal{T}, \phi)]$.

**Mutlitask Finetuning.** Suppose we have $M$ auxiliary tasks $\{\mathcal{T}_1, \mathcal{T}_2, \ldots, \mathcal{T}_M\}$, each with $m$ labeled samples $\mathcal{S}_i := \{(x^i_j, y^i_j) : j \in [m]\}$. The finetuning data are $\mathcal{S} := \cup_{i \in [M]} \mathcal{S}_i$. Given a pretrained model $\hat{\phi}$, we further finetune it using the objective:

$$\min_{\phi \in \Phi} \frac{1}{M} \sum_{i=1}^{M} \widehat{\mathcal{L}}_{sup}(\mathcal{T}_i, \phi), \quad \text{where} \quad \widehat{\mathcal{L}}_{sup}(\mathcal{T}_i, \phi) := \min_{\boldsymbol{w}_i \in \mathbb{R}^d} \frac{1}{m} \sum_{j=1}^{m} \ell(\boldsymbol{w}_i^\top \phi(x^i_j), y^i_j). \tag{15}$$

This can be done via gradient descent from the initialization $\hat{\phi}$ (see Algorithm 2).

Algorithm 2 has similar pipeline as Raghu et al. (2020) where in the inner loop only a linear layer on top of the embeddings is learned. However, our algorithm is centered on multitask finetuning, where no inner loop is executed.

Finally, we formalize our assumption Assumption 1 below.

**Assumption 2** (Regularity Conditions). *The following regularity conditions hold:*

*(A1) Representation function $\phi$ satisfies $\|\phi\|_2 \leq R$.*

*(A2) Linear operator $W$ satisfies bounded spectral norm $\|W\|_2 \leq B$.*

*(A3) The loss function $\ell_u$ are bounded by $[0, C]$ and $\ell(\cdot)$ is L-Lipschitz.*

*(A4) The supervised loss $\mathcal{L}_{sup}(\mathcal{T}, \phi)$ is $\tilde{L}$-Lipschitz with respect to $\phi$ for $\forall \mathcal{T}$.*

## C.1   Contrastive Pretraining

In this section, we will show how multitask finetuning improves the model from contrastive pretraining. We present pretraining error in binary classification and $\mathcal{D}_\mathcal{T}(y)$ as uniform. See the result for the general condition with multi-class in Appendix D.

### C.1.1   Contrastive Pretraining and Direct Adaptation

In this section, we show the error bound of a foundation model on a target task, where the model is pretrained by contrastive loss followed directly by adaptation.

We first show how pretraining guarantees the expected supervised loss:

$$\mathcal{L}_{sup}(\phi) = \underset{\mathcal{T} \sim \zeta}{\mathbb{E}} [\mathcal{L}_{sup}(\mathcal{T}, \phi)]. \tag{16}$$

The error on the target task can be bounded by $\mathcal{L}_{sup}(\phi)$. We use $\epsilon^*$ denote $\mathcal{L}_{sup}(\phi^*_\zeta)$.

**Lemma C.1** (Lemma 4.3 in Arora et al. (2019)). *For $\forall \phi \in \Phi$ pretrained in contrastive loss, we have $\mathcal{L}_{sup}(\phi) \leq \frac{1}{1-\tau}(\mathcal{L}_{con-pre}(\phi) - \tau)$.*

We state the theorem below.

**Theorem C.2.** *Assume Assumption 1 and that $\Phi$ has $\nu$-diversity and $\kappa$-consistency with respect to $\phi^*, \phi_\zeta^*$. Suppose $\hat{\phi}$ satisfies $\hat{\mathcal{L}}_{con-pre}(\hat{\phi}) \leq \epsilon_0$. Let $\tau := \Pr_{(y_1, y_2) \sim \eta^2} \{y_1 = y_2\}$. Consider pretraining set $\mathcal{S}_{con-pre} = \left\{ x_j, x_j^+, x_j^- \right\}_{j=1}^N$. For any $\delta \geq 0$, if*

$$N \geq \frac{1}{\epsilon_0} \left[ 8 L R \mathcal{R}_N(\Phi) + \frac{8C^2}{\epsilon_0} \log(\frac{2}{\delta}) \right].$$

*Then with probability $1 - \delta$, for any target task $\mathcal{T}_0 \subset \mathcal{C}_0$,*

$$\mathcal{L}_{sup}(\mathcal{T}_0, \hat{\phi}) - \mathcal{L}_{sup}(\mathcal{T}_0, \phi^*) \leq \frac{1}{\nu} \left[ \frac{1}{1-\tau} (2\epsilon_0 - \tau) - \mathcal{L}_{sup}(\phi^*) \right] + \kappa. \tag{17}$$

The pretraining sample complexity is $O(\frac{\mathcal{R}_N(\Phi)}{\epsilon_0} + \frac{\log(1/\delta)}{\epsilon_0^2})$. The first term is the Rademacher complexity of the entire representation space $\Phi$ with sample size $N$. The second term relates to the generalization bound. Pretraining typically involves a vast and varied dataset, sample complexity is usually not a significant concern during this stage.

*Proof of Theorem C.2.* Recall in binary classes, $\mathcal{S}_{con-pre} = \left\{ x_j, x_j^+, x_j^- \right\}_{j=1}^N$ denote our contrastive training set, sampled from $\mathcal{D}_{con}(\eta)$. Then by Lemma A.2 in Arora et al. (2019), with **(A1)** and **(A3)**, we have for $\forall \phi \in \Phi$ with probability $1 - \delta$,

$$\mathcal{L}_{con-pre}(\phi) - \hat{\mathcal{L}}_{con-pre}(\phi) \leq \frac{4 L R \mathcal{R}_N(\Phi)}{N} + C \sqrt{\frac{\log \frac{1}{\delta}}{N}}. \tag{18}$$

To have above $\leq \epsilon_0$, we have sample complexity

$$N \geq \frac{1}{\epsilon_0} \left[ 8 L R \mathcal{R}_N(\Phi) + \frac{8C^2}{\epsilon_0} \log(\frac{2}{\delta}) \right].$$

In pretraining, we have $\hat{\phi}$ such that

$$\hat{\mathcal{L}}_{con-pre}(\hat{\phi}) \leq \epsilon_0.$$

Then with the above sample complexity, we have pretraining $\hat{\phi}$

$$\mathcal{L}_{con-pre}(\hat{\phi}) \leq 2\epsilon_0.$$

Recall $\nu$-diversity and $\kappa$-consistency, for target task $\mathcal{T}_0$, we have that for $\hat{\phi}$ and $\phi^*$,

$$\mathcal{L}_{sup}(\mathcal{T}_0, \hat{\phi}) - \mathcal{L}_{sup}(\mathcal{T}_0, \phi^*) = \mathcal{L}_{sup}(\mathcal{T}_0, \hat{\phi}) - \mathcal{L}_{sup}(\mathcal{T}_0, \phi_\zeta^*) + \mathcal{L}_{sup}(\mathcal{T}_0, \phi_\zeta^*) - \mathcal{L}_{sup}(\mathcal{T}_0, \phi^*) \tag{19}$$

$$\leq d_{\mathcal{C}_0}(\hat{\phi}, \phi_\zeta^*) + \mathcal{L}_{sup}(\mathcal{T}_0, \phi_\zeta^*) - \mathcal{L}_{sup}(\mathcal{T}_0, \phi^*) \tag{20}$$

$$\leq \bar{d}_\zeta(\hat{\phi}, \phi_\zeta^*)/\nu + \kappa \tag{21}$$

$$\leq \frac{1}{\nu} \left[ \mathcal{L}_{sup}(\hat{\phi}) - \mathcal{L}_{sup}(\phi_\zeta^*) \right] + \kappa \tag{22}$$

$$= \frac{1}{\nu} \left[ \frac{1}{1-\tau} (\mathcal{L}_{con-pre}(\hat{\phi}) - \tau) - \epsilon^* \right] + \kappa \tag{23}$$

$$\leq \frac{1}{\nu} \left[ \frac{1}{1-\tau} (2\epsilon_0 - \tau) - \epsilon^* \right] + \kappa, \tag{24}$$

where the second to last inequality comes from Lemma C.1. $\square$

### C.1.2 CONTRASTIVE PRETRAINING AND MULTITASK FINETUNING

In this section, we show the error bound of a foundation model on a target task can be further reduced by multitask finetuning. We achieve this by showing that expected supervised loss $\mathcal{L}_{sup}(\phi)$ can be further reduced after multitask finetuning. The error on the target task can be bounded by $\mathcal{L}_{sup}(\phi)$. We use $\epsilon^*$ denote $\mathcal{L}_{sup}(\phi_\zeta^*)$.

Following the intuition in Garg & Liang (2020), we first re-state the definition of representation space.

**Definition 3.** *The subset of representation space is*

$$\Phi(\tilde{\epsilon}) = \left\{ \phi \in \Phi : \hat{\mathcal{L}}_{pre}(\phi) \le \tilde{\epsilon} \right\}.$$

Recall $\mathcal{S} = \{(x_j^i, y_j^i) : i \in [M], j \in [m]\}$ as finetuning dataset.

We define two function classes and associated Rademacher complexity.

**Definition 4.** *Consider function class*

$$\mathcal{G}_\ell(\tilde{\epsilon}) = \left\{ g_{W,\phi}(x,y) : g_{W,\phi}(x,y) = \ell(W\phi(x_j^i), y_j^i), \phi \in \Phi(\tilde{\epsilon}), \|W\|_2 \le B \right\}.$$

*We define Rademacher complexity as*

$$\mathcal{R}_n(\mathcal{G}_\ell(\tilde{\epsilon})) = \underset{\{\sigma_i\}_{j=1}^n, \{x_j, y_j\}_{j=1}^n}{\mathbb{E}} \left[ \sup_{\ell \in \mathcal{G}_\ell(\tilde{\epsilon})} \sum_{j=1}^n \sigma_j \ell(W \cdot \phi(x_j), y_j) \right].$$

**Definition 5.** *Consider function class*

$$\mathcal{G}(\tilde{\epsilon}) = \{ g_\phi : g_\phi(\mathcal{T}) = \mathcal{L}_{sup}(T, \phi), \phi \in \Phi(\tilde{\epsilon}) \}.$$

*We define Rademacher complexity as*

$$\mathcal{R}_M(\mathcal{G}(\tilde{\epsilon})) = \underset{\{\sigma_i\}_{i=1}^M, \{\mathcal{T}_i\}_{i=1}^M}{\mathbb{E}} \left[ \sup_{\phi \in \Phi(\tilde{\epsilon})} \sum_{i=1}^M \sigma_i \mathcal{L}_{sup}(\mathcal{T}_i, \phi) \right].$$

The key idea is multitask finetuning further reduce the expected supervised loss of a pretrained foundation model $\phi$:

$$\mathcal{L}_{sup}(\phi) = \underset{\mathcal{T} \sim \zeta}{\mathbb{E}} \left[ \mathcal{L}_{sup}(\mathcal{T}, \phi) \right]. \tag{25}$$

We first introduce some key lemmas. These lemmas apply to general $r$ classes in a task $\mathcal{T}$.

**Lemma C.3** (Bounded Rademacher complexity). *By (A2) and (A3), we have for $\forall n$*

$$\mathcal{R}_n(\mathcal{G}_\ell(\tilde{\epsilon})) \le 4\sqrt{r-1}LB\mathcal{R}_n(\Phi(\tilde{\epsilon})).$$

*Proof of Lemma C.3.* We first prove $\ell(g(\phi(x)), y)$ is $\sqrt{2(r-1)}LB$-Lipschitz with respect to $\phi$ for all $\forall y \in \mathcal{C}$. Consider

$$f_y(g(\phi(\mathbf{x}))) = \{g(\phi(\mathbf{x}))_y - g(\phi(\mathbf{x}))_{y'}\}_{y' \neq y},$$

where $f_y : \mathbb{R}^r \to \mathbb{R}^{r-1}$. Note that

$$\ell(g \circ \phi(x), y) = \ell \left( \{g(\phi(\mathbf{x}))_y - g(\phi(\mathbf{x}))_{y'}\}_{y' \neq y} \right)$$
$$= \ell(f_y(g(\phi(\mathbf{x})))).$$

By (A3), we have $\ell$ is $L$-Lipschitz. We then prove $f_y$ is $\sqrt{2(r-1)}$-Lipschitz. Without loss generality, consider $y = r$. We have $f_y(y) = [y_r - y_i]_{i=1}^{r-1}$. We have $\frac{\partial f_j}{y_i} = -\mathbb{1}\{j = i\}, i = 1, \ldots, r-1$, $\frac{\partial f_j}{y_r} = 1$. The Jacobian $J$ satisfies $\|J\|_2 \le \|J\|_F = \sqrt{2(r-1)}$.

Since $g$ is $B$-Lipschitz by (A2):$\|W\|_2 \le B$. Then $\ell(g(\phi(x)), y)$ is $\sqrt{2(r-1)}LB$-Lipschitz with respect to $\phi$ for all $\forall y \in \mathcal{C}$. The conclusion follows Corollary 4 in Maurer (2016). $\qquad \square$

**Lemma C.4** (Bounded $\tilde{\epsilon}$). *After finite steps in Multitask finetuning in Algorithm 2, we solve Equation (2) with empirical loss lower than $\epsilon_1 = \frac{\alpha}{3}\frac{1}{1-\tau}(2\epsilon_0 - \tau)$ and obtain $\phi'$. Then there exists a bounded $\tilde{\epsilon}$ such that $\phi' \in \Phi(\tilde{\epsilon})$.*

*Proof of Lemma C.4.* Given finite number of steps and finite step size $\gamma$ in Algorithm 2, we have bounded $\|\phi' - \hat{\phi}\|$. Then with **(A2)** and **(A3)**, using Lemma C.3 we have $\ell(g(\phi(x)), y)$ is $\sqrt{2(r-1)}LB$-Lipschitz with respect to $\phi$ for all $\forall y$, using theorem A.2 in Arora et al. (2019) we have $l_u$ is $LC$-Lipschitz with respect to $\phi$, we have $\widehat{\mathcal{L}}_{pre}(\phi)$ is $M$-Lipschitz with respect to $\phi$ with bounded $M$. We have $\exists\ \epsilon$ such that $\widehat{\mathcal{L}}_{pre}(\phi') - \widehat{\mathcal{L}}_{pre}(\hat{\phi}) \leq \epsilon\|\phi' - \hat{\phi}\|$. We have $\widehat{\mathcal{L}}_{pre}(\phi') \leq \epsilon_0 + \epsilon\|\phi' - \hat{\phi}\|$. Take $\tilde{\epsilon} = \epsilon_0 + \epsilon\|\phi' - \hat{\phi}\|$ yields the result. $\qquad\square$

**Lemma C.5.** *Assume Assumption 1 and that $\Phi$ has $\nu$-diversity and $\kappa$-consistency with respect to $\phi^*, \phi^*_\zeta$. Suppose for some small constant $\alpha \in (0,1)$ and $\tilde{\epsilon}$, we solve Equation (2) with empirical loss lower than $\epsilon_1 = \frac{\alpha}{3}\frac{1}{1-\tau}(2\epsilon_0 - \tau)$ and obtain $\phi'$. For any $\delta > 0$, if*

$$M \geq \frac{1}{\epsilon_1}\left[4\sqrt{2}\tilde{L}\mathcal{R}_M(\Phi(\tilde{\epsilon})) + \frac{4C^2}{\epsilon_1}\log(\frac{2}{\delta})\right], Mm \geq \frac{1}{\epsilon_1}\left[8\sqrt{r-1}LB\mathcal{R}_{Mm}(\Phi(\tilde{\epsilon})) + \frac{4C^2}{\epsilon_1}\log(\frac{2}{\delta})\right],$$

*then expected supervised loss $\mathcal{L}_{sup}(\phi') \leq \alpha\frac{1}{1-\tau}(2\epsilon_0 - \tau)$, with probability $1 - \delta$.*

*Proof of Lemma C.5.* Recall $\mathcal{S} := \{(x^i_j, y^i_j) : i \in [M], j \in [m]\}$ as finetuning dataset. Consider in Equation (2) we have $\widehat{\mathbf{W}} := (\widehat{W}_1, \ldots, \widehat{W}_M)$ and $\phi'$ such that $\frac{1}{M}\sum_{i=1}^M \frac{1}{m}\sum_{j=1}^m \ell(\widehat{W}_i \cdot \phi'(x^i_j), y^i_j) \leq \epsilon_1 < \frac{\alpha}{3}\epsilon_0$.

We tried to bound

$$\mathcal{L}_{sup}(\phi') - \frac{1}{m}\sum_{j=1}^m \ell(\widehat{W}_i \cdot \phi'(x^i_j), y^i_j).$$

Recall that

$$\mathcal{L}_{sup}(\mathcal{T}_i, \phi) = \min_{W \in \mathbb{R}^{r \times d}} \mathbb{E}_{y \sim \mathcal{T}_i} \mathbb{E}_{x \sim \mathcal{D}(y)}\left[\ell(W \cdot \phi(x), y)\right].$$

For $\forall \phi \in \Phi(\tilde{\epsilon})$

$$\mathcal{L}_{sup}(\phi) = \mathbb{E}_{\mathcal{T} \sim \zeta}\left[\mathcal{L}_{sup}(\mathcal{T}, \phi)\right] = \mathbb{E}_{\mathcal{T} \sim \zeta}\left[\min_{W \in \mathbb{R}^{r \times d}} \mathbb{E}_{y \sim \mathcal{T}} \mathbb{E}_{x \sim \mathcal{D}(y)}\left[\ell(W \cdot \phi(\mathbf{x}), y)\right]\right].$$

We have for $\forall \phi \in \Phi(\tilde{\epsilon})$, by uniform convergence (see Mohri et al. (2018) Theorem 3.3), we have with probability $1 - \delta/2$

$$\mathbb{E}_{\mathcal{T} \sim \zeta}\left[\mathcal{L}_{sup}(\mathcal{T}, \phi)\right] - \frac{1}{M}\sum_{i=1}^M \mathcal{L}_{sup}(\mathcal{T}_i, \phi) \leq \frac{2\mathcal{R}_M(\mathcal{G}(\tilde{\epsilon}))}{M} + \sqrt{\frac{\log(2/\delta)}{M}} \qquad (26)$$

$$\leq \frac{2\sqrt{2}\tilde{L}\mathcal{R}_M(\Phi(\tilde{\epsilon}))}{M} + \sqrt{\frac{\log(2/\delta)}{M}}, \qquad (27)$$

where the last inequality comes from **(A4)** and Corollary 4 in Maurer (2016). To have above $\leq \epsilon_1/2$, we have sample complexity

$$M \geq \frac{1}{\epsilon_1}\left[4\sqrt{2}\tilde{L}\mathcal{R}_M(\Phi(\tilde{\epsilon})) + \frac{4C^2}{\epsilon_1}\log(\frac{2}{\delta})\right].$$

Then we consider generalization bound for $\forall \phi$ and $\mathbf{W} := (W_1, \ldots, W_M)$

$$\mathcal{L}_{sup}(\phi, \mathbf{W}) = \frac{1}{M}\sum_{i=1}^M \mathbb{E}_{y^i \sim \mathcal{T}_i} \mathbb{E}_{x^i \sim \mathcal{D}(y^i)} \ell(W_i \cdot \phi(x^i), y^i) \qquad (28)$$

$$\hat{\mathcal{L}}_{sup}(\phi, \mathbf{W}) = \frac{1}{M}\sum_{i=1}^M \frac{1}{m}\sum_{j=1}^m \ell(W_i \cdot \phi(x^i_j), y^i_j), \qquad (29)$$

where $\mathbf{W} = (W_1, \ldots, W_M)$.

By uniform convergence (see Mohri et al. (2018) Theorem 3.3), we have with probability $1 - \delta/2$,

$$\mathcal{L}_{sup}(\phi, \mathbf{W}) - \hat{\mathcal{L}}_{sup}(\phi, \mathbf{W}) \leq \frac{2\mathcal{R}_{Mm}(\mathcal{G}_\ell)}{Mm} + \sqrt{\frac{\log(2/\delta)}{Mm}} \leq \frac{8\sqrt{r-1}LB\mathcal{R}_{Mm}(\Phi(\tilde{\epsilon}))}{Mm} + C\sqrt{\frac{\log(2/\delta)}{Mm}},$$

where the last inequality comes from Lemma C.3. To have above $\leq \epsilon_1/2$, we have sample complexity

$$Mm \geq \frac{1}{\epsilon_1}\left[8\sqrt{r-1}LB\mathcal{R}_{Mm}(\Phi(\tilde{\epsilon})) + \frac{4C^2}{\epsilon_1}\log(\frac{2}{\delta})\right],$$

satisfying $\forall \phi \in \Phi(\tilde{\epsilon})$

$$\begin{aligned}
\frac{1}{M}\sum_{i=1}^{M}\mathcal{L}_{sup}(\mathcal{T}_i, \phi) &= \frac{1}{M}\sum_{i=1}^{M}\min_{W \in \mathbb{R}^{r \times d}} \mathop{\mathbb{E}}_{y \sim \mathcal{T}_i} \mathop{\mathbb{E}}_{x \sim \mathcal{D}(y)}[\ell(W \cdot \phi(x), y)] \\
&\leq \frac{1}{M}\sum_{i=1}^{M} \mathop{\mathbb{E}}_{y \sim \mathcal{T}_i} \mathop{\mathbb{E}}_{x \sim \mathcal{D}(y)}\left[\ell\left(\widehat{W}_i \cdot \phi(x), y\right)\right] \\
&= \mathcal{L}_{sup}(\phi, \widehat{\mathbf{W}}) \\
&\leq \hat{\mathcal{L}}_{sup}(\phi, \widehat{\mathbf{W}}) + \epsilon_1/2.
\end{aligned}$$

Then combine above with Equation (26)

$$\begin{aligned}
\mathcal{L}_{sup}(\phi) &= \mathop{\mathbb{E}}_{\mathcal{T} \sim \zeta}[\mathcal{L}_{sup}(\mathcal{T}, \phi)] \\
&\leq \hat{\mathcal{L}}_{sup}(\phi, \widehat{\mathbf{W}}) + \epsilon_1.
\end{aligned}$$

We have

$$\mathcal{L}_{sup}(\phi') - \frac{1}{m}\sum_{j=1}^{m}\ell(\widehat{W}_i \cdot \phi'(x_j^i), y_j^i) \leq \epsilon_1$$

$$\mathcal{L}_{sup}(\phi') \leq 2\epsilon_1 \leq \alpha\frac{1}{1-\tau}(2\epsilon_0 - \tau).$$

The boundedness of $\tilde{\epsilon}$ follows Lemma C.4. $\qquad\square$

We state the theorem below.

**Theorem C.6.** *Assume Assumption 1 and that $\Phi$ has $\nu$-diversity and $\kappa$-consistency with respect to $\phi^*, \phi_\zeta^*$. Suppose for some small constant $\alpha \in (0, 1)$, we solve Equation (2) with empirical loss lower than $\epsilon_1 = \frac{\alpha}{3}\frac{1}{1-\tau}(2\epsilon_0 - \tau)$ and obtain $\phi'$. For any $\delta > 0$, if*

$$M \geq \frac{1}{\epsilon_1}\left[4\sqrt{2}\tilde{L}\mathcal{R}_M(\Phi(\tilde{\epsilon})) + \frac{4C^2}{\epsilon_1}\log(\frac{2}{\delta})\right], Mm \geq \frac{1}{\epsilon_1}\left[8LB\mathcal{R}_{Mm}(\Phi(\tilde{\epsilon})) + \frac{4C^2}{\epsilon_1}\log(\frac{2}{\delta})\right],$$

*then with probability $1 - \delta$, for any target task $\mathcal{T}_0 \subseteq \mathcal{C}_0$,*

$$\mathcal{L}_{sup}(\mathcal{T}_0, \phi') - \mathcal{L}_{sup}(\mathcal{T}_0, \phi^*) \leq \frac{1}{\nu}\left[\alpha\frac{1}{1-\tau}(2\epsilon_0 - \tau) - \mathcal{L}_{sup}(\phi_\zeta^*)\right] + \kappa. \tag{30}$$

*Proof of Theorem C.6.* Recall with $\nu$-diversity and $\kappa$-consistency with respect to $\phi^*, \phi_\zeta^*$, for target task $\mathcal{T}_0$, we have that for $\phi'$ and $\phi^*$,

$$\begin{aligned}
\mathcal{L}_{sup}(\mathcal{T}_0, \phi') - \mathcal{L}_{sup}(\mathcal{T}_0, \phi^*) &= \mathcal{L}_{sup}(\mathcal{T}_0, \phi') - \mathcal{L}_{sup}(\mathcal{T}_0, \phi_\zeta^*) + \mathcal{L}_{sup}(\mathcal{T}_0, \phi_\zeta^*) - \mathcal{L}_{sup}(\mathcal{T}_0, \phi^*) \\
&\leq \frac{1}{\nu}\bar{d}_\zeta(\phi', \phi_\zeta^*) + \kappa \\
&\leq \frac{1}{\nu}\left[\mathcal{L}_{sup}(\phi') - \mathcal{L}_{sup}(\phi_\zeta^*)\right] + \kappa \\
&\leq \frac{1}{\nu}\left[\alpha\frac{1}{1-\tau}(2\epsilon_0 - \tau) - \epsilon^*\right] + \kappa,
\end{aligned}$$

where the last inequality comes from Lemma C.5, where taking $r = 2$. $\qquad\square$

## C.2 Supervised Pretraining

In this section, we will show how multitask finetuning improves the model from supervised pretraining. We present pretraining error in binary classification and $\mathcal{D}_\mathcal{T}(y)$ as uniform. See the result for the general condition with multi-class in Appendix D.

### C.2.1 Supervised Pretraining and Direct Adaptation

In this section, we show the error bound of a foundation model on a target task, where the model is pretrained by supervised loss followed directly by adaptation. For general $y \sim \eta$. Let $p_i := \Pr_{y \sim \eta} \{y = y_i\}$, where $\sum_{i=1}^K p_i = 1$.

**Lemma C.7.** *Suppose $y \sim \eta$ and $l \leq \Pr_{y \sim \eta} \{y = y_i\} \leq u$. Consider a task $\mathcal{T}$ containing $r$ classes, which is a subset of the total class set $\mathcal{C}$. We have $\forall \phi \in \Phi$,*

$$\mathcal{L}_{sup}(\phi) \leq \left(\frac{u}{l}\right)^r \mathcal{L}_{sup-pre}(\phi),$$

*where*

$$\mathcal{L}_{sup-pre}(\phi) = \min_{f \in \mathcal{F}} \mathbb{E}_{x,y} \left[\ell \left(f \circ \phi(x), y\right)\right]. \tag{31}$$

*Proof of Appendix C.2.1.* We first prove $r = 3$, where $\mathcal{T} = \{y_1, y_2, y_3\}$. Then in supervised pretraining, we have:

$$\mathcal{L}_{sup-pre}(\phi) = \min_{f \in \mathcal{F}} \mathbb{E}_{y \sim \mathcal{T}} \mathbb{E}_{x \sim y} \left[\ell \left(f \circ \phi(x), y\right)\right]. \tag{32}$$

Let $f = (f_1, f_2, f_3)^\top$ be the best linear classifier on top of $\phi$, the prediction logits are $g(x) = f \circ \phi(x) = (g_1(x), g_2(x), g_3(x))^\top$. Then we have:

$$\mathbb{E}_{x \sim y_1} \left[\ell \left(g \circ \phi(x), y\right)\right] = -\log \frac{\exp(g_1(x))}{\sum_{k=1}^3 \exp(g_k(x))}.$$

We let $y_k(x) = \exp(g_k(x)), k = 1, 2, 3$. Then

$$\mathcal{L}_{sup-pre}(\phi)$$
$$= -\left[p_1 \mathbb{E}_{x \sim y_1} \left(\log \frac{y_1(x)}{\sum_{k=1}^3 y_k(x)}\right) + p_2 \mathbb{E}_{x \sim y_2} \left(\log \frac{y_2(x)}{\sum_{k=1}^3 y_k(x)}\right) + p_3 \mathbb{E}_{x \sim y_3} \left(\log \frac{y_3(x)}{\sum_{k=1}^3 y_k(x)}\right)\right]$$
$$= p_1 \mathbb{E}_{x \sim y_1} \left(\log \frac{\sum_{k=1}^3 y_k(x)}{y_1(x)}\right) + p_2 \mathbb{E}_{x \sim y_2} \left(\log \frac{\sum_{k=1}^3 y_k(x)}{y_2(x)}\right) + p_3 \mathbb{E}_{x \sim y_3} \left(\log \frac{\sum_{k=1}^3 y_k(x)}{y_3(x)}\right).$$

Recall

$$\mathcal{L}_{sup}(\mathcal{T}, \phi) := \min_{\boldsymbol{w}} \mathbb{E}_{y \sim \mathcal{T}} \mathbb{E}_{x \sim \mathcal{D}(y)} \left[\ell \left(\boldsymbol{w}^\top \phi(x), y\right)\right]. \tag{33}$$

Consider

$$\mathcal{L}^*_{sup}(\mathcal{T}, \phi) := \mathbb{E}_{y \sim \mathcal{T}} \mathbb{E}_{x \sim \mathcal{D}(y)} \left[\ell \left(\boldsymbol{w}^\top \phi(x), y\right)\right], \tag{34}$$

where $w$ is the corresponding sub-vector of $f$ according to task (for e.g., $w = (f_1, f_2)^\top$ if $\mathcal{T} = \{y_1, y_2\}$). Then we have

$$
\begin{aligned}
\mathcal{L}^*_{sup}(\mathcal{T}, \phi) = &- \frac{p_1 p_2}{p_1 p_2 + p_1 p_3 + p_2 p_3} \cdot \frac{1}{2} \left[ \mathbb{E}_{x \sim y_1} \left( \log \frac{y_1(x)}{y_1(x) + y_2(x)} \right) + \mathbb{E}_{x \sim y_2} \left( \log \frac{y_2(x)}{y_1(x) + y_2(x)} \right) \right] \\
&- \frac{p_1 p_3}{p_1 p_2 + p_1 p_3 + p_2 p_3} \cdot \frac{1}{2} \left[ \mathbb{E}_{x \sim y_1} \left( \log \frac{y_1(x)}{y_1(x) + y_3(x)} \right) + \mathbb{E}_{x \sim y_3} \left( \log \frac{y_3(x)}{y_1(x) + y_3(x)} \right) \right] \\
&- \frac{p_2 p_3}{p_1 p_2 + p_1 p_3 + p_2 p_3} \cdot \frac{1}{2} \left[ \mathbb{E}_{x \sim y_2} \left( \log \frac{y_2(x)}{y_2(x) + y_3(x)} \right) + \mathbb{E}_{x \sim y_3} \left( \log \frac{y_3(x)}{y_2(x) + y_3(x)} \right) \right] \\
= &\frac{p_1 p_2}{p_1 p_2 + p_1 p_3 + p_2 p_3} \cdot \frac{1}{2} \left[ \mathbb{E}_{x \sim y_1} \left( \log \frac{y_1(x) + y_2(x)}{y_1(x)} \right) + \mathbb{E}_{x \sim y_2} \left( \log \frac{y_1(x) + y_2(x)}{y_2(x)} \right) \right] \\
&\frac{p_1 p_3}{p_1 p_2 + p_1 p_3 + p_2 p_3} \cdot \frac{1}{2} \left[ \mathbb{E}_{x \sim y_1} \left( \log \frac{y_1(x) + y_3(x)}{y_1(x)} \right) + \mathbb{E}_{x \sim y_3} \left( \log \frac{y_1(x) + y_3(x)}{y_3(x)} \right) \right] \\
&\frac{p_2 p_3}{p_1 p_2 + p_1 p_3 + p_2 p_3} \cdot \frac{1}{2} \left[ \mathbb{E}_{x \sim y_2} \left( \log \frac{y_2(x) + y_3(x)}{y_2(x)} \right) + \mathbb{E}_{x \sim y_3} \left( \log \frac{y_2(x) + y_3(x)}{y_3(x)} \right) \right].
\end{aligned}
$$

By observing the terms with $y_1(x)$ as denominator (similar as $y_2(x), y_3(x)$), we want to prove:

$$
p_1 \left( \frac{u}{l} \right)^2 \geq \frac{1}{2} \left( \frac{p_1 p_2 + p_1 p_3}{p_1 p_2 + p_1 p_3 + p_2 p_3} \right).
$$

This obtained by $\left( \frac{u}{l} \right)^2 \geq \frac{1}{3} \frac{u}{l^2}$.

We have

$$
\mathcal{L}^*_{sup}(\mathcal{T}, \phi) \leq \left( \frac{u}{l} \right)^2 \mathcal{L}_{sup-pre}(\phi).
$$

For the general $K$-class setting, we follow similar steps, we have

$$
\mathcal{L}_{sup-pre}(\phi) = - \left[ \sum_{i=1}^{r} p_i \, \mathbb{E}_{x \sim y_i} \left( \log \frac{y_i(x)}{\sum_{k=1}^{K} y_k(x)} \right) \right].
$$

We denote $J$ as all possible $r$ product of $p_i \in \{p_1, \dots, p_K\}$, $J = \{p_1 \cdots p_r, \dots\}$. Similarly, we have

$$
\mathcal{L}^*_{sup}(\mathcal{T}, \phi) = - \frac{1}{r} \left\{ \sum_{\mathcal{T} \subsetneq \mathcal{C}} \left[ \frac{\prod_{i \in \mathcal{T}} p_i}{J} \sum_{i \in \mathcal{T}} \mathbb{E}_{x \sim y_i} \left( \log \frac{y_i(x)}{\sum_{j \in \mathcal{T}} y_j(x)} \right) \right] \right\},
$$

where $\mathcal{T}$ are all tasks with $r$ classes. By observing, inside the summation there are in total $\binom{K-1}{r-1}$ terms with $y_1(x)$ as the numerator, where corresponding probability is

$$
\frac{p_1 \prod_{i \in \mathcal{T}, i \neq 1} p_i}{J},
$$

where each term can be upper bounded by $- \left( \frac{u}{l} \right)^r p_1 \, \mathbb{E}_{x \sim y_i} \left( \log \frac{y_i(x)}{\sum_{k=1}^{K} y_k(x)} \right)$ (similar as $y_j(x), j \in \mathcal{T}$). $\qquad \square$

We state the theorem below.

**Theorem C.8.** *Assume Assumption 1 and that $\Phi$ has $\nu$-diversity and $\kappa$-consistency with respect to $\phi^*, \phi^*_\zeta$. Suppose $\hat{\phi}$ satisfies $\hat{\mathcal{L}}_{sup-pre}(\hat{\phi}) \leq \epsilon_0$. Let $p_i := \Pr_{y \sim \eta} \{y = y_i\}$, where $\sum_{i=1}^{K} p_i = 1$. Let $\rho := \frac{\max_i p_i}{\min_j p_j}$. Consider pretraining set $\mathcal{S}_{sup-pre} := \{x_i, y_i\}_{i=1}^{N}$, for any $\delta \geq 0$, if*

$$
N \geq \frac{1}{\epsilon_0} \left[ 8LR\sqrt{K} \mathcal{R}_N(\Phi) + \frac{8C^2}{\epsilon_0} \log(\frac{2}{\delta}) \right].
$$

*Then with probability $1 - \delta$, for any target task $\mathcal{T}_0 \subset \mathcal{C}_0$,*

$$
\mathcal{L}_{sup}(\mathcal{T}_0, \hat{\phi}) - \mathcal{L}_{sup}(\mathcal{T}_0, \phi^*) \leq \frac{1}{\nu} \left[ 2\rho^2 \epsilon_0 - \epsilon^* \right] + \kappa. \tag{35}
$$

*Proof of Theorem C.8.* The proof follows similar steps in Theorem C.2. For supervised pretraining, the sample complexity is similar to Theorem C.2, note that there is an extra $\sqrt{K}$ term. We show how we have this term below:

Consider function class
$$\mathcal{G}_\ell = \left\{ g_{W,\phi}(x,y) : g_{W,\phi}(x,y) = \ell(W^\top \phi(x_j^i), y_j^i), \phi \in \Phi, \|W\|_2 \le B \right\}.$$

The Rademacher complexity is
$$\mathcal{R}_n(\mathcal{G}_\ell) = \underset{\{\sigma_i\}_{j=1}^n, \{x_j, y_j\}_{j=1}^n}{\mathbb{E}} \left[ \sup_{\ell \in \mathcal{G}_\ell} \sum_{j=1}^n \sigma_j \ell(W \cdot \phi(x_j), y_j) \right].$$

Then from Lemma C.3, the pretraining is a large task with classification among $K$ classes.
$$\mathcal{R}_n(\mathcal{G}_\ell) \le 4\sqrt{K} L B \mathcal{R}_n(\Phi).$$

Then by Theorem 3.3 in Mohri et al. (2018), with **(A1)** and **(A3)**, we have for $\forall \phi \in \Phi$ with probability $1 - \delta$,

$$\mathcal{L}_{sup-pre}(\phi) - \widehat{\mathcal{L}}_{sup-pre}(\phi) \le \frac{4LR\sqrt{K}\mathcal{R}_N(\Phi)}{N} + C\sqrt{\frac{\log \frac{1}{\delta}}{N}}. \tag{36}$$

To have above $\le \epsilon_0$, we have sample complexity
$$N \ge \frac{1}{\epsilon_0} \left[ 8LR\sqrt{K}\mathcal{R}_N(\Phi) + \frac{8C^2}{\epsilon_0} \log(\frac{2}{\delta}) \right].$$

With the above sample complexity of $\mathcal{S}_{sup-pre} = \{x_i, y_i\}_{i=1}^N$, we have pretraining $\hat{\phi}$
$$\mathcal{L}_{sup-pre}(\hat{\phi}) \le 2\epsilon_0.$$

Recall $\nu$-diversity and $\kappa$-consistency, with respect to $\phi^*, \phi_\zeta^*$, for target task $\mathcal{T}_0$, we have that for $\hat{\phi}$ and $\phi^*$,

$$\mathcal{L}_{sup}(\mathcal{T}_0, \hat{\phi}) - \mathcal{L}_{sup}(\mathcal{T}_0, \phi^*) \le d_{\mathcal{C}_0}(\hat{\phi}, \phi_\zeta^*) + \mathcal{L}_{sup}(\mathcal{T}_0, \phi_\zeta^*) - \mathcal{L}_{sup}(\mathcal{T}_0, \phi^*) \tag{37}$$

$$\le \bar{d}_\zeta(\hat{\phi}, \phi_\zeta^*)/\nu + \kappa \tag{38}$$

$$\le \frac{1}{\nu} \left[ \mathcal{L}_{sup}(\hat{\phi}) - \mathcal{L}_{sup}(\phi_\zeta^*) \right] + \kappa \tag{39}$$

$$\le \frac{1}{\nu} \left[ \rho^2 \mathcal{L}_{sup-pre}(\hat{\phi}) - \epsilon^* \right] + \kappa \tag{40}$$

$$\le \frac{1}{\nu} \left[ 2\rho^2 \epsilon_0 - \epsilon^* \right] + \kappa \tag{41}$$

where the second to last inequality comes from Lemma C.7.

$\square$

### C.2.2 SUPERVISED PRETRAINING AND MULTITASK FINETUNING

In this section, we show the error bound of a supervised pretrained foundation model on a target task can be further reduced by multitask finetuning. We follow similar steps in Appendix C.1.2. Recall Definition 3, similar to Lemma C.5, we introduce the following lemma under supervised pretraining loss.

**Lemma C.9.** *Assume Assumption 1 and that $\Phi$ has $(\nu, \epsilon)$-diversity for $\zeta$ and $\mathcal{C}_0$. Suppose for some small constant $\alpha \in (0, 1)$, we solve Equation (2) with empirical loss lower than $\epsilon_1 = \frac{\alpha}{3} 2\rho^2 \epsilon_0$ and obtain $\phi'$. For any $\delta > 0$, if*

$$M \ge \frac{1}{\epsilon_1} \left[ 4\sqrt{2}\tilde{L}\mathcal{R}_M(\Phi(\tilde{\epsilon})) + \frac{4C^2}{\epsilon_1} \log(\frac{2}{\delta}) \right], Mm \ge \frac{1}{\epsilon_1} \left[ 16LB\mathcal{R}_{Mm}(\Phi(\tilde{\epsilon})) + \frac{4C^2}{\epsilon_1} \log(\frac{2}{\delta}) \right],$$

*then expected supervised loss $\mathcal{L}_{sup}(\phi') \le 2\alpha\rho^2 \epsilon_0$, with probability $1 - \delta$.*

*Proof of Lemma C.9.* The steps follow similar steps in Lemma C.5. □

We state the main theorem below.

**Theorem C.10.** *Assume Assumption 1 and that $\Phi$ has $(\nu, \epsilon)$-diversity for $\zeta$ and $\mathcal{C}_0$. Suppose for some small constant $\alpha \in (0, 1)$, we solve Equation (2) with empirical loss lower than $\epsilon_1 = \frac{\alpha}{3} 2\rho^2 \epsilon_0$ and obtain $\phi'$. For any $\delta > 0$, if*

$$M \geq \frac{1}{\epsilon_1} \left[ 4\sqrt{2}\tilde{L}\mathcal{R}_M(\Phi(\tilde{\epsilon})) + \frac{4C^2}{\epsilon_1} \log(\frac{2}{\delta}) \right], Mm \geq \frac{1}{\epsilon_1} \left[ 16LB\mathcal{R}_{Mm}(\Phi(\tilde{\epsilon})) + \frac{4C^2}{\epsilon_1} \log(\frac{2}{\delta}) \right],$$

*then with probability $1 - \delta$, for any target task $\mathcal{T}_0 \subseteq \mathcal{C}_0$,*

$$\mathcal{L}_{sup}(\mathcal{T}_0, \phi') - \mathcal{L}_{sup}(\mathcal{T}_0, \phi^*) \leq \frac{1}{\nu} \left( 2\alpha\rho^2\epsilon_0 - \mathcal{L}_{sup}(\phi^*) \right) + \epsilon. \tag{42}$$

*Proof of Theorem C.10.* Recall $\nu$-diversity and $\kappa$-consistency, with respect to $\phi^*, \phi_\zeta^*$, for target task $\mathcal{T}_0$, we have that for $\hat{\phi}$ and $\phi^*$,

$$\mathcal{L}_{sup}(\mathcal{T}_0, \phi') - \mathcal{L}_{sup}(\mathcal{T}_0, \phi^*) \leq d_{\mathcal{C}_0}(\phi', \phi_\zeta^*) + \mathcal{L}_{sup}(\mathcal{T}_0, \phi_\zeta^*) - \mathcal{L}_{sup}(\mathcal{T}_0, \phi^*) \tag{43}$$

$$\leq \bar{d}_\zeta(\phi', \phi_\zeta^*)/\nu + \kappa \tag{44}$$

$$\leq \frac{1}{\nu} \left[ \mathcal{L}_{sup}(\phi') - \mathcal{L}_{sup}(\phi_\zeta^*) \right] + \kappa \tag{45}$$

$$\leq \frac{1}{\nu} \left( 2\alpha\rho^2\epsilon_0 - \epsilon^* \right) + \kappa, \tag{46}$$

where the last inequality comes from Lemma C.9.

□

## C.3 MASKED LANGUAGE PRETRAINING

The theoretical guarantee in masked language pretraining follows the same error bound in supervised pretraining, with $K$ representing the size of the vocabulary.

## C.4 UNIFIED MAIN THEORY

We now prove the main theory below. We first re-state the theorem.

**Theorem 3.1.** (No Multitask Finetuning) *Assume Assumption 1 and that $\Phi$ has $\nu$-diversity and $\kappa$-consistency with respect to $\phi^*$ and $\phi_\zeta^*$. Suppose $\hat{\phi}$ satisfies $\hat{\mathcal{L}}_{pre}(\hat{\phi}) \leq \epsilon_0$. Let $\tau := \Pr_{(y_1, y_2) \sim \eta^2} \{y_1 = y_2\}$. Then for any target task $\mathcal{T}_0 \subseteq \mathcal{C}_0$,*

$$\mathcal{L}_{sup}(\mathcal{T}_0, \hat{\phi}) - \mathcal{L}_{sup}(\mathcal{T}_0, \phi^*) \leq \frac{1}{\nu} \left[ \frac{2\epsilon_0}{1 - \tau} - \mathcal{L}_{sup}(\phi_\zeta^*) \right] + \kappa. \tag{3}$$

*Proof of Theorem 3.1.* The result is a direct combination of Theorem C.2 and Theorem C.8. □

**Theorem 3.2.** (With Multitask Finetuning) *Assume Assumption 1 and that $\Phi$ has $\nu$-diversity and $\kappa$-consistency with respect to $\phi^*$ and $\phi_\zeta^*$. Suppose for some constant $\alpha \in (0, 1)$, we solve Equation (2) with empirical loss lower than $\epsilon_1 = \frac{\alpha}{3} \frac{2\epsilon_0}{1 - \tau}$ and obtain $\phi'$. For any $\delta > 0$, if for $\tilde{\epsilon} = \hat{\mathcal{L}}_{pre}(\phi')$,*

$$M \geq \frac{1}{\epsilon_1} \left[ 4\sqrt{2}\tilde{L}\mathcal{R}_M(\Phi(\tilde{\epsilon})) + \frac{4C^2}{\epsilon_1} \log(\frac{2}{\delta}) \right], Mm \geq \frac{1}{\epsilon_1} \left[ 16LB\mathcal{R}_{Mm}(\Phi(\tilde{\epsilon})) + \frac{4C^2}{\epsilon_1} \log(\frac{2}{\delta}) \right],$$

*then with probability $1 - \delta$, for any target task $\mathcal{T}_0 \subseteq \mathcal{C}_0$,*

$$\mathcal{L}_{sup}(\mathcal{T}_0, \phi') - \mathcal{L}_{sup}(\mathcal{T}_0, \phi^*) \leq \frac{1}{\nu} \left[ \alpha \frac{2\epsilon_0}{1 - \tau} - \mathcal{L}_{sup}(\phi_\zeta^*) \right] + \kappa. \tag{4}$$

*Proof of Theorem 3.2.* Follow the similar steps in proof of Lemma C.5, we have

$$\mathcal{L}_{sup}(\phi') \le 2\epsilon_1 \le \alpha \frac{2\rho^2}{1-\tau}\epsilon_0.$$

Recall $\nu$-diversity and $\kappa$-consistency, with respect to $\phi^*, \phi^*_\zeta$, for target task $\mathcal{T}_0$, we have that for $\phi'$ and $\phi^*$,

$$\mathcal{L}_{sup}(\mathcal{T}_0, \phi') - \mathcal{L}_{sup}(\mathcal{T}_0, \phi^*) \le d_{\mathcal{C}_0}(\phi', \phi^*_\zeta) + \mathcal{L}_{sup}(\mathcal{T}_0, \phi^*_\zeta) - \mathcal{L}_{sup}(\mathcal{T}_0, \phi^*) \tag{47}$$

$$\le \bar{d}_\zeta(\phi', \phi^*_\zeta)/\nu + \kappa \tag{48}$$

$$\le \frac{1}{\nu}\left[\mathcal{L}_{sup}(\phi') - \mathcal{L}_{sup}(\phi^*_\zeta)\right] + \kappa \tag{49}$$

$$\le \frac{1}{\nu}\left[\alpha\frac{2\rho^2}{1-\tau}\epsilon_0 - \epsilon^*\right] + \kappa. \tag{50}$$

$\square$

The sample complexity of finetuning depends on $\tilde{\epsilon} = \widehat{\mathcal{L}}_{pre}(\phi')$. Below we show that $\tilde{\epsilon}$ can be upper bounded in finite step finetuning.

**Lemma C.11** (Bounded $\tilde{\epsilon}$). *After finite steps in Multitask finetuning in Algorithm 2, we solve Equation (2) with empirical loss lower than $\epsilon_1 = \frac{\alpha}{3}\frac{1}{1-\tau}(2\epsilon_0 - \tau)$ and obtain $\phi'$. Then there exists a bounded $\tilde{\epsilon}$ such that $\phi' \in \Phi(\tilde{\epsilon})$.*

*Proof of Lemma C.11.* Given finite number of steps and finite step size $\gamma$ in Algorithm 2, we have bounded $\|\phi' - \hat{\phi}\|$. Then with **(A2)** and **(A3)**, using Lemma C.3 and lemma A.3 in Arora et al. (2019), we have $\widehat{\mathcal{L}}_{pre}(\phi)$ is $M$-Lipschitz with respect to $\phi$ with bounded $M$. We have $\exists\, \epsilon$ such that $\widehat{\mathcal{L}}_{pre}(\phi') - \widehat{\mathcal{L}}_{pre}(\hat{\phi}) \le \epsilon\|\phi' - \hat{\phi}\|$. We have $\widehat{\mathcal{L}}_{pre}(\phi') \le \epsilon_0 + \epsilon\|\phi' - \hat{\phi}\|$. Take $\tilde{\epsilon} = \epsilon_0 + \epsilon\|\phi' - \hat{\phi}\|$ yields the result. $\square$

## C.5 Bounded Task Loss by Task Diversity

By the previous lemma and claim, we have the below corollary.

**Corollary C.12.** *Suppose we have $\phi$ in pretraining: for $\forall \phi \in \Phi$, $\mathcal{L}_{sup}(\phi) \le \frac{1}{1-\tau}\left(\frac{u}{l}\right)^r \mathcal{L}_{pre}(\phi)$, where $\mathcal{L}_{pre}(\phi)$ is $\mathcal{L}_{con-pre}(\phi)$ if contrastive learning and $\mathcal{L}_{sup-pre}(\phi)$ if supervised learning.*

Consider $\rho = \frac{u}{l}$ and Corollary C.12,

Recall $\nu$-diversity and $\kappa$-consistency, with respect to $\phi^*, \phi^*_\zeta$, for target task $\mathcal{T}_0$, we have that for $\hat{\phi}$ and $\phi^*$,

$$\mathcal{L}_{sup}(\mathcal{T}_0, \hat{\phi}) - \mathcal{L}_{sup}(\mathcal{T}_0, \phi^*) \le d_{\mathcal{C}_0}(\hat{\phi}, \phi^*_\zeta) + \mathcal{L}_{sup}(\mathcal{T}_0, \phi^*_\zeta) - \mathcal{L}_{sup}(\mathcal{T}_0, \phi^*) \tag{51}$$

$$\le \bar{d}_\zeta(\hat{\phi}, \phi^*_\zeta)/\nu + \kappa \tag{52}$$

$$\le \frac{1}{\nu}\left[\mathcal{L}_{sup}(\hat{\phi}) - \mathcal{L}_{sup}(\phi^*_\zeta)\right] + \kappa \tag{53}$$

$$\le \frac{1}{\nu}\left[\frac{\rho^r}{1-\tau}\mathcal{L}_{pre}(\hat{\phi}) - \mathcal{L}_{sup}(\phi^*)\right] + \kappa. \tag{54}$$

## D  Multi-class Classification

In this section, we provide a general result for multi-classes.

### D.1 CONTRASTIVE PRETRAINING

**Lemma D.1** (Theorem 6.1 in Arora et al. (2019))**.** *For multi-classes, we have*

$$\mathcal{L}_{sup}(\phi) \leq \mathcal{L}_{sup}^{\mu}(\phi) \leq \frac{1}{1-\tau_r}\mathcal{L}_{con-pre}(\phi), \tag{55}$$

*where* $\tau_r = \mathbb{E}_{(y,y_1^-,\ldots,y_{r-1}^-)\sim\eta^r} \mathbb{1}\{y \text{ does not appear in } (y_1^-,\ldots,y_{r-1}^-)\}.$

*Proof of Lemma D.1.* The proof of Lemma D.1 follows the first two steps in the proof of Theorem B.1 of Arora et al. (2019). we denote distribution of $y \sim \mathcal{T}$ as $\mathcal{D}_{\mathcal{T}}(y)$ and it's uniform distribution. □

We first provide contrastive pretraining error similar to Theorem C.2 in a multiclass setting.

**Theorem D.2.** *Assume Assumption 1 and that $\Phi$ has $\nu$-diversity and $\kappa$-consistency with respect to $\phi^*, \phi_\zeta^*$. Suppose $\hat{\phi}$ satisfies $\hat{\mathcal{L}}_{con-pre}(\hat{\phi}) \leq \epsilon_0$. Consider a pretraining set $\mathcal{S}_{un} = \left\{x_j, x_j^+, x_{j1}^-, \ldots, x_{j(r-1)}^-\right\}_{j=1}^{N}$. For target task $\mathcal{T}_0$, with sample complexity*

$$N \geq \frac{1}{\epsilon_0}\left[8LR\sqrt{r-1}\mathcal{R}_N(\Phi) + \frac{8C^2}{\epsilon_0}\log(\frac{2}{\delta})\right],$$

*it's sufficient to learn an $\hat{\phi}$ with classification error $\mathcal{L}_{sup}(\mathcal{T}_0, \hat{\phi}) - \mathcal{L}_{sup}(\mathcal{T}_0, \phi^*) \leq \frac{1}{\nu}\left[\frac{2}{1-\tau_r}\epsilon_0 - \epsilon^*\right] + \epsilon$, with probability $1 - \delta$.*

*Proof of Theorem D.2.* Following similar step of proof of Theorem C.2, we have with

$$N \geq \frac{1}{\epsilon_0}\left[8LR\sqrt{r-1}\mathcal{R}_N(\Phi) + \frac{8C^2}{\epsilon_0}\log(\frac{2}{\delta})\right].$$

Then pretraining $\hat{\phi}$

$$\mathcal{L}_{con-pre}(\hat{\phi}) \leq 2\epsilon_0.$$

Recall $\nu$-diversity and $\kappa$-consistency, for target task $\mathcal{T}_0$, we have that for $\hat{\phi}$ and $\phi^*$,

$$\mathcal{L}_{sup}(\mathcal{T}_0, \hat{\phi}) - \mathcal{L}_{sup}(\mathcal{T}_0, \phi^*) \leq \bar{d}_\zeta(\hat{\phi}, \phi_\zeta^*)/\nu + \kappa \tag{56}$$

$$\leq \frac{1}{\nu}\left[\mathcal{L}_{sup}(\hat{\phi}) - \mathcal{L}_{sup}(\phi_\zeta^*)\right] + \kappa \tag{57}$$

$$\tag{58}$$

Consider Lemma D.1, we have:

$$\mathcal{L}_{sup}(\mathcal{T}_0, \hat{\phi}) - \mathcal{L}_{sup}(\mathcal{T}_0, \phi^*) \leq \frac{1}{\nu}\left[\frac{1}{1-\tau_r}\mathcal{L}_{con-pre}(\hat{\phi}) - \epsilon^*\right] + \kappa \tag{59}$$

$$= \frac{1}{\nu}\left(\frac{2\epsilon_0}{1-\tau_r} - \epsilon^*\right) + \kappa. \tag{60}$$

□

Below, we provide our main result similar to Theorem C.6 for multi-classes setting.

**Theorem D.3.** *For target evaluation task $\mathcal{T}_0$, consider the error bound in pretraining is $\mathcal{L}_{sup}(\mathcal{T}_0, \hat{\phi}) - \mathcal{L}_{sup}(\mathcal{T}_0, \phi^*) \leq \frac{1}{\nu}\left[\frac{2\epsilon_0}{1-\tau_r} - \epsilon^*\right] + \kappa$. Consider $\alpha$ as any small constant, for any $\epsilon_1 < \frac{\alpha}{3}\frac{2\epsilon_0}{1-\tau_r}$, consider*

*a multitask finetuning set* $\mathcal{S} = \{(x_j^i, y_j^i) : i \in [M], j \in [m]\}$, *with $M$ number of tasks, and $m$ number of samples in each task. Then, with sample complexity*

$$M \geq \frac{1}{\epsilon_1} \left[ 4\sqrt{2}\tilde{L}\mathcal{R}_M(\Phi(\tilde{\epsilon})) + \frac{4C^2}{\epsilon_1} \log(\frac{2}{\delta}) \right]$$

$$Mm \geq \frac{1}{\epsilon_1} \left[ 8LB\sqrt{r-1}\mathcal{R}_{Mm}(\Phi(\tilde{\epsilon})) + \frac{4C^2}{\epsilon_1} \log(\frac{2}{\delta}) \right].$$

*Solving Equation* (2) *with empirical risk lower than $\epsilon_1$ is sufficient to learn an $\phi'$ with classification error* $\mathcal{L}_{sup}(\mathcal{T}_0, \phi') - \mathcal{L}_{sup}(\mathcal{T}_0, \phi^*) \leq \frac{1}{\nu}(\alpha \frac{2\epsilon_0}{1-\tau_r} - \epsilon^*) + \kappa$, *with probability $1 - \delta$.*

*Proof of Theorem D.3.* Recalling Lemma C.3 and Lemma C.5, the proof follows the same steps in the proof of Theorem C.6, with different $r$.

$\square$

## D.2 SUPERVISED PRETRAINING

We first provide contrastive pretraining error similar to Theorem C.8 in the multiclass setting.

**Theorem D.4.** *Assume Assumption 1 and that $\Phi$ has $\nu$-diversity and $\kappa$-consistency with respect to $\phi^*, \phi_\zeta^*$. Suppose $\hat{\phi}$ satisfies $\hat{\mathcal{L}}_{sup-pre}(\hat{\phi}) \leq \epsilon_0$. Let $p_i := \Pr_{y \sim \eta}\{y = y_i\}$, where $\sum_{i=1}^K p_i = 1$. Let $\rho := \frac{\max_i p_i}{\min_j p_j}$. Consider pretraining set $\mathcal{S}_{sup-pre} := \{x_i, y_i\}_{i=1}^N$, for any $\delta \geq 0$, if*

$$N \geq \frac{1}{\epsilon_0} \left[ 8LR\sqrt{K}\mathcal{R}_N(\Phi) + \frac{8C^2}{\epsilon_0} \log(\frac{2}{\delta}) \right].$$

*Then with probability $1 - \delta$, for any target task $\mathcal{T}_0 \subset \mathcal{C}_0$,*

$$\mathcal{L}_{sup}(\mathcal{T}_0, \hat{\phi}) - \mathcal{L}_{sup}(\mathcal{T}_0, \phi^*) \leq \frac{1}{\nu} \left[ 2\rho^r \epsilon_0 - \mathcal{L}_{sup}(\phi_\zeta^*) \right] + \kappa. \tag{61}$$

*Proof of Theorem D.4.* The proof follows similar steps of Theorem C.8. $\square$

Below, we provide our main result similar to Theorem C.10 for multi-classes setting.

**Theorem D.5.** *Assume Assumption 1 and that $\Phi$ has $\nu$-diversity and $\kappa$-consistency with respect to $\phi^*, \phi_\zeta^*$. Suppose for some small constant $\alpha \in (0, 1)$, we solve Equation (2) with empirical loss lower than $\epsilon_1 = \frac{\alpha}{3} 2\rho^r \epsilon_0$ and obtain $\phi'$. For any $\delta > 0$, if*

$$M \geq \frac{1}{\epsilon_1} \left[ 4\sqrt{2}\tilde{L}\mathcal{R}_M(\Phi(\tilde{\epsilon})) + \frac{4C^2}{\epsilon_1} \log(\frac{2}{\delta}) \right], Mm \geq \frac{1}{\epsilon_1} \left[ 8LB\sqrt{r-1}\mathcal{R}_{Mm}(\Phi(\tilde{\epsilon})) + \frac{4C^2}{\epsilon_1} \log(\frac{2}{\delta}) \right],$$

*then with probability $1 - \delta$, for any target task $\mathcal{T}_0 \subseteq \mathcal{C}_0$,*

$$\mathcal{L}_{sup}(\mathcal{T}_0, \phi') - \mathcal{L}_{sup}(\mathcal{T}_0, \phi^*) \leq \frac{1}{\nu} \left( 2\alpha\rho^r \epsilon_0 - \mathcal{L}_{sup}(\phi_\zeta^*) \right) + \kappa. \tag{62}$$

*Proof of Theorem D.5.* Recalling Lemma C.3 and Lemma C.5, the proof follows the same steps in the proof of Theorem C.10, with different $r$. $\square$

## E LINEAR CASE STUDY

In this section, we provide a full analysis of the linear case study to provide intuition about our consistency, diversity, and task selection algorithm. Intuitively, we have multiple classes, each centered around its mean vector. Target data has a subset of classes, while training data has another subset of classes. Consistency and diversity are related to how these two subsets overlap, i.e., the number of shared features and the number of disjoint features. Then, we can link it to the task selection algorithm.

In this section, $z_i$ means the $i$-th dimension of vector $z$ rather than the sample index.

### E.1 Problem Setup

**Linear Data and Tasks.** We consider dictionary learning or sparse coding settings, which is a classic data model (e.g., Olshausen & Field (1997); Vinje & Gallant (2000); Blei et al. (2003); Shi et al. (2022; 2023c)). Let $\mathcal{X} \subseteq \mathbb{R}^d$ be the input space and we have input data $x \in \mathcal{X}$. Suppose $Q \in \mathbb{R}^{d \times D}$ is an unknown dictionary with $D$ columns that can be regarded as patterns or features. For simplicity, assume $d = D$ and $Q$ is orthonormal. We have $z \in \{0, -1, +1\}^d$ as a latent class, where $z$ is a hidden vector that indicates the presence of each pattern. Each latent class $z$ has a distribution $\mathcal{D}_z(x)$ over inputs $x$. We assume $\mathcal{D}_z(x)$ be a distribution with mean $Qz$, i.e., $x = Qz + \boldsymbol{e}_z$, where $\boldsymbol{e}_z \in \mathbb{R}^d$ is some noise vector drawing from a zero-mean distribution.

For simplicity, we consider each task to be a binary classification task, where $\mathcal{Y} = \{-1, +1\}$ is the label space. In each task (in multitask finetuning or target task), we have two latent classes $z, z'$ (denote the task as $\mathcal{T}_{z,z'}$) and we randomly assign $-1$ and $+1$ to each latent class. W.l.o.g., we have in $\mathcal{T}_{z,z'}$:

$$x = \begin{cases} Qz + \boldsymbol{e}_z, & \text{if } y = -1 \\ Qz' + \boldsymbol{e}_{z'}, & \text{if } y = +1. \end{cases} \tag{63}$$

For simplicity, we consider a balanced class setting in all tasks, i.e., $p(y = -1) = p(y = +1) = \frac{1}{2}$.

Now, we define multitask finetuning tasks and target tasks. Suppose there is a set of latent classes $\mathcal{C} \subseteq \{0, -1, +1\}^d$ used for multitask finetuning tasks, which has an index set $J_{\mathcal{C}} \subseteq [d], k_{\mathcal{C}} := |J_{\mathcal{C}}|$ such that for any $z \in \mathcal{C}$, we have $z_{J_{\mathcal{C}}} \in \{-1, +1\}^{k_{\mathcal{C}}}$ and $z_{[d] \setminus J_{\mathcal{C}}} \in \{0\}^{d - k_{\mathcal{C}}}$. Similarly, suppose there is a set of latent classes $\mathcal{C}_0 \subseteq \{0, -1, +1\}^d$ used for target tasks whose index set is $J_0 \subseteq [d], k_0 := |J_0|$. Note that $J_{\mathcal{C}}$ may or may not overlap with $J_0$ and denote the set of features encoded both by $\mathcal{C}_0$ and $\mathcal{C}$ as $L_{\mathcal{C}} := J_0 \cap J_{\mathcal{C}}, l_{\mathcal{C}} := |L_{\mathcal{C}}|$. Intuitively, $L_{\mathcal{C}}$ represents the target features covered by training data. Let $\zeta$ over $\mathcal{C} \times \mathcal{C}$ be the distribution of multitask finetuning tasks. Then, in short, our data generation pipeline for multitask finetuning tasks is (1) sample two latent classes $(z, z') \sim \zeta$ as a task $\mathcal{T}_{z,z'}$; (2) assign label $-1, +1$ to two latent classes; (3) sample input data from $\mathcal{D}_z(x)$ and $\mathcal{D}_{z'}(x)$ with balanced probabilities.

For simplicity, we have a symmetric assumption and a non-degenerate assumption for $\zeta$. Symmetric assumption means each dimension is equal important and non-degenerate assumption means any two dimensions are not determined by each other in all tasks.

**Assumption 3** (Symmetric). *We assume for any multitask finetuning tasks distribution $\zeta$, for any $j, k \in J_{\mathcal{C}}$, switching two dimensions $z_j$ and $z_k$ does not change the distribution $\zeta$.*

**Assumption 4** (Non-degenerate). *We assume for any multitask finetuning tasks distribution $\zeta$, for any $j, k \in J_{\mathcal{C}}$, over $\zeta$ we have $p(z_j = z'_j, z_k \neq z'_k) > 0$.*

**Remark E.1.** *There exists many $\zeta$ satisfying above assumptions, e.g., (1) $z_{J_{\mathcal{C}}}$ and $z'_{J_{\mathcal{C}}}$ uniformly sampling from $\{-1, +1\}^{k_{\mathcal{C}}}$; or (2) let $k_{\mathcal{C}} = 2$, $z_{J_{\mathcal{C}}}$ and $z'_{J_{\mathcal{C}}}$ uniformly sampling from $\{(+1, +1), (-1, +1), (+1, -1)\}$ (note that uniformly sampling from $\{(+1, +1), (-1, -1)\}$ does not satisfy non-degenerate assumption). Also, we note that even when $\mathcal{C} = \mathcal{C}_0$, the target latent class may not exist in the multitask finetuning tasks.*

**Linear Model and Loss Function.** Let $\Phi$ be the hypothesis class of models $\phi : \mathcal{X} \to \overline{\mathcal{Z}}$, where $\overline{\mathcal{Z}} \subseteq \mathbb{R}^d$ is the output space of the model. We consider a linear model class with regularity Assumption 1, i.e., $\Phi = \{\phi \in \mathbb{R}^{d \times d} : \|\phi\|_F \leq R\}$ and linear head $w \in \mathbb{R}^d$ where $\|w\|_2 \leq B$. Thus, the final output of the model and linear head is $w^\top \phi x$. We use linear loss in Shi et al. (2023a), i.e., $\ell(w^\top \phi x, y) = -y w^\top \phi x$ and we have

$$\mathcal{L}_{sup}(\mathcal{T}, \phi) := \min_{w \in \mathbb{R}^d, \|w\|_2 \leq B} \mathbb{E}_{z, y \sim \mathcal{T}} \mathbb{E}_{x \sim \mathcal{D}_z(x)} \left[ \ell\left( w^\top \phi x, y \right) \right] \tag{64}$$

$$\mathcal{L}_{sup}(\phi) := \mathbb{E}_{\mathcal{T} \sim \zeta} \left[ \mathcal{L}_{sup}(\mathcal{T}, \phi) \right] \tag{65}$$

$$\phi^*_\zeta := \arg\min_{\phi \in \Phi} \mathcal{L}_{sup}(\phi), \tag{66}$$

where $\phi^*_\zeta$ is the optimal representation for multitask finetuning.

### E.2 DIVERSITY AND CONSISTENCY ANALYSIS

#### E.2.1 OPTIMAL REPRESENTATION FOR MULTITASK FINETUNING

**Lemma E.1.** *Assume Assumption 3 and Assumption 4. We have $\phi_\zeta^* = U\Lambda^*Q^{-1}$, where $U$ is any orthonormal matrix, $\Lambda^* = diag(\lambda^*)$. For any $i \in J_\mathcal{C}$, $\lambda_i^* = \frac{R}{\sqrt{k_\mathcal{C}}}$ and $\lambda_i^* = 0$ otherwise.*

*Proof of Lemma E.1.* We have the singular value decomposition $\phi = U\Lambda V^\top$, where $\Lambda = \text{diag}(\lambda)$, where $\lambda \in \mathbb{R}^d$. Then, we have

$$\mathcal{L}_{sup}(\phi) = \mathop{\mathbb{E}}_{\mathcal{T}\sim\zeta}[\mathcal{L}_{sup}(\mathcal{T}, \phi)] \tag{67}$$

$$= \mathop{\mathbb{E}}_{\mathcal{T}\sim\zeta}\left[\min_{w\in\mathbb{R}^d,\|w\|_2\leq B}\mathop{\mathbb{E}}_{z,y\sim\mathcal{T}}\mathop{\mathbb{E}}_{x\sim\mathcal{D}_z(x)}\left[\ell\left(w^\top\phi x, y\right)\right]\right] \tag{68}$$

$$= \mathop{\mathbb{E}}_{\mathcal{T}_{z,z'}\sim\zeta}\left[\min_{w\in\mathbb{R}^d,\|w\|_2\leq B}\frac{1}{2}\left(\mathop{\mathbb{E}}_{x\sim\mathcal{D}_z(x)}\left[\ell\left(w^\top\phi x, -1\right)\right] + \mathop{\mathbb{E}}_{x\sim\mathcal{D}_{z'}(x)}\left[\ell\left(w^\top\phi x, +1\right)\right]\right)\right] \tag{69}$$

$$= \frac{1}{2}\mathop{\mathbb{E}}_{\mathcal{T}_{z,z'}\sim\zeta}\left[\min_{w\in\mathbb{R}^d,\|w\|_2\leq B}\mathop{\mathbb{E}}_{x\sim\mathcal{D}_z(x)}\left[w^\top\phi x\right] + \mathop{\mathbb{E}}_{x\sim\mathcal{D}_{z'}(x)}\left[-w^\top\phi x\right]\right] \tag{70}$$

$$= \frac{1}{2}\mathop{\mathbb{E}}_{\mathcal{T}_{z,z'}\sim\zeta}\left[\min_{w\in\mathbb{R}^d,\|w\|_2\leq B}w^\top\phi Qz - w^\top\phi Qz'\right] \tag{71}$$

$$= -\frac{B}{2}\mathop{\mathbb{E}}_{\mathcal{T}_{z,z'}\sim\zeta}[\|\phi Q(z-z')\|_2] \tag{72}$$

$$= -\frac{B}{2}\mathop{\mathbb{E}}_{\mathcal{T}_{z,z'}\sim\zeta}\left[\|\Lambda V^\top Q(z-z')\|_2\right]. \tag{73}$$

W.l.o.g., we can assume $V^\top = Q^{-1}$. As $\|\phi\|_F = \|\Lambda\|_F = \|\lambda\|_2$ thus we have

$$\min_{\phi\in\Phi}\mathcal{L}_{sup}(\phi) = -\frac{B}{2}\max_{\|\Lambda\|_F\leq R}\mathop{\mathbb{E}}_{\mathcal{T}_{z,z'}\sim\zeta}[\|\Lambda(z-z')\|_2]$$

$$= -\frac{B}{2}\max_{\|\lambda\|_2\leq R}\mathop{\mathbb{E}}_{\mathcal{T}_{z,z'}\sim\zeta}\left[\sqrt{\sum_{i=1}^d\lambda_i^2(z_i-z_i')^2}\right]$$

$$= -\frac{B}{2}\max_{\|\lambda\|_2=R}\mathop{\mathbb{E}}_{\mathcal{T}_{z,z'}\sim\zeta}\left[\sqrt{\sum_{i=1}^d\lambda_i^2(z_i-z_i')^2}\right]$$

$$= -B\max_{\|\lambda\|_2=R}\mathop{\mathbb{E}}_{\mathcal{T}_{z,z'}\sim\zeta}\left[\sqrt{\sum_{i\in J_\mathcal{C}}\lambda_i^2\mathbb{1}[z_i\neq z_i']}\right], \tag{74}$$

where $\mathbb{1}[z_i \neq z_i']$ is a Boolean function, mapping True to 1 and False to 0.

Let $\phi_\zeta^* = U\Lambda^* Q^{-1}$ with corresponding $\lambda^*$. Now, we use contradiction to prove for any $j, k \in J_\mathcal{C}$, we have $\lambda_j^* = \lambda_k^*$. W.l.o.g., suppose $\lambda_j^* < \lambda_k^*$,

$$\mathcal{L}_{sup}(\phi_\zeta^*)$$

$$= -B \mathop{\mathbb{E}}_{\mathcal{T}_{z,z'}\sim\zeta}\left[\sqrt{\lambda_j^{*2}\mathbb{1}[z_j \neq z_j'] + \lambda_k^{*2}\mathbb{1}[z_k \neq z_k'] + \sum_{i\in J_\mathcal{C}\setminus\{j,k\}}\lambda_i^{*2}\mathbb{1}[z_i \neq z_i']}\right]$$

$$= -B\left\{p(z_j \neq z_j', z_k \neq z_k')\mathop{\mathbb{E}}_{\mathcal{T}_{z,z'}\sim\zeta}\left[\sqrt{\lambda_j^{*2} + \lambda_k^{*2} + \sum_{i\in J_\mathcal{C}\setminus\{j,k\}}\lambda_i^{*2}\mathbb{1}[z_i \neq z_i']}\,\bigg|\, z_j \neq z_j', z_k \neq z_k'\right]\right.$$

$$+ p(z_j = z_j', z_k \neq z_k')\mathop{\mathbb{E}}_{\mathcal{T}_{z,z'}\sim\zeta}\left[\sqrt{\lambda_k^{*2} + \sum_{i\in J_\mathcal{C}\setminus\{j,k\}}\lambda_i^{*2}\mathbb{1}[z_i \neq z_i']}\,\bigg|\, z_j = z_j', z_k \neq z_k'\right]$$

$$+ p(z_j \neq z_j', z_k = z_k')\mathop{\mathbb{E}}_{\mathcal{T}_{z,z'}\sim\zeta}\left[\sqrt{\lambda_j^{*2} + \sum_{i\in J_\mathcal{C}\setminus\{j,k\}}\lambda_i^{*2}\mathbb{1}[z_i \neq z_i']}\,\bigg|\, z_j \neq z_j', z_k = z_k'\right]\right\}.$$

By symmetric Assumption 3 and non-degenerate Assumption 4, we have $p(z_j = z_j', z_k \neq z_k') = p(z_j \neq z_j', z_k = z_k') > 0$, and

$$\mathop{\mathbb{E}}_{\mathcal{T}_{z,z'}\sim\zeta}\left[\sqrt{\lambda_k^{*2} + \sum_{i\in J_\mathcal{C}\setminus\{j,k\}}\lambda_i^{*2}\mathbb{1}[z_i \neq z_i']}\,\bigg|\, z_j = z_j', z_k \neq z_k'\right]$$

$$+ \mathop{\mathbb{E}}_{\mathcal{T}_{z,z'}\sim\zeta}\left[\sqrt{\lambda_j^{*2} + \sum_{i\in J_\mathcal{C}\setminus\{j,k\}}\lambda_i^{*2}\mathbb{1}[z_i \neq z_i']}\,\bigg|\, z_j \neq z_j', z_k = z_k'\right]$$

$$= \mathop{\mathbb{E}}_{\mathcal{T}_{z,z'}\sim\zeta}\left[\sqrt{\lambda_k^{*2} + \sum_{i\in J_\mathcal{C}\setminus\{j,k\}}\lambda_i^{*2}\mathbb{1}[z_i \neq z_i']}\,\bigg|\, z_j = z_j', z_k \neq z_k'\right]$$

$$+ \mathop{\mathbb{E}}_{\mathcal{T}_{z,z'}\sim\zeta}\left[\sqrt{\lambda_j^{*2} + \sum_{i\in J_\mathcal{C}\setminus\{j,k\}}\lambda_i^{*2}\mathbb{1}[z_i \neq z_i']}\,\bigg|\, z_k \neq z_k', z_j = z_j'\right]$$

$$< 2\mathop{\mathbb{E}}_{\mathcal{T}_{z,z'}\sim\zeta}\left[\sqrt{\frac{\lambda_j^{*2} + \lambda_k^{*2}}{2} + \sum_{i\in J_\mathcal{C}\setminus\{j,k\}}\lambda_i^{*2}\mathbb{1}[z_i \neq z_i']}\,\bigg|\, z_j = z_j', z_k \neq z_k'\right]$$

$$= \mathop{\mathbb{E}}_{\mathcal{T}_{z,z'}\sim\zeta}\left[\sqrt{\frac{\lambda_j^{*2} + \lambda_k^{*2}}{2} + \sum_{i\in J_\mathcal{C}\setminus\{j,k\}}\lambda_i^{*2}\mathbb{1}[z_i \neq z_i']}\,\bigg|\, z_j = z_j', z_k \neq z_k'\right]$$

$$+ \mathop{\mathbb{E}}_{\mathcal{T}_{z,z'}\sim\zeta}\left[\sqrt{\frac{\lambda_j^{*2} + \lambda_k^{*2}}{2} + \sum_{i\in J_\mathcal{C}\setminus\{j,k\}}\lambda_i^{*2}\mathbb{1}[z_i \neq z_i']}\,\bigg|\, z_k \neq z_k', z_j = z_j'\right].$$

where two equality follows Assumption 3 and the inequality follows Jensen's inequality. Let $\phi' = U\Lambda' Q^{-1}$ with corresponding $\lambda'$, where $\lambda_j' = \lambda_k' = \sqrt{\frac{\lambda_j^{*2} + \lambda_k^{*2}}{2}}$ and for any $i \in J_\mathcal{C} \setminus \{j, k\}$, $\lambda_i' = \lambda_i^*$. We have $\|\phi'\|_F = \|\phi_\zeta^*\|_F$ and $\mathcal{L}_{sup}(\phi_\zeta^*) > \mathcal{L}_{sup}(\phi')$ which is a contradiction as we have $\phi_\zeta^*$ is the optimal solution. Thus, for any $j, k \in J_\mathcal{C}$, we have $\lambda_j^* = \lambda_k^*$ and we finish the proof under simple calculation. $\qquad\square$

Now, we are ready to analyze consistency and diversity under this linear case study.

### E.2.2 CONSISTENCY

The intuition is that $\zeta$ not only covers $\mathcal{C}_0$ but contains too much unrelated information. Recall that the consistent term in Definition 2 is $\kappa = \sup_{\mathcal{T}_0 \subseteq \mathcal{C}_0} \left[ \mathcal{L}_{sup}(\mathcal{T}_0, \phi_\zeta^*) - \mathcal{L}_{sup}(\mathcal{T}_0, \phi_0^*) \right]$.

We first define some notation we will use later. Let $\zeta_0$ be a multitask finetuning tasks distribution over $\mathcal{C}_0 \times \mathcal{C}_0$ and denote the corresponding optimal representation model as $\phi_0^*$. Suppose for any target task $\mathcal{T}_0$ contains two latent classes $z, z'$ from $\mathcal{C}_0$. W.l.o.g., denote $z, z'$ differ in $n_0$ entries $(1 \leq n_0 \leq k_0)$, whose $n_{\mathcal{C}}$ entries fall in $L_{\mathcal{C}}$ , where $0 \leq n_{\mathcal{C}} \leq n_0$. Then, we get the lemma below:

**Lemma E.2.** *Assume Assumption 3 and Assumption 4. We have*

$$\kappa = \sup_{\mathcal{T}_0 \subseteq \mathcal{C}_0} \left[ \mathcal{L}_{sup}(\mathcal{T}_0, \phi_\zeta^*) - \mathcal{L}_{sup}(\mathcal{T}_0, \phi_0^*) \right] = BR \left( \sqrt{\frac{n_0}{k_0}} - \sqrt{\frac{n_{\mathcal{C}}}{k_{\mathcal{C}}}} \right). \tag{75}$$

*Proof of Lemma E.2.* Recall $1 \leq n_0 \leq k_0$ and $0 \leq n_{\mathcal{C}} \leq n_0$. By Lemma E.1, we have $\phi_\zeta^* = U\Lambda^* Q^{-1}$, where $U$ is any orthonormal matrix, $\Lambda^* = \text{diag}(\lambda^*)$. For any $i \in J_{\mathcal{C}}$, $\lambda_i^* = \frac{R}{\sqrt{k_{\mathcal{C}}}}$ and $\lambda_i^* = 0$ otherwise. We also have $\phi_0^* = U_0 \Lambda_0^* Q^{-1}$, where $U_0$ is any orthonormal matrix, $\Lambda_0^* = \text{diag}(\lambda^{0,*})$. For any $i \in J_0$, $\lambda_i^{0,*} = \frac{R}{\sqrt{k_0}}$ and $\lambda_i^{0,*} = 0$ otherwise. Thus, we have

$$\kappa = \sup_{\mathcal{T}_0 \subseteq \mathcal{C}_0} \left[ \mathcal{L}_{sup}(\mathcal{T}_0, \phi_\zeta^*) - \mathcal{L}_{sup}(\mathcal{T}_0, \phi_0^*) \right] \tag{76}$$

$$= BR \left( \sqrt{\frac{n_0}{k_0}} - \sqrt{\frac{n_{\mathcal{C}}}{k_{\mathcal{C}}}} \right). \tag{77}$$

$\square$

Let $n_{\mathcal{C}}' = k_{\mathcal{C}} - n_{\mathcal{C}}$. Note this $k_{\mathcal{C}}$ is an increasing factor if $\mathcal{C}$ contains more features. Moreover, $n_{\mathcal{C}}$ is the number of features encoded by both target and training data, representing the information of target data covered by training data, $n_{\mathcal{C}}$ increases as more target information covered by training data, the loss will decrease. $n_{\mathcal{C}}'$ is the number of features encoded in training data but not encoded by target data, representing the un-useful information, $n_{\mathcal{C}}'$ increases as more un-related information is covered by training data, the loss will increase.

### E.2.3 DIVERSITY

We first review some definitions in Definition 1. The **averaged representation difference** for two model $\phi, \tilde{\phi}$ on a distribution $\zeta$ over tasks is

$$\bar{d}_\zeta(\phi, \tilde{\phi}) := \mathop{\mathbb{E}}_{\mathcal{T} \sim \zeta} \left[ \mathcal{L}_{sup}(\mathcal{T}, \phi) - \mathcal{L}_{sup}(\mathcal{T}, \tilde{\phi}) \right] = \mathcal{L}_{sup}(\phi) - \mathcal{L}_{sup}(\tilde{\phi}). \tag{78}$$

The **worst-case representation difference** between representations $\phi, \tilde{\phi}$ on the family of classes $\mathcal{C}_0$ is

$$d_{\mathcal{C}_0}(\phi, \tilde{\phi}) := \sup_{\mathcal{T}_0 \subseteq \mathcal{C}_0} \left| \mathcal{L}_{sup}(\mathcal{T}_0, \phi) - \mathcal{L}_{sup}(\mathcal{T}_0, \tilde{\phi}) \right|. \tag{79}$$

We say the model class $\Phi$ has $\nu$-diversity for $\zeta$ and $\mathcal{C}_0$ if for any $\phi \in \Phi$ and $\phi_\zeta^*$,

$$d_{\mathcal{C}_0}(\phi, \phi_\zeta^*) \leq \bar{d}_\zeta(\phi, \phi_\zeta^*)/\nu. \tag{80}$$

To find the minimum value of $\nu$ in Definition 1, we need further information about $\zeta$. For simplicity, we have a fixed distance assumption, e.g., uniformly sampling from $\{(+1, +1, -1), (+1, -1, +1), (-1, +1, +1)\}$. Then, we consider two different cases below. We consider that all $\mathcal{T}_0 \subseteq \mathcal{C}_0$ such containing $z, z'$ that differ in only 1 entry.

**Assumption 5** (Fixed Distance)**.** *We assume for any multitask finetuning tasks distribution $\zeta$, for any two latent classes $(z, z') \sim \zeta$, we have $z, z'$ differ in $n_k$ entries.*

**Case $L_\mathcal{C} \neq J_0$.** In this case, $J_0 \setminus L_\mathcal{C} \neq \emptyset$, we have the features learned in multitask finetuning that do not cover all features used in the target task. Then, we have the following lemma, which means if $L_\mathcal{C} \neq J_0$ we can have infinitesimal $\nu$ to satisfy the diversity definition.

**Lemma E.3.** *Assume Assumption 3, Assumption 4 and Assumption 5. When $L_\mathcal{C} \neq J_0$, we have $\nu \to 0$.*

*Proof of Lemma E.3.* As features in $\mathcal{C}_0$ not covered by $\mathcal{C}$, we can always find a $\mathcal{T}_0$ such containing $z, z'$ that only differ in entries in $J_0 \setminus L_\mathcal{C}$, we say as entry $\tilde{i}$.

By Lemma E.1, we have $\phi_\zeta^* = U\Lambda^* Q^{-1}$, where $U$ is any orthonormal matrix, $\Lambda^* = \text{diag}(\lambda^*)$. For any $i \in J_\mathcal{C}$, $\lambda_i^* = \frac{R}{\sqrt{k_\mathcal{C}}}$ and $\lambda_i^* = 0$ otherwise. We have $\mathcal{L}_{sup}(\mathcal{T}_0, \phi_\zeta^*) = 0$ and by Equation (74),

$$\mathcal{L}_{sup}(\phi_\zeta^*) = -B \mathop{\mathbb{E}}_{\mathcal{T}_{z,z'} \sim \zeta} \left[ \sqrt{\sum_{i \in J_\mathcal{C}} \lambda_i^{*2} \mathbb{1}[z_i \neq z_i']} \right] \tag{81}$$

$$= -BR\sqrt{\frac{n_k}{k_\mathcal{C}}}. \tag{82}$$

On the other hand, for any $\phi \in \Phi$, we have $\mathcal{L}_{sup}(\mathcal{T}_0, \phi) = -B|\lambda_{\tilde{i}}|$. Thus, we have

$$\begin{aligned}
\nu &= \min_{\phi \in \Phi} \frac{\mathcal{L}_{sup}(\phi) - \mathcal{L}_{sup}(\phi_\zeta^*)}{\left| \mathcal{L}_{sup}(\mathcal{T}_0, \phi) - \mathcal{L}_{sup}(\mathcal{T}_0, \phi_\zeta^*) \right|} \\
&= \min_{\phi \in \Phi} \frac{\mathcal{L}_{sup}(\phi) + BR\sqrt{\frac{n_k}{k_\mathcal{C}}}}{B|\lambda_{\tilde{i}}|} \\
&= \min_{\phi \in \Phi} \frac{-B \mathop{\mathbb{E}}_{\mathcal{T}_{z,z'} \sim \zeta} \left[ \sqrt{\sum_{i \in J_\mathcal{C}} \lambda_i^2 \mathbb{1}[z_i \neq z_i']} \right] + BR\sqrt{\frac{n_k}{k_\mathcal{C}}}}{B|\lambda_{\tilde{i}}|} \\
&\leq \frac{-\mathop{\mathbb{E}}_{\mathcal{T}_{z,z'} \sim \zeta} \left[ \sqrt{\sum_{i \in J_\mathcal{C}} \frac{R^2 - \lambda_{\tilde{i}}^2}{k_\mathcal{C}} \mathbb{1}[z_i \neq z_i']} \right] + R\sqrt{\frac{n_k}{k_\mathcal{C}}}}{|\lambda_{\tilde{i}}|} \\
&= \frac{-\sqrt{\frac{(R^2 - \lambda_{\tilde{i}}^2) n_k}{k_\mathcal{C}}} + R\sqrt{\frac{n_k}{k_\mathcal{C}}}}{|\lambda_{\tilde{i}}|},
\end{aligned} \tag{83}$$

where the first inequality is by constructing a specific $\phi$. Note that Equation (83) $\to 0$ when $|\lambda_{\tilde{i}}| \to 0$. $\phi$ is constructed as: for any $i \in J_\mathcal{C}$, $\lambda_i = \sqrt{\frac{R^2 - \lambda_{\tilde{i}}^2}{k_\mathcal{C}}}$ and $|\lambda_{\tilde{i}}| \to 0$. Thus, we finish the proof. $\qquad \square$

**Case $L_\mathcal{C} = J_0$.** In this case $J_0 \setminus L_\mathcal{C} = \emptyset$, we have all features in $\mathcal{C}_0$ covered by $\mathcal{C}$.

**Lemma E.4.** *Assume Assumption 3, Assumption 4 and Assumption 5. When all $\mathcal{T}_0 \subseteq \mathcal{C}_0$ such containing $z, z'$ that differ in only 1 entry and $L_\mathcal{C} = J_0$, we have $\nu$ is lower bounded by some constant $\tilde{c} = \sqrt{n_k}\left(1 - \sqrt{\frac{1}{k_\mathcal{C}(k_\mathcal{C}-1)}}\left(\sqrt{n_k(n_k-1)} + k_\mathcal{C} - n_k\right)\right)$.*

*Proof of Lemma E.4.* We say the differ entry in $\mathcal{T}_0$ as entry $\tilde{i}$. By Lemma E.1, we have $\phi_\zeta^* = U\Lambda^* Q^{-1}$, where $U$ is any orthonormal matrix, $\Lambda^* = \text{diag}(\lambda^*)$. For any $i \in J_\mathcal{C}$, $\lambda_i^* = \frac{R}{\sqrt{k_\mathcal{C}}}$ and $\lambda_i^* = 0$ otherwise. By Equation (74), we have $\mathcal{L}_{sup}(\mathcal{T}_0, \phi_\zeta^*) = -BR\sqrt{\frac{n_0}{k_\mathcal{C}}}$ and $\mathcal{L}_{sup}(\phi_\zeta^*) = -BR\sqrt{\frac{n_k}{k_\mathcal{C}}}$.

On the other hand, for any $\phi \in \Phi$, we have $\mathcal{L}_{sup}(\mathcal{T}_0, \phi) = -B|\lambda_{\tilde{i}}|$. Thus, by Assumption 3, we have

$$
\begin{aligned}
\nu &= \min_{\mathcal{T}_0 \subseteq \mathcal{C}_0, \phi \in \Phi} \frac{\mathcal{L}_{sup}(\phi) - \mathcal{L}_{sup}(\phi_\zeta^*)}{\left| \mathcal{L}_{sup}(\mathcal{T}_0, \phi) - \mathcal{L}_{sup}(\mathcal{T}_0, \phi_\zeta^*) \right|} \\
&= \min_{\mathcal{T}_0 \subseteq \mathcal{C}_0, \phi \in \Phi} \frac{\mathcal{L}_{sup}(\phi) + BR\sqrt{\frac{n_k}{k_\mathcal{C}}}}{\left| -B|\lambda_{\tilde{i}}| + BR\sqrt{\frac{1}{k_\mathcal{C}}} \right|} \\
&= \min_{\mathcal{T}_0 \subseteq \mathcal{C}_0, \phi \in \Phi} \frac{-B \underset{\mathcal{T}_{z,z'} \sim \zeta}{\mathbb{E}} \left[ \sqrt{\sum_{i \in J_\mathcal{C}} \lambda_i^2 \mathbb{1}[z_i \neq z_i']} \right] + BR\sqrt{\frac{n_k}{k_\mathcal{C}}}}{\left| -B|\lambda_{\tilde{i}}| + BR\sqrt{\frac{1}{k_\mathcal{C}}} \right|} \\
&= \min_{\mathcal{T}_0 \subseteq \mathcal{C}_0, \phi \in \Phi} \frac{- \underset{\mathcal{T}_{z,z'} \sim \zeta}{\mathbb{E}} \left[ \sqrt{\lambda_{\tilde{i}}^2 \mathbb{1}[z_{\tilde{i}} \neq z_{\tilde{i}}'] + \sum_{i \in J_\mathcal{C} \setminus \{\tilde{i}\}} \frac{R^2 - \lambda_{\tilde{i}}^2}{k_\mathcal{C} - 1} \mathbb{1}[z_i \neq z_i']} \right] + R\sqrt{\frac{n_k}{k_\mathcal{C}}}}{\left| -|\lambda_{\tilde{i}}| + R\sqrt{\frac{1}{k_\mathcal{C}}} \right|} \\
&= \min_{\mathcal{T}_0 \subseteq \mathcal{C}_0, \phi \in \Phi} \frac{- \left[ \frac{n_k}{k_\mathcal{C}} \sqrt{\lambda_{\tilde{i}}^2 + \frac{R^2 - \lambda_{\tilde{i}}^2}{k_\mathcal{C} - 1}(n_k - 1)} + \frac{k_\mathcal{C} - n_k}{k_\mathcal{C}} \sqrt{\frac{n_k(R^2 - \lambda_{\tilde{i}}^2)}{k_\mathcal{C} - 1}} \right] + R\sqrt{\frac{n_k}{k_\mathcal{C}}}}{\left| -|\lambda_{\tilde{i}}| + R\sqrt{\frac{1}{k_\mathcal{C}}} \right|} \\
&= \sqrt{n_k} \left( 1 - \sqrt{\frac{1}{k_\mathcal{C}(k_\mathcal{C} - 1)}} \left( \sqrt{n_k(n_k - 1)} + k_\mathcal{C} - n_k \right) \right), \quad (84)
\end{aligned}
$$

where the last equality take $\lambda_{\tilde{i}} = 0$. $\qquad \square$

### E.3 Proof of Main Results

*Proof of Theorem 3.3.* Note that $R = B = n_0 = k_0 = 1, n_k = 2$.

We see that $\zeta$ satisfies Assumption 3, Assumption 4 and Assumption 5. We finish the proof by Lemma E.2, Lemma E.3 and Lemma E.4 with some simple calculations. $\qquad \square$

Thus, we can link our diversity and consistency parameters to the number of features in $z$ encoded by training tasks or target tasks. Based on this intuition, we propose a selection algorithm, where selection is based on $x$, we want to select data that encodes more relevant features of $z$, this can be achieved by comparing $x$ from target data and training data either using cosine similarity or KDE.

## F Vision Experimental Results

We first provide a summary of dataset and protocal we use, we provide details in following sections.

**Datasets and Models.** We use four widely used few-shot learning benchmarks: miniImageNet (Vinyals et al., 2016), tieredImageNet (Ren et al., 2018), DomainNet (Peng et al., 2019) and Meta-dataset (Triantafillou et al., 2020), following the protocol in Chen et al. (2021b); Tian et al. (2020b). We use exemplary foundation models with different pretraining schemes (MoCo-v3 (Chen et al., 2021a), DINO-v2 (Oquab et al., 2023), and supervised learning with ImageNet (Russakovsky et al., 2015)) and architectures (ResNet (He et al., 2016) and ViT (Dosovitskiy et al., 2021)).

**Experiment Protocol.** We consider few-shot tasks consisting of $N$ classes with $K$ support samples and $Q$ query samples per class (known as $N$-way $K$-shot). The goal is to classify the query samples into the $N$ classes based on the support samples. Tasks used for finetuning are constructed by samples from the training split. Each task is formed randomly by sampling 15 classes, with every class drawing 1 or 5 support samples and 10 query samples. Target tasks are similarly constructed, yet from the test set. We follow (Chen et al., 2021b) for multitask finetuning and target task adaptation. During multitask finetuning, we update all parameters in the model using a nearest centroid classifier,

in which all samples are encoded, class centroids are computed, and cosine similarity between a query sample and those centroids are treated as the class logits. For adaptation to a target task, we only retain the model encoder and consider a similar nearest centroid classifier. This experiment protocol applies to all three major experiments (Sections 4.1 to 4.3).

## F.1 DATASETS

The miniImageNet dataset is a common benchmark for few-shot learning. It contains 100 classes sampled from ImageNet, then is randomly split into 64, 16, and 20 classes as training, validation, and testing set respectively.

The tieredImageNet dataset is another widely used benchmark for few-shot learning. It contains 608 classes from 34 super-categories sampled from ImageNet. These categories are then subdivided into 20 training categories with 351 classes, 6 validation categories with 97 classes, and 8 testing categories with 160 classes

DomainNet is the largest domain adaptation benchmark with about 0.6 million images. It consists of around 0.6 million images of 345 categories from 6 domains: clipart (clp), infograph (inf), quickdraw (qdr), real (rel) and sketch (skt). We split it into 185, 65, 100 classes as training, validation, and testing set respectively. We conduct experiments on Sketch (skt) subsets.

Meta-Dataset encompasses ten publicly available image datasets covering a wide array of domains: ImageNet-1k, Omniglot, FGVC-Aircraft, CUB-200-2011, Describable Textures, QuickDraw, FGVCx Fungi, VGG Flower, Traffic Signs, and MSCOCO. Each of these datasets is split into training, validation, and testing subsets. For additional information on the Meta-Dataset can be found in Appendix 3 of Triantafillou et al. (2020).

## F.2 EXPERIMENT PROTOCOLS

Our evaluation and the finetuning process take the form of few-shot tasks, where a target task consists of $N$ classes with $K$ support samples and $Q$ query samples in each class. The objective is to classify the query samples into the $N$ classes based on the support samples. To accomplish this, we take the support samples in each class and feed them through an image encoder to obtain representations for each sample. We then calculate the average of these representations within each class to obtain the centroid of each class. For a given query sample $x$, we compute the probability that $x$ belongs to class $y$ based on the cosine similarity between the representation of $x$ and the centroid of class $y$.

In our testing stage, we constructed 1500 target tasks, each consisting of 15 classes randomly sampled from the test split of the dataset. Within each class, we randomly selected 1 or 5 of the available images as shot images and 15 images as query images. These tasks are commonly referred to as 1-shot or 5-shot tasks. We evaluated the performance of our model on these tasks and reported the average accuracy along with a 95% confidence interval.

During multitask finetuning, the image encoder is directly optimized on few-shot classification tasks. To achieve this, we construct multitasks in the same format as the target tasks and optimize from the same evaluation protocol. Specifically, we create a total of 200 finetuning tasks, each task consists of 15 classes sampled from the train split of data, where each class contains 1 support image and 9 query images, resulting in 150 images per task. The classes in a finetuning task are sampled from the train split of the data.

To ensure a fair comparison with the finetuning baseline, we used the same training and testing data, as well as batch size, and applied standard finetuning. During standard finetuning, we added a linear layer after the encoder and trained the model. We also utilized the linear probing then finetuning (LP-FT) technique proposed by Kumar et al. (2022), which has been shown to outperform finetuning alone on both in-distribution and out-of-distribution data. In the testing stage, we removed the linear layer and applied the same few-shot testing pipeline to the finetuned encoders.

For task selection, we employ the CLIP ViT-B image encoder to obtain image embeddings. We assess consistency by measuring the cosine similarity of the mean embeddings and we evaluate diversity through a coverage score derived from the ellipsoid formula outlined in Section 3.2.

For optimization, we use the SGD optimizer with momentum 0.9, the learning rate is 1e-5 for CLIP and moco v3 pretrained models, and is 2e-6 for DINO v2 pretrained models. The models were finetuned over varying numbers of epochs in each scenario until they reached convergence.

## F.3    EXISTENCE OF TASK DIVERSITY

Task diversity is crucial for the foundation model to perform well on novel classes in target tasks.

In this section, we prove for task satisfying consistency, greater diversity in the related data can help reduce the error on the target task. Specifically, for the target task, where the target tasks data originates from the test split of a specific dataset, we utilized the train split of the same dataset as the finetuning tasks data. Then finetuning tasks satisfied consistency. In experiments, we varied the number of classes accessible to the model during the finetuning stage, while keeping the total sample number the same. This serves as a measure of the diversity of training tasks.

### F.3.1    MINIIMAGENET AND OMNIGLOT

We show the results of CLIP encoder on miniImageNet and Omniglot. We vary the number of classes model access to in finetuning stage. The number of classes varies from all classes, i.e., 64 classes, to 8 classes. Each task contains 5 classes. For finetuning tasks, each class contains 1 shot image and 10 query images. For target tasks, each class contains the 1-shot image and 15 query images.

| # limited classes | 64 | 32 | 16 | 8 | 0 |
|---|---|---|---|---|---|
| Accuracy | $90.02 \pm 0.15$ | $88.54 \pm 1.11$ | $87.94 \pm 0.22$ | $87.07 \pm 0.20$ | $83.03 \pm 0.24$ |

Table 3: Class diversity on ViT-B32 backbone on miniImageNet.

Table 3 shows the accuracy of ViT-B32 across different numbers of classes during the finetuning stage. The "Class 0" represents direct evaluation without any finetuning. We observe that finetuning the model leads to an average accuracy improvement of 4%. Furthermore, as the diversity of classes increases, we observe a corresponding increase in performance. This indicates that incorporating a wider range of classes during the finetuning process enhances the model's overall accuracy.

For task diversity, we also use dataset Omniglot (Lake et al., 2015). The Omniglot dataset is designed to develop more human-like learning algorithms. It contains 1623 different handwritten characters from 50 different alphabets. The 1623 classes are divided into 964, 301, and 358 classes as training, validation, and testing sets respectively. We sample multitask in finetuning stage from training data and the target task from testing data.

| # limited classes | 964 | 482 | 241 | 50 | 10 | 0 |
|---|---|---|---|---|---|---|
| Accuracy | $95.35 \pm 0.14$ | $95.08 \pm 0.14$ | $94.29 \pm 0.15$ | $88.48 \pm 0.20$ | $80.26 \pm 0.24$ | $74.69 \pm 0.26$ |

Table 4: Class diversity on ViT-B32 backbone on Omniglot.

Table 4 shows the accuracy of ViT-B32 on different numbers of classes in finetuning stage, where class 0 indicates direct evaluation without finetuning. Finetuning improves the average accuracy by 5.5%. As class diversity increases, performance increases.

### F.3.2    TIEREDIMAGENET

We then show results on tieredImageNet across learning settings for the ViT-B backbone. We follow the same setting where we restrain each task that contains 15 classes.

We found that using more classes from related data sources during finetuning improves accuracy. This result indicates that upon maintaining consistency, a trend is observed where increased diversity leads to an enhancement in performance.

| Pretrained | 351 | 175 | 43 | 10 |
|---|---|---|---|---|
| DINOv2 | 84.74 | 82.75 | 82.60 | 82.16 |
| CLIP | 68.57 | 67.70 | 67.06 | 63.52 |
| Supervised | 89.97 | 89.69 | 89.19 | 88.92 |

Table 5: The performance of the ViT-B backbone using different pretraining methods on tieredImagenet, varying the number of classes accessible to the model during the finetuning stage. Each column represents the number of classes within the training data.

## F.4   ABLATION STUDY

In Section 4 and the result in Table 2, we utilize the train split from the same dataset to construct the finetuning data. It is expected that the finetuning data possess a diversity and consistency property, encompassing characteristics that align with the test data while also focusing on its specific aspects.

In the following ablation study, we explore the relationship between the diversity and consistency of data in finetuning tasks, sample complexity, and finetuning methods. We seek to answer the following questions: Does multitask finetuning benefit only from certain aspects? How do these elements interact with each other?

### F.4.1   VIOLATE BOTH CONSISTENCY AND DIVERSITY: ALTERING FINETUNING TASK DATA WITH INVARIANT SAMPLE COMPLEXITY

In this portion, we examine the performance when the model is finetuned using data completely unrelated to the target task data. With the same finetuning sample complexity, the performance cannot be compared to the accuracy we have currently attained.

In this section, we present the performance of MoCo v3 with a ViT-B backbone on the DomainNet dataset. We finetuned the model using either ImageNet data or DomainNet train-split data and evaluated its performance on the test-split of DomainNet. We observed that finetuning the model with data selected from the DomainNet train-split resulted in improved performance on the target task. This finding aligns with our expectations and highlights the significance of proper finetuning data selection.

When considering the results presented in Table 6, we also noticed that for MoCo v3 with a ResNet50 backbone and DINO v2 with a ViT-S backbone, multitask finetuning on ImageNet led to a decrease in model performance compared to direct adaptation. This suggests that inappropriate data selection can have a detrimental effect on the final model performance. This conclusion is also supported by the findings of Kumar et al. (2022).

### F.4.2   VIOLATING CONSISTENCY WHILE RETAINING DIVERSITY: THE TRADE-OFF BETWEEN TASK CONSISTENCY AND SAMPLE COMPLEXITY

Finetuning tasks with superior data are expected to excel under identical complexity, a natural question can be proposed: Does additional data enhance performance? Our results in this section negate this question. Testing the model on the DomainNet test-split, we employ two settings. In the first setting, we finetune the model on the DomainNet train-split. In the second, the model is finetuned with a combination of the same data from DomainNet as in the first setting, along with additional data from ImageNet.

Within our theoretical framework, mixing data satisfies diversity but fails consistency. The finetuning data, although containing related information, also encompasses excessive unrelated data. This influx of unrelated data results in a larger consistency parameter $\kappa$ in our theoretical framework, adversely impacting model performance on the target task. We offer empirical evidence to affirm our theoretical conclusion.

Table 7 shows mixed data of domainNet and ImageNet will doesn't provide the same advantages as using only DomainNet data. In this case, an increasing in data does not necessarily mean better performance.

| pretrained | backbone | FT data | Accuracy |
|---|---|---|---|
| MoCo v3 | ViT-B | ImageNet | 24.88 (0.25) |
| | | DomainNet | 32.88 (0.29) |
| | ResNet50 | ImageNet | 27.22 (0.27) |
| | | DomainNet | 33.53 (0.30) |
| DINO v2 | ViT-S | ImageNet | 51.69 (0.39) |
| | | DomainNet | 61.57 (0.40) |
| | ViT-B | ImageNet | 62.32 (0.40) |
| | | DomainNet | 68.22 (0.40) |
| Supervised | ViT-B | ImageNet | 31.16 (0.31) |
| | | DomainNet | 48.02 (0.38) |
| | ResNet50 | ImageNet | 29.56 (0.28) |
| | | DomainNet | 39.09 (0.34) |

Table 6: Finetuning data selection on model performance. FT data: dataset we select for multitask finetuning. Report the accuracy on the test-split of DomainNet.

| Pretrained | DomainNet | DomainNet + ImageNet |
|---|---|---|
| DINOv2 | 68.22 | 66.93 |
| CLIP | 64.97 | 63.48 |
| Supervised | 48.02 | 43.76 |

Table 7: Results evaluating on DomainNet test-split using ViT-B backbone. First column shows performance where model finetune on data from DomainNet train-split alone, second column shows the performance of the model finetuned using a blend of the same data from DomainNet, combined with additional data from ImageNet.

### F.4.3 DIVERSITY AND CONSISTENCY OF TASK DATA AND FINETUNING METHODS

To provide a more comprehensive understanding of the impact of task data and finetuning methods on model performance, we conduct additional experiments, utilizing varying finetuning methods and data. The model is tested on the DomainNet test split. We employ either multitask finetuning or standard finetuning, where a linear layer is added after the pretrained model. This linear layer maps the representations learned by encoders to the logits. The data of finetuning tasks derive from either the DomainNet train-split or ImageNet.

| | 1 | 2 | 3 | 4 | 5 |
| Pretrained | Adaptation | ImageNet (SFT) | ImageNet (Ours) | DomainNet (SFT) | DomainNet (Ours) |
|---|---|---|---|---|---|
| DINOv2 | 61.65 | 59.80 | 62.32 | 61.84 | 68.22 |
| CLIP | 46.39 | 46.50 | 58.94 | 47.72 | 64.97 |
| Supervised | 28.70 | 28.52 | 31.16 | 30.93 | 48.02 |

Table 8: Results evaluating on DomainNet test-split using ViT-B backbone. Adaptation: Direction adaptation without finetuning; SFT: Standard finetuning; Ours: Our multitask finetuning. Col-1 shows performance without any finetuning, Col-2,3,4,5 shows performance with different finetuning methods and data.

In Table 8, we detail how data quality and finetuning methods of tasks impact the ultimate performance. Standard finetuning (SFT) with unrelated data diminishes performance compared to direct adaptation (col-1 vs col-2). On the other hand, multitask finetuning using unrelated data (ImageNet), or SFT with related data (DomainNet), both outperform direct adaptation. However, multitask finetuning

| Pretrained | Selection | INet | Omglot | Acraft | CUB | QDraw | Fungi | Flower | Sign | COCO |
|---|---|---|---|---|---|---|---|---|---|---|
| CLIP | All | 60.87 | 70.53 | 31.67 | 66.98 | 40.28 | 34.88 | 80.80 | 37.82 | 33.71 |
| | Selected | 60.87 | **77.93** | **32.02** | **69.15** | **42.36** | **36.66** | **80.92** | **38.46** | **37.21** |
| DINOv2 | All | 83.04 | 72.92 | 36.52 | 94.01 | 49.65 | 52.72 | 98.54 | 34.59 | 47.05 |
| | Selected | 83.04 | **80.29** | **36.91** | **94.12** | **52.21** | **53.31** | **98.65** | **36.62** | **50.09** |
| MoCo v3 | All | 59.62 | 60.85 | 18.72 | 40.49 | 40.96 | 32.65 | 59.60 | 33.94 | 33.42 |
| | Selected | 59.62 | **63.08** | **19.03** | **40.74** | **41.16** | **32.89** | **59.64** | **35.25** | **33.51** |

Table 9: Results evaluating our task selection algorithm on Meta-dataset using ViT-B backbone.

with unrelated data proves more beneficial than the latter (col-3 vs col-4). The peak performance is attained through multitask finetuning on related data (col-5).

### F.4.4   ABLATION STUDY ON TASK SELECTION ALGORITHM

We show a simplified diagram for task selection in Figure 2.

We first provide some details of Table 1. We first create an array of finetuning tasks, and then apply our task selection algorithm to these tasks. Specifically, we design 100 finetuning tasks by randomly selecting 15 classes, each providing 1 support sample and 10 query samples. The target tasks remain consistent with those discussed in Section 4. For a more comprehensive analysis of our algorithm, we performed ablation studies on the task selection algorithm, concentrating solely on either consistency or diversity, while violating the other. **Violate Diversity**: If the algorithm terminates early without fulfilling the stopping criteria, the data utilized in finetuning tasks fails to encompass all the attributes present in the target data. This leads to a breach of the diversity principle. **Violate Consistency**: Conversely, if the algorithm persists beyond the stopping criteria, the finetuning tasks become overly inclusive, incorporating an excessive amount of unrelated data, thus breaching the consistency.

This section details an ablation study on task selection for the dataset, we implement our task selection process on a meta-dataset, treating each dataset as a distinct task and choosing datasets to serve as data sources for the finetuning tasks. We show the result in Table 9.

Table 9 indicates that maintaining both consistency and diversity in the task selection algorithm is essential for optimal performance. This is evident from the comparison between the Random selection and the our approach, where the latter often shows improved performance across multiple datasets. ImageNet as the target task is an exception where the two approaches give the best results. Due to its extensive diversity, all samples from all other datasets are beneficial for finetuning. Consequently, the task selection algorithm tends to select all the candidate tasks.

### F.5   TASK SELECTION ALGORITHM ON DOMAINNET

We verify our task selection algorithm by applying it on DomainNet. Here, the mini-ImageNet test-split is regarded as the target task source, and diverse domains (such as clipart (clp), infograph (inf), quickdraw (qdr), real (rel), and sketch (skt)) are considered as sources for finetuning tasks. We view different domain datasets as distinct finetuning tasks. With 6 domains in focus, our objective is to select a subset that optimizes model performance. We systematically apply Algorithm 1. Initially, we calculate the cosine similarity of mean embeddings between each domain and target tasks, ordering them from most to least similar: real, painting, sketch, clipart, infograph, and quickdraw. Sequentially adding datasets in this order, the process continues until the diversity score (1 over Mahalanobis distance) stops exhibiting significant increase.

As we can see in Figure 4, the diversity does not increase when we just select *real* and *painting* as our finetuning task data. For a comprehensive analysis, each combination is finetuned and the model performance accuracy on the target task is displayed.

As we can see in Figure 5, the accuracy aligns with the conclusions drawn based on consistency and diversity. Remarkably, only *real* and *painting* suffice for the model to excel on the target task.

### F.6   MORE RESULTS WITH CLIP ENCODER

In this section, we show additional results on CLIP (Radford et al., 2021) model.

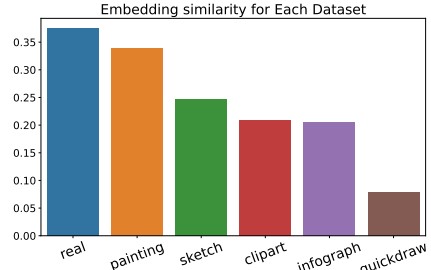

(a) Mean embedding similarity for each data sort from most similar to least similar.

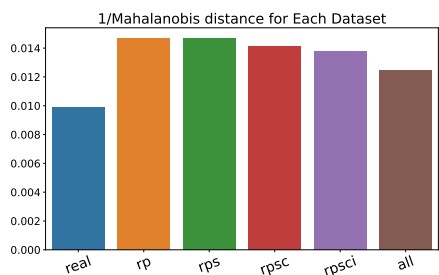

(b) Diversity score when adding task one by one, where *rp*: *real* and *painting*; *rps*: *real* and *painting* and *sketch* and so on.

Figure 4: Dataset selection based on consistency and diversity on domainNet. Figure 4a shows the consistency. Figure 4b shows the diversity.

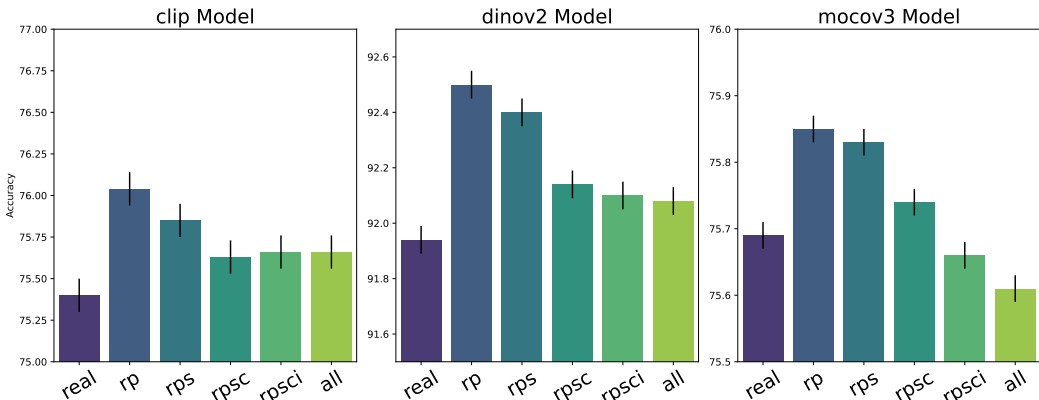

Figure 5: Finetuning with different selection of domain datasets, where *rp*: *real* and *painting*; *rps*: *real* and *painting* and *sketch* and so on.

We can observe from Table 10 standard finetuning improves performance compared to direct adaptation. However, our proposed multitask finetuning approach consistently achieves even better results than the standard baseline.

**Task ($M$) vs Sample ($m$).** We vary the task size and sample size per task during finetuning. We verify the trend of different numbers of tasks and numbers of images per task. Each task contains 5 classes. For finetuning tasks, $m = 50$ indicates each class contains the 1-shot image and 9-query images. $m = 100$ indicates each class contains 2-shot and 18-query images. $m = 200$ indicates each class contains 4-shot and 36-query images. $M = m = 0$ indicates direct evaluation without finetuning. For target tasks, each class contains the 1-shot image and 15 query images.

Table 11 shows the results on the pretrained CLIP model using the ViT backbone. For direct adaptation without finetuning, the model achieves 83.03% accuracy. Multitask finetuning improves the average accuracy at least by 6%. For a fixed number of tasks or samples per task, increasing samples or tasks improves accuracy. These results suggest that the total number of samples ($M \times m$) will determine the overall performance, supporting our main theorem.

**Few-shot Effect.** We perform experiments on the few-shot effects of finetuning tasks. We aim to evaluate whether increasing the number of few-shot images in the finetuning task leads to significant improvements. Each finetuning task consists of 5 classes, and we maintain a fixed number of 10 query images per class while gradually increasing the number of shot images, as illustrated in Table 12. As for the target tasks, we ensure each class contains 1 shot image and 15 query images for evaluation.

| backbone | method | miniImageNet | | tieredImageNet | | DomainNet | |
|---|---|---|---|---|---|---|---|
| | | 1-shot | 5-shot | 1-shot | 5-shot | 1-shot | 5-shot |
| CLIP-ViTB32 | Direct Adaptation | 68.41 (0.54) | 87.43 (0.15) | 59.55 (0.21) | 79.51 (0.27) | 46.48 (0.37) | 72.01 (0.29) |
| | Standard FT | 69.39 (0.30) | 88.39 (0.15) | 61.20 (0.37) | 80.65 (0.27) | 47.72 (0.37) | 72.82 (0.29) |
| | Multitask FT (Ours) | **78.62** (0.15) | **93.22** (0.11) | **68.57** (0.37) | **84.79** (0.22) | **64.97** (0.39) | **80.49** (0.25) |
| CLIP-ResNet50 | Direct Adaptation | 61.31 (0.31) | 82.03 (0.18) | 51.76 (0.36) | 71.40 (0.30) | 40.55 (0.36) | 64.90 (0.31) |
| | Standard FT | 63.15 (0.31) | 83.45 (0.17) | 55.77 (0.35) | 75.28 (0.29) | 43.77 (0.38) | 67.30 (0.31) |
| | Multitask FT (Ours) | **67.03** (0.30) | **85.09** (0.17) | **57.56** (0.36) | **75.80** (0.28) | **52.67** (0.39) | **72.19** (0.30) |

Table 10: **Comparison on 15-way classification.** Average few-shot classification accuracies (%) with 95% confidence intervals clip encoder.

| Sample (m) / Task (M) | 0 | 50 | 100 | 200 |
|---|---|---|---|---|
| 0 | $83.03 \pm 0.24$ | | | |
| 200 | | $89.07 \pm 0.20$ | $89.95 \pm 0.19$ | $\mathbf{90.09 \pm 0.19}$ |
| 400 | | $89.31 \pm 0.19$ | $\mathbf{90.11 \pm 0.19}$ | $90.70 \pm 0.18$ |
| 800 | | $\mathbf{89.71 \pm 0.19}$ | $90.27 \pm 0.19$ | $90.80 \pm 0.18$ |

Table 11: Accuracy with a varying number of tasks and samples (ViT-B32 backbone).

Table 12 displays the accuracy results of ViT-B32 when varying the number of few-shot images in the finetuning tasks. We observe that increasing the number of few-shot images, thereby augmenting the sample size within each task, leads to improved performance. This finding is quite surprising, considering that the finetuning tasks and target tasks have different numbers of shot images. However, this aligns with our understanding of sample complexity, indicating that having access to more training examples can enhance the model's ability to generalize and perform better on unseen data.

### F.7  SAMPLE COMPLEXITY ON PERFORMANCE FOR TIEREDIMAGENET

We provide a table and visualization of the trend of the number of tasks and the number of samples per task for the MoCo v3 ViT model on tieredImageNet in Table 13 and Figure 6.

As demonstrated in the paper, we have observed that increasing the number of tasks generally leads to performance improvements, while keeping the number of samples per task constant. Conversely, when the number of samples per task is increased while maintaining the same number of tasks, performance also tends to improve. These findings emphasize the positive relationship between the number of tasks and performance, as well as the influence of sample size within each task.

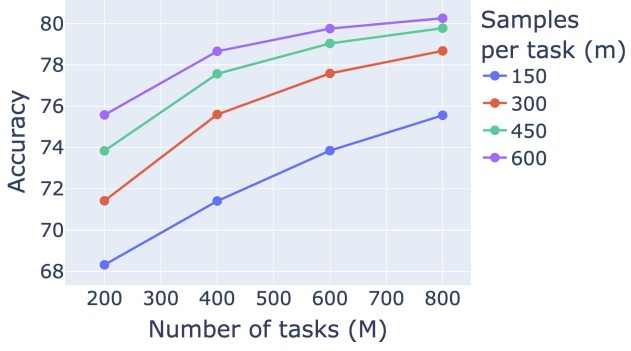

Figure 6: Finetuning using tieredImageNet train-split, test on test-split.

### F.8  FULL RESULTS FOR EFFECTIVENESS OF MULTITASK FINETUNING

In this section, we provide another baseline in complement to the results in Section 4.3.

| # shot images | 20 | 10 | 5 | 1 | 0 |
|---|---|---|---|---|---|
| **Accuracy** | $91.03 \pm 0.18$ | $90.93 \pm 0.18$ | $90.54 \pm 0.18$ | $90.02 \pm 0.15$ | $83.03 \pm 0.24$ |

Table 12: Few-shot effect on ViT-B32 backbone on miniImageNet.

| Task (M) \ Sample (m) | 150 | 300 | 450 | 600 |
|---|---|---|---|---|
| 200 | 68.32 (0.35) | 71.42 (0.35) | 73.84 (0.35) | 75.58 (0.35) |
| 400 | 71.41 (0.35) | 75.60 (0.35) | 77.57 (0.34) | 78.66 (0.34) |
| 600 | 73.85 (0.35) | 77.59 (0.34) | 79.04 (0.33) | 79.76 (0.33) |
| 800 | 75.56 (0.35) | 78.68 (0.34) | 79.78 (0.33) | 80.26 (0.33) |

Table 13: Accuracy with a varying number of tasks and samples (ViT-B32 backbone).

We incorporated the Model-Agnostic Meta-Learning (MAML) algorithm, as outlined by Finn et al. (2017), as another baseline for our few-shot tasks. MAML operates in a two-step process: it initially updates parameters based on within-episode loss (the inner loop), then it evaluates and updates loss based on learned parameters (the outer loop). We follow the pipeline in Triantafillou et al. (2020) to implement MAML for few-shot tasks. We show results in Table 14.

Table 14 reveals that MAML exhibits variable performance across different settings. For instance, it outperforms both Adaptation and Standard FT methods in scenarios like MoCo v3 ViT-B on miniImageNet, DomainNet, and ResNet 50 on supervised training for tieredImageNet. However, its performance is less impressive in other contexts, such as DINOv2 ViT-B on miniImageNet and ViT-B on supervised training for miniImageNet. This variability in performance is attributed to the constraints of our few-shot tasks, where the limited number of support samples restricts the model's capacity to adapt to new tasks. Despite these fluctuations, our multitask finetuning approach consistently surpasses the mentioned baselines, often by a significant margin, across all evaluated scenarios.

| pretrained | backbone | method | miniImageNet | | tieredImageNet | | DomainNet | |
|---|---|---|---|---|---|---|---|---|
| | | | 1-shot | 5-shot | 1-shot | 5-shot | 1-shot | 5-shot |
| MoCo v3 | ViT-B | Adaptation | 75.33 (0.30) | 92.78 (0.10) | 62.17 (0.36) | 83.42 (0.23) | 24.84 (0.25) | 44.32 (0.29) |
| | | Standard FT | 75.38 (0.30) | 92.80 (0.10) | 62.28 (0.36) | 83.49 (0.23) | 25.10 (0.25) | 44.76 (0.27) |
| | | MAML | 79.26 (0.28) | 93.02 (0.08) | 67.96 (0.32) | 84.66 (0.19) | 28.91 (0.39) | 51.12 (0.28) |
| | | Ours | **80.62** (0.26) | **93.89** (0.09) | **68.32** (0.35) | **85.49** (0.22) | **32.88** (0.29) | **54.17** (0.30) |
| | ResNet50 | Adaptation | 68.80 (0.30) | 88.23 (0.13) | 55.15 (0.34) | 76.00 (0.26) | 27.34 (0.27) | 47.50 (0.28) |
| | | Standard FT | 68.85 (0.30) | 88.23 (0.13) | 55.23 (0.34) | 76.07 (0.26) | 27.43 (0.27) | 47.65 (0.28) |
| | | MAML | 69.28 (0.26) | 88.78 (0.12) | 55.31 (0.32) | 75.51 (0.19) | 27.53 (0.39) | 47.73 (0.28) |
| | | Ours | **71.16** (0.29) | **89.31** (0.12) | **58.51** (0.35) | **78.41** (0.25) | **33.53** (0.30) | **55.82** (0.29) |
| DINO v2 | ViT-S | Adaptation | 85.90 (0.22) | 95.58 (0.08) | 74.54 (0.32) | 89.20 (0.19) | 52.28 (0.39) | 72.98 (0.28) |
| | | Standard FT | 86.75 (0.22) | 95.76 (0.08) | 74.84 (0.32) | 89.30 (0.19) | 54.48 (0.39) | 74.50 (0.28) |
| | | MAML | 86.67 (0.24) | 95.54 (0.08) | 74.63 (0.34) | 89.60 (0.19) | 52.72 (0.34) | 73.35 (0.28) |
| | | Ours | **88.70** (0.22) | **96.08** (0.08) | **77.78** (0.32) | **90.23** (0.18) | **61.57** (0.40) | **77.97** (0.27) |
| | ViT-B | Adaptation | 90.61 (0.19) | 97.20 (0.06) | 82.33 (0.30) | 92.90 (0.16) | 61.65 (0.41) | 79.34 (0.25) |
| | | Standard FT | 91.07 (0.19) | 97.32 (0.06) | 82.40 (0.30) | 93.07 (0.16) | 61.84 (0.39) | 79.63 (0.25) |
| | | MAML | 90.77 (0.18) | 97.20 (0.08) | 82.54 (0.32) | 92.88 (0.19) | 62.30 (0.39) | 79.01 (0.28) |
| | | Ours | **92.77** (0.18) | **97.68** (0.06) | **84.74** (0.30) | **93.65** (0.16) | **68.22** (0.40) | **82.62** (0.24) |
| Supervised pretraining on ImageNet | ViT-B | Adaptation | 94.06 (0.15) | 97.88 (0.05) | 83.82 (0.29) | 93.65 (0.13) | 28.70 (0.29) | 49.70 (0.28) |
| | | Standard FT | 95.28 (0.13) | 98.33 (0.04) | 86.44 (0.27) | 94.91 (0.12) | 30.93 (0.31) | 52.14 (0.29) |
| | | MAML | 95.35 (0.12) | 98.50 (0.08) | 86.79 (0.32) | 94.72 (0.19) | 30.53 (0.39) | 52.21 (0.28) |
| | | Ours | **96.91** (0.11) | **98.76** (0.04) | **89.97** (0.25) | **95.84** (0.11) | **48.02** (0.38) | **67.25** (0.29) |
| | ResNet50 | Adaptation | 81.74 (0.24) | 94.08 (0.09) | 65.98 (0.34) | 84.14 (0.21) | 27.32 (0.27) | 46.67 (0.28) |
| | | Standard FT | 84.10 (0.22) | 94.81 (0.09) | 74.48 (0.33) | 88.35 (0.19) | 34.10 (0.31) | 55.08 (0.29) |
| | | MAML | 82.07 (0.28) | 94.12 (0.08) | 75.69 (0.32) | 89.30 (0.19) | 35.10 (0.39) | 56.51 (0.28) |
| | | Ours | **87.61** (0.20) | **95.92** (0.07) | **77.74** (0.32) | **89.77** (0.17) | **39.09** (0.34) | **60.60** (0.29) |

Table 14: **Results of few-shot image classification.** We report average classification accuracy (%) with 95% confidence intervals on test splits. Adaptation: Direction adaptation without finetuning; Standard FT: Standard finetuning; MAML: MAML algorithm in Finn et al. (2017); Ours: Our multitask finetuning; 1-/5-shot: number of labeled images per class in the target task.

# G   NLP EXPERIMENTAL RESULTS

We first provide a summary of the experimental setting and results in the below subsection. Then we provide details in the following subsections.

## G.1   SUMMARY

To further validate our approach, we conducted prompt-based finetuning experiments on masked language models, following the procedure outlined in Gao et al. (2021a).

**Datasets and Models.**    We consider a collection of 14 NLP datasets, covering 8 single-sentence and 6 sentence-pair English tasks. This collection includes tasks from the GLUE benchmark (Wang et al., 2018), as well as 7 other popular sentence classification tasks. The objective is to predict the label based on a single sentence or a sentence-pair. Specifically, the goal is to predict sentiments for single sentences or to estimate the relationship between sentence pairs. Each of the datasets is split into training and test set. See details in Appendix G.2. We experiment with a pretrained model RoBERTa (Liu et al., 2019).

**Experiment Protocols.**    We consider prompt-based finetuning for language models (Gao et al., 2021a). This approach turns a prediction task into a masked language modeling problem, where the model generates a text response to a given task-specific prompt as the label. Our experiment protocol follows Gao et al. (2021a). The experiments are divided into 14 parallel experiments, each corresponding to a dataset. For the few-shot experiment, we use test split data as the target task data and sample 16 examples per class from the train split as finetuning data. The evaluation metric is measured by prompt-based prediction accuracy.

During the testing stage, we conduct experiments in zero-shot and few-shot settings for a given dataset. In the zero-shot setting, we directly evaluate the model's prompt-based prediction accuracy. In the few-shot setting, we finetune the model using support samples from the same dataset and assess its accuracy on the test split. For multitask finetuning, we select support samples from other datasets

| | SST-2 (acc) | SST-5 (acc) | MR (acc) | CR (acc) | MPQA (acc) | Subj (acc) | TREC (acc) | CoLA (Matt.) |
|---|---|---|---|---|---|---|---|---|
| Prompt-based zero-shot | 83.6 | 35.0 | 80.8 | 79.5 | 67.6 | 51.4 | 32.0 | 2.0 |
| Multitask FT zero-shot | **92.9** | 37.2 | 86.5 | 88.8 | 73.9 | 55.3 | 36.8 | -0.065 |
| Prompt-based FT† | 92.7 (0.9) | 47.4 (2.5) | 87.0 (1.2) | 90.3 (1.0) | 84.7 (2.2) | **91.2** (1.1) | 84.8 (5.1) | **9.3** (7.3) |
| Multitask Prompt-based FT | 92.0 (1.2) | **48.5** (1.2) | 86.9 (2.2) | 90.5 (1.3) | **86.0** (1.6) | 89.9 (2.9) | 83.6 (4.4) | 5.1 (3.8) |
| + task selection | 92.6 (0.5) | 47.1 (2.3) | **87.2** (1.6) | **91.6** (0.9) | 85.2 (1.0) | 90.7 (1.6) | **87.6** (3.5) | 3.8 (3.2) |
| | MNLI (acc) | MNLI-mm (acc) | SNLI (acc) | QNLI (acc) | RTE (acc) | MRPC (F1) | QQP (F1) | |
| Prompt-based zero-shot | 50.8 | 51.7 | 49.5 | 50.8 | 51.3 | 61.9 | 49.7 | |
| Multitask FT zero-shot | 63.2 | 65.7 | 61.8 | 65.8 | 74.0 | 81.6 | 63.4 | |
| Prompt-based FT† | 68.3 (2.3) | 70.5 (1.9) | 77.2 (3.7) | 64.5 (4.2) | 69.1 (3.6) | 74.5 (5.3) | 65.5 (5.3) | |
| Multitask Prompt-based FT | 70.9 (1.5) | 73.4 (1.4) | **78.7** (2.0) | 71.7 (2.2) | **74.0** (2.5) | **79.5** (4.8) | 67.9 (1.6) | |
| + task selection | **73.5** (1.6) | **75.8** (1.5) | 77.4 (1.6) | **72.0** (1.6) | 70.0 (1.6) | 76.0 (6.8) | **69.8** (1.7) | |

Table 15: **Results of few-shot learning with NLP benchmarks.** All results are obtained using RoBERTa-large. We report mean (and standard deviation) of metrics over 5 different splits. †: Result in Gao et al. (2021a); FT: finetuning; task selection: select multitask data from customized datasets.

and construct tasks for prompt-based finetuning. We then evaluate the performance of the finetuned model on the target task. More details can be found in Appendix G.3.

**Task Selection.** We select datasets by using task selection algorithm of feature vectors, which are obtained by computing the representations of each dataset and analyzing their relationship. We first obtain text features for each data point in the dataset. We select few-shot samples for generating text features. For each example, we replace the masked word with the true label in its manual template, then we forward them through the BERT backbone. Then, we compute the first principal component to obtain a feature vector for each dataset. Dataset selection provides certain improvements on some datasets, as elaborated below. Further details can be found in Appendix G.4.

**Results.** Our results are presented in Table 15. Again, our method outperforms direct adaptation on target tasks across most datasets. For zero-shot prediction, our method provides improvements on all datasets except CoLA. Our multitask finetuning approach results in performance improvements on 12 out of 15 target tasks for few-shot prediction, with the exceptions being SST-2, Subj, and CoLA. CoLA is also reported by Gao et al. (2021a) as an exception that contains non-grammatical sentences that are outside of the distribution of the pretrained language model. SST-2 already achieves high accuracy in zero-shot prediction, and our model performs best in such setting. Subj is unique in that its task is to predict whether a given sentence is subjective or objective, therefore multitasking with few-shot samples from other datasets may not provide significant improvement for this task.

## G.2 DATASETS AND MODELS

The text dataset consisted of 8 single-sentence and 6 sentence-pair English tasks, including tasks from the GLUE benchmark (Wang et al., 2018), as well as 7 other popular sentence classification tasks (SNLI (Bowman et al., 2015), SST-5 (Socher et al., 2013), MR (Pang & Lee, 2005), CR (Hu & Liu, 2004), MPQA (Wiebe et al., 2005), Subj (Pang & Lee, 2004), TREC (Voorhees & Tice, 2000)). The objective was to predict the label based on a single sentence or a sentence-pair. Specifically, for single sentences, we aimed to predict their semantics as either positive or negative, while for sentence-pairs, we aimed to predict the relationship between them. We experiment with the pretrained model RoBERTa. We have 14 datasets in total. We split each dataset into train and test split, see details below. We experiment with the pretrained model RoBERTa.

We follow Gao et al. (2021a) in their train test split. We use the original development sets of SNLI and datasets from GLUE for testing. For datasets such as MR, CR, MPQA, and Subj that require a cross-validation evaluation, we randomly select 2,000 examples for testing and exclude them from training. For SST5 and TREC, we utilize their official test sets.

To construct multitask examples from support samples, we gather support samples from all datasets except the testing dataset. For each task, we randomly select ten support samples and prompt-based finetuning the model.

| Task | Template | Label words |
|------|----------|-------------|
| SST-2 | $<S_1>$ It was [MASK] . | positive: great, negative: terrible |
| SST-5 | $<S_1>$ It was [MASK] . | v.positive: great, positive: good, neutral: okay, negative: bad, v.negative: terrible |
| MR | $<S_1>$ It was [MASK] . | positive: great, negative: terrible |
| CR | $<S_1>$ It was [MASK] . | positive: great, negative: terrible |
| Subj | $<S_1>$ This is [MASK] . | subjective: subjective, objective: objective |
| TREC | [MASK] : $<S_1>$ | abbreviation: Expression, entity: Entity, description: Description |
|  |  | human: Human, location: Location, numeric: Number |
| COLA | $<S_1>$ This is [MASK] . | grammatical: correct, not_grammatical: incorrect |
| MNLI | $<S_1>$ ? [MASK] , $<S_2>$ | entailment: Yes, netural: Maybe, contradiction: No |
| SNLI | $<S_1>$ ? [MASK] , $<S_2>$ | entailment: Yes, netural: Maybe, contradiction: No |
| QNLI | $<S_1>$ ? [MASK] , $<S_2>$ | entailment: Yes, not_entailment: No |
| RTE | $<S_1>$ ? [MASK] , $<S_2>$ | entailment: Yes, not_entailment: No |
| MRPC | $<S_1>$ [MASK] , $<S_2>$ | equivalent: Yes, not_equivalent: No |
| QQP | $<S_1>$ [MASK] , $<S_2>$ | equivalent: Yes, not_equivalent: No |

Table 16: Manual templates and label words that we used in our experiments, following Gao et al. (2021a).

## G.3 EXPERIMENT PROTOCOLS

Gao et al. (2021a) proposed a prompt-based finetuning pipeline for moderately sized language models such as BERT, RoBERTa. Prompt-based prediction converts the downstream prediction task as a (masked) language modeling problem, where the model directly generates a textual response also known as a label word, to a given prompt defined by a task-specific template. As an illustration, consider the SST-2 dataset, which comprises sentences expressing positive or negative sentiment. The binary classification task can be transformed into a masked prediction problem using the template , it was <MASK>., where  represents the input sentence and <MASK> is the label word (e.g., "great" or "terrible") that the model is supposed to predict, see full templates in Table 16. Prompt-based finetuning updates the model with prompt-based prediction loss for a given example, such as a sentence or sentence-pair.

To conduct the few-shot experiment, we use all data from the test split as the target task data for each dataset, and sample 16 examples per class from the train split as the support samples. The experiments are divided into 14 parallel experiments, with each corresponding to one dataset. The evaluation accuracy is measured as the prompt-based prediction accuracy. We subsampled 5 different sets of few-shot examples to run replicates experiments and report average performance.

During the testing stage, for a given dataset (e.g. QNLI), we consider the entire test split as the target task and divide the experiment into zero-shot and few-shot settings. In the zero-shot setting, we directly evaluate the model by measuring the accuracy of prompt-based predictions. In the few-shot setting, we first prompt-based finetune the model with support samples from the same dataset (QNLI) and then evaluate the accuracy on the test split. This experimental protocol follows the same pipeline as described in Gao et al. (2021a).

To perform multitask finetuning for a target task on a particular dataset (e.g. QNLI), we select support samples from other datasets (e.g. SST-2, Subj, QQP, etc.) as finetuning examples. We construct tasks using these examples and apply the same prompt-based finetuning protocol to multitask finetune the model on these tasks. Finally, we evaluate the performance of the finetuned model on the target task.

## G.4 TASK SELECTION

The importance of the relationship between the data used in the training tasks and the target task cannot be overstated in multitask finetuning. Our theory measures this relationship through diversity and consistency statements, which require that our finetuning data are diverse enough to capture the characteristics of the test data, while still focusing on the specific regions where the test data aligns. We visualize this diversity and relationship through the feature maps of the datasets.

To visualize the relationship between feature vectors of different datasets, we first obtain text features for each data point in the dataset. We select few-shot samples for generating text features. For each example, we replace the masked word with the true label in its manual template, then we forward them through the BERT backbone. The reason for using BERT over RoBERTa is that the latter only

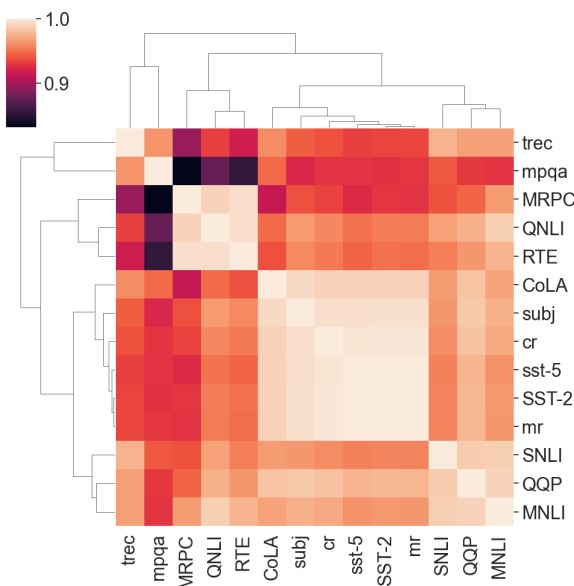

Figure 7: Linear similarity among features vectors among 14 language datasets.

| cola: mr, cr,sst-2,sst-5,subj |
|---|
| sst-2: cola,mr, cr,sst-5,subj, |
| mrpc: qnli, rte |
| qqp: snli, mnli |
| mnli: snli, qqp |
| snli: qqp, mnli |
| qnli: mrpc, rte |
| rte: mrpc, qnli |
| mr: cola, cr,sst-2,sst-5,subj |
| sst-5: cola,mr, cr,sst-2,subj |
| subj: cola,mr, cr,sst-2,sst-5 |
| trec: mpqa |
| cr: cola,mr,sst-2,sst-5,subj |
| mpqa: trec |

Table 17: Dataset selection.

has masked token prediction in pretraining, the `[CLS]` in pretrained `RoBERTa` model might not contain as much sentence information as `BERT`. Then, we compute the first principal component to obtain a feature vector for each dataset. We illustrate the relationship between these feature vectors in Figure 7.

We further perform training data selection based on the task selection algorithm among the feature vectors, the selected dataset is shown in table Table 17.

By performing task selection, we observed further improvements in multitask prompt-based finetuning on MR, CR, TREC, MNLI, QNLI, and QQP datasets. However, it's worth noting that the CoLA dataset is an exception, as it involves predicting the grammaticality of sentences, and its inputs may include non-grammatical sentences that are outside the distribution of masked language models, as

| | SST-2 (acc) | SST-5 (acc) | MR (acc) | CR (acc) | MPQA (acc) | Subj (acc) | TREC (acc) | CoLA (Matt.) |
|---|---|---|---|---|---|---|---|---|
| Prompt-based zero-shot | 83.6 | 35.0 | 80.8 | 79.5 | 67.6 | 51.4 | 32.0 | 2.0 |
| Multitask FT zero-shot | **92.9** | 37.2 | 86.5 | 88.8 | 73.9 | 55.3 | 36.8 | -0.065 |
| + task selection | 92.5 | 34.2 | 87.1 | 88.7 | 71.8 | 72.0 | 36.8 | 0.001 |
| Prompt-based FT† | 92.7 (0.9) | 47.4 (2.5) | 87.0 (1.2) | 90.3 (1.0) | 84.7 (2.2) | **91.2** (1.1) | 84.8 (5.1) | **9.3** (7.3) |
| Multitask Prompt-based FT | 92.0 (1.2) | **48.5** (1.2) | 86.9 (2.2) | 90.5 (1.3) | **86.0** (1.6) | 89.9 (2.9) | 83.6 (4.4) | 5.1 (3.8) |
| + task selection | 92.6 (0.5) | 47.1 (2.3) | **87.2** (1.6) | **91.6** (0.9) | 85.2 (1.0) | 90.7 (1.6) | **87.6** (3.5) | 3.8 (3.2) |

| | MNLI (acc) | MNLI-mm (acc) | SNLI (acc) | QNLI (acc) | RTE (acc) | MRPC (F1) | QQP (F1) | |
|---|---|---|---|---|---|---|---|---|
| Prompt-based zero-shot | 50.8 | 51.7 | 49.5 | 50.8 | 51.3 | 61.9 | 49.7 | |
| Multitask FT zero-shot | 63.2 | 65.7 | 61.8 | 65.8 | 74.0 | 81.6 | 63.4 | |
| + task selection | 62.4 | 64.5 | 65.5 | 61.6 | 64.3 | 75.4 | 57.6 | |
| Prompt-based FT† | 68.3 (2.3) | 70.5 (1.9) | 77.2 (3.7) | 64.5 (4.2) | 69.1 (3.6) | 74.5 (5.3) | 65.5 (5.3) | |
| Multitask Prompt-based FT | 70.9 (1.5) | 73.4 (1.4) | **78.7** (2.0) | 71.7 (2.2) | **74.0** (2.5) | **79.5** (4.8) | 67.9 (1.6) | |
| + task selection | **73.5** (1.6) | **75.8** (1.5) | 77.4 (1.6) | **72.0** (1.6) | 70.0 (1.6) | 76.0 (6.8) | **69.8** (1.7) | |

Table 18: **Results of few-shot learning with NLP benchmarks.** All results are obtained using RoBERTa-large. We report the mean (and standard deviation) of metrics over 5 different splits. †: Result in Gao et al. (2021a) in our paper; FT: finetuning; task selection: select multitask data from customized datasets.

noted in Gao et al. (2021a). Overall, our approach shows promising results for multitask learning in language tasks.

### G.4.1 FULL RESULTS WITH TASK SELECTION

To complement task selection in Table 15, we provide full results here and explain each method thoroughly.

We first elaborate on what each method did in each stage. During the testing stage, we conducted experiments in zero-shot and few-shot settings for a given dataset following Gao et al. (2021a), who applied prompt-based methods on moderately sized language models such as RoBERTa. Prompt-based finetuning method updates the model with prompt-based prediction loss for a given example. The given example can either be from a testing dataset or other datasets.

Table 18 shows our multitask finetuning and task selection provide helps on target tasks, as detailed in Appendix G.1. We will elaborate on what each method did in the "Multitask fintuning phase" and "Downstream phase".

In the "Multitask fintuning phase": For prompt-based zero-shot (col-1) and prompt-based FT (col-4) we do not finetune any model. For Multitask Prompt-based finetuning (col-2,3,5,6), we conduct prompt-based finetuning methods using finetuning(auxiliary) tasks. The data of tasks are from datasets other than testing datasets. For instance, consider a model designated to adapt to a dataset (say SST-2), we choose data from other datasets (mr, cr, etc. ) and combine these data together and form multiple auxiliary tasks, these tasks updated the model using prompt-based finetuning methods. In the "downstream phase" where we adapt the model: In the zero-shot setting (col-1,2,3), we directly evaluate the model's prompt-based prediction accuracy. In the few-shot setting (col-4,5,6), we finetune the model using shot samples from the same dataset (sst-2) and assess its accuracy on the test split.

### G.4.2 ADDITIONAL RESULTS ON SIMCSE

We present our results using the same approach as described in our paper. However, we used a different pretrained loss, namely simCSE, as proposed by Gao et al. (2021b). However, the results are not promising, The reason is simCSE is trained with a contrastive loss instead of masked language prediction, making it less suitable for prompt-based finetuning.

## H   VISION LANGUAGE TASKS

Pretrained vision-language as another type of foundation model has achieved tremendous success across various downstream tasks. These models, such as CLIP (Radford et al., 2021) and ALIGN (Jia

| | SST-2 (acc) | SST-5 (acc) | MR (acc) | CR (acc) | MPQA (acc) | Subj (acc) | TREC (acc) | CoLA (Matt.) |
|---|---|---|---|---|---|---|---|---|
| Prompt-based zero-shot | 50.9 | 19.3 | 50 | 50 | 50 | 50.4 | **27.2** | 0 |
| Multitask FT zero-shot | 51.3 | 13.8 | 50 | 50 | 50 | 50.6 | 18.8 | 0 |
| Prompt-based FT[†] | **51.8**(2.6) | 20.5 (6.1) | **50.6** (0.8) | 50.8 (1.1) | 52.3 (1.9) | **55.4** (3.7) | 19.8 (7.3) | 0.8 (0.9) |
| Multitask Prompt-based FT | 50.6 (0.7) | **22.1** (6.2) | 50.5 (1.0) | 51.5 (1.7) | **53.4** (2.7) | 51.0 (1.4) | 26.4 (8.5) | **0.9** (1.3) |
| + task selection | 51.7 (1.7) | 19.7 (5.6) | **50.6** (0.8) | **51.6** (1.6) | 52.3 (2.7) | 54.7 (2.5) | 23.2 (9.9) | 0.5 (0.7) |

| | MNLI (acc) | MNLI-mm (acc) | SNLI (acc) | QNLI (acc) | RTE (acc) | MRPC (F1) | QQP (F1) | |
|---|---|---|---|---|---|---|---|---|
| Prompt-based zero-shot | 35.4 | 35.2 | 33.8 | 50.5 | 47.3 | 1.4 | 1.5 | |
| Multitask FT zero-shot | 35.4 | 35.2 | 33.6 | 49.5 | 47.3 | 53.8 | 53.8 | |
| Prompt-based FT[†] | 32.9 (0.8) | 33.0 (0.7) | 33.7 (0.6) | **50.6** (1.4) | 48.7 (3.7) | **79.2** (4.1) | 53.5 (2.7) | |
| Multitask Prompt-based FT | 32.5 (0.6) | 32.5 (0.7) | 33.5 (0.4) | **50.6** (2.4) | 50.0 (2.0) | 76.3 (6.5) | **54.2** (0.8) | |
| + task selection | **33.2** (1.2) | **33.2** (1.1) | **35.0** (0.8) | 50.3 (0.4) | **51.8** (2.0) | 72.2 (10.8) | 52.9 (3.0) | |

Table 19: Our main results using simCSE (Gao et al., 2021b). We report mean (and standard deviation) performance over 5 splits of few-shot examples. FT: fine-tuning; task selection: select multitask data from customized dataset.

et al., 2021), align images and text in a shared space, enabling zero-shot classification in target tasks. Finetuning such models has resulted in state-of-the-art accuracy in several benchmarks.

Vision-language model enables the classification of images through prompting, where classification weights are calculated by a text encoder. The text encoder inputs text prompts containing class information, and outputs features aligned with features from the vision encoder in the same space.

However, standard finetuning can be affected by minor variations underperforming direct adaptation (Kumar et al., 2022; Wortsman et al., 2022). Additionally, standard finetuning can be computationally expensive, as it requires training the model on a large amount of target task data.

We perform our multitask finetuning pipeline on the vision-language model and observe certain improvements. It's worth mentioning although the vision-language model is pretrained using contrastive learning, the model does not align with our framework. Vision-language model computes contrastive loss between image and text encoder, whereas our pretraining pipeline formulates the contrastive loss between the same representation function $\phi$ for positive and negative sample pairs. Despite the discrepancy, we provide some results below.

### H.1 IMPROVING ZERO-SHOT PERFORMANCE

We investigate the performance of CLIP models in a zero-shot setting, following the established protocol for our vision tasks. Each task includes 50 classes, with one query image per class. We employ text features combined with class information as the centroid to categorize query images within the 50 classes. During adaptation, we classify among randomly selected classes in the test split, which consists of 50 classes.

We experimented with our methods on tieredImageNet and DomainNet. The text template utilized in *tieredImageNet* was adapted from the CLIP documentation. In adaptation, we classify among all classes in the test split (160 classes in tieredImageNet and 100 classes in DomainNet). For text features on tieredImageNet, we use 8 templates adapted from CLIP `a photo of a {}, itap of a {}, a bad photo of the {}, a origami {}, a photo of the large {}, a {} in a video game, art of the {}, a photo of the small {}.` For templates on *DomainNet*, we simply use `a photo of a {}`. In the DomainNet The text template used for this experiment is `"a photo of {}"`. We perform Locked-Text Tuning, where we fixed the text encoder and update the vision encoder alone.

Table 20 demonstrates that CLIP already exhibits a high level of zero-shot performance. This is due to the model classifying images based on text information rather than relying on another image from the same class, which enables the model to utilize more accurate information to classify among query images. We show the effectiveness of zero-shot accuracy in tieredImageNet and DomainNet. It is worth highlighting that our multitask finetuning approach enhances the model's zero-shot performance, particularly on the more realistic DomainNet dataset. We have observed that our

| Backbone | Method | tieredImageNet | DomainNet |
|----------|--------|----------------|-----------|
| ViT-B | Adaptation | 84.43 (0.25) | 70.93 (0.32) |
| | Ours | 84.50 (0.25) | 73.31 (0.30) |
| ResNet50 | Adaptation | 81.01 (0.28) | 63.61 (0.34) |
| | Ours | 81.02 (0.27) | 65.55 (0.34) |

Table 20: Multitask finetune on zero-shot performance with CLIP model.

multitask finetuning pipeline yields greater improvements for tasks on which the model has not been extensively trained.

## H.2 UPDATING TEXT ENCODER AND VISION ENCODER

We also investigated whether updating the text encoder will provide better performance. On the tieredImageNet dataset, We finetune the text encoder and vision encoder simultaneously using the contrastive loss, following the protocol in Goyal et al. (2023).

| Method | Zero-shot | Multitask finetune |
|--------|-----------|--------------------|
| Accuracy | 84.43 (0.25) | 85.01 (0.76) |

Table 21: Multitask finetune on zero-shot performance with ViT-B32 backbone on tieredImageNet.

In Table 21, we observed slightly better improvements compared to updating the vision encoder alone. We anticipate similar performance trends across various datasets and backbone architectures. We plan to incorporate these findings into our future work.

## H.3 COCOOP

We also multitask finetune the vision language model following the protocol outlined in Zhou et al. (2022a). This approach involved prepending an image-specific token before the prompt to enhance prediction accuracy. To generate this token, we trained a small model on the input image. We evaluate the performance of our model on all classes in the test split, which corresponds to a 160-way classification task. This allows us to comprehensively assess the model's ability to classify among a large number of categories.

| Method | Zero-shot | Multitask finetune |
|--------|-----------|--------------------|
| ViT-B32 | 69.9 | 71.4 |

Table 22: Multitask finetune on zero-shot performance with ViT-B32 backbone on tieredImageNet.

Table 22 showed the result of the performance of the CoCoOp method. We observed an improvement of 1.5% in accuracy on direct adaptation.

